# INCENTIVE-AWARE FEDERATED LEARNING WITH TRAINING-TIME MODEL REWARDS

**Zhaoxuan Wu**[†]**, Mohammad Mohammadi Amiri**[¶]**, Ramesh Raskar**[§] **& Bryan Kian Hsiang Low**[‡]

[†]Institute of Data Science, National University of Singapore, Republic of Singapore
[¶]Department of Computer Science, Rensselaer Polytechnic Institute, USA
[§]Media Lab, Massachusetts Institute of Technology, USA
[‡]Department of Computer Science, National University of Singapore, Republic of Singapore
{wu.zhaoxuan,lowkh}@comp.nus.edu.sg[†‡] mamiri@rpi.edu[¶] raskar@mit.edu[§]

## ABSTRACT

In federated learning (FL), incentivizing contributions of training resources (e.g., data, compute) from potentially competitive clients is crucial. Existing incentive mechanisms often distribute post-training monetary rewards, which suffer from practical challenges of timeliness and feasibility of the rewards. Rewarding the clients after the completion of training may incentivize them to abort the collaboration, and monetizing the contribution is challenging in practice. To address these problems, we propose an incentive-aware algorithm that offers differentiated training-time model rewards for each client at each FL iteration. We theoretically prove that such a *local* design ensures the *global* objective of client incentivization. Through theoretical analyses, we further identify the issue of error propagation in model rewards and thus propose a stochastic reference-model recovery strategy to ensure theoretically that all the clients eventually obtain the optimal model in the limit. We perform extensive experiments to demonstrate the superior incentivizing performance of our method compared to existing baselines.

## 1 INTRODUCTION

*Federated learning* (FL) is a popular framework that fosters collaboration among distributed clients while keeping the raw data on their local devices (McMahan et al., 2017). The clients perform local optimization on local data, while the server performs centralized parameter updates by aggregating these local model updates (Li et al., 2020a). It is essential to motivate potentially competing clients to contribute their training resources since they incur nontrivial costs for data collection (Sim et al., 2020), local computation (Sarikaya & Ercetin, 2020), and federated communication (Lim et al., 2020). Moreover, self-interested clients may refrain from contributing to the best of their abilities or drop out of the FL due to insufficient rewards (Zhan et al., 2022), which can delay model training and worsen model performance (Tu et al., 2022). To address these issues, a number of incentive mechanisms have been proposed (Tu et al., 2022; Khan et al., 2020; Zhang et al., 2021; Liu et al., 2023).

Most existing incentive mechanisms distribute external resources (e.g., money) post-training (Zhan et al., 2022; Tu et al., 2022), which poses practical challenges regarding the *timeliness* and *feasibility* of the incentives. Firstly, rewarding clients only after the completion of the FL process can discourage their participation as they usually anticipate timely rewards due to the continuous contribution of costly resources throughout the process (IMDA, 2019). It is also impractical to withhold compensation for clients until the end of the FL process as they have the freedom to exit the collaboration in the middle of the process. Secondly, monetary incentives are often infeasible in situations where the source of revenue is unclear, the budget is limited (Sim et al., 2020), and the contribution-to-dollar value denomination is difficult to determine (Xu et al., 2021). To tackle these challenges, it is vital to design training-time model rewards at each iteration of the FL process to achieve the overall incentivization goal. However, few work have explored this direction (Xu et al., 2021; Kong et al., 2022), and their heuristic methods overlook the important theoretical implications of local design choices on the performance of rewarded models and thus client incentivization.

A central issue is the *global-to-local* design: *How should each client be rewarded locally in each FL iteration, given the global incentivization objective?* Inspired by the game-theoretic insights,

we propose that the proportion of local model updates, which a client can aggregate as a local reward, should be commensurate with his contribution in each FL iteration. Therefore, clients receive different models (each aggregated using a specific proportion of local model updates) such that higher-contributing clients receive models aggregated from a larger proportion of local model updates. We prove this local reward scheme ensures that a higher-contributing client gets a final model with a better performance guarantee. However, we observe an undesirable phenomenon of error propagation from the performance bound: Low-contributing clients can worsen every client model as aggregating low-quality model updates can adversely influence the high-quality models (Deng et al., 2022).

This phenomenon implies that achieving the incentivization goal prevents the client models from reaching optimality. Ideally, our algorithm should allow clients to eventually get the globally optimal model. *How to adjust our local reward scheme if we want every client to obtain the best model in the limit yet without hurting their incentives?* To address this challenge, we propose a reference-model recovery strategy that stochastically provides every client with the same reference model at each FL iteration. This approach mitigates the error propagation, and we further prove that all the client models asymptotically converge to the global optimum as long as their contributions do not diminish "too quickly" with the FL iterations. Consequently, every client is better off and is expected to eventually receive the best model while still being incentivized to contribute in finite FL iterations.

In summary, we propose an *incentive-aware federated learning* (IAFL) algorithm with training-time model rewards commensurate with client contributions, while being agnostic to any contribution measures (Section 4). Then, we theoretically justify our design choices through convergence analyses (Section 5). Finally, we demonstrate through extensive experiments that our method enjoys superior incentivizing performance compared to other baselines (Section 6).

## 2 RELATED WORKS

**Mechanism design for FL incentives.** Popular tools to model the behaviors of the server and clients include Stackelberg games (Khan et al., 2020; Zhan et al., 2020), auctions (Zhang et al., 2021; Cong et al., 2020) and contract theory (Liu et al., 2023; Kang et al., 2019), which all utilize post-training monetary incentive schemes to motivate client contributions (Zhan et al., 2022; Tu et al., 2022). Karimireddy et al. (2022) suggested post-training model rewards of various utilities achieved through noisy model perturbation. However, this method may not be effective in FL since clients have already obtained the best global model during the model broadcasting of the FL process. Our paper instead directly integrates the incentive mechanism into the FL algorithm using training-time model rewards.

**Heterogeneity and personalized FL.** Data heterogeneity among distributed clients is a common challenge in FL (Wang et al., 2020a; Chen et al., 2022). Methods like FedProx (Li et al., 2020b) and FedNova (Wang et al., 2020a) train a single shared model that aligns clients' potentially mismatched objectives caused by heterogeneous local data distributions. Personalization in FL yields a personalized model for each client and focuses on improving performances on local test sets (Tan et al., 2022). Personalized models are achieved by personalizing layers or structures of the shared model (Liang et al., 2020; Collins et al., 2021; Li et al., 2021; Pillutla et al., 2022). In contrast, we focus on tailored client model rewards trained by optimizing a shared global objective with data heterogeneity (e.g., a client with only MNIST digits from class 0-2 is still interested in a model that classifies all digits).

**Training-time model incentives for FL.** CGSV (Xu et al., 2021) and Rank (Kong et al., 2022) have explored training-time model incentives during FL. However, their heuristic approaches lack theoretical convergence analyses. Additionally, Rank impractically assumes access to a validation set to rank local model performances for aggregation. CGSV presents a contrasting perspective to ours: CGSV zeroes out partial model update parameter values for all clients, while our IAFL effectively zeroes out partial client model updates for all parameter values. Our aggregation approach is more aligned with the partial client participation setting in FL (Li et al., 2020d). CGSV's local performance guarantee cannot be generalized to the global incentivization objective for the entire FL training.

## 3 NOTATIONS AND BACKGROUNDS

We consider $N$ federated clients collaboratively learning a predictive model $\boldsymbol{\theta} \in \mathbb{R}^d$. Client $i$ has a local dataset $\mathcal{D}_i$ of size $B_i$ such that the grand dataset $\mathcal{D} = \cup_{i=1}^{N} \mathcal{D}_i$ has size $B := \sum_{i=1}^{N} B_i$. The global

loss $F(\boldsymbol{\theta}) = \sum_{i=1}^{N} w_i F_i(\boldsymbol{\theta})$ is a weighted sum of local loss functions $F_i(\boldsymbol{\theta}) = \frac{1}{B_i} \sum_{d \in \mathcal{D}_i} f_i(\boldsymbol{\theta}, d)$ where $f_i(\cdot, \cdot)$ is an empirical loss function. Let $\boldsymbol{\theta}_t$ denote the global model at FL iteration $t$. FL conducts two steps for $T$ iterations: (1) Clients download the global model $\boldsymbol{\theta}_{i,t}^1 = \boldsymbol{\theta}_t$ where $\boldsymbol{\theta}_{i,t}^j$ denotes the model at client $i$ before the $j$-th step of the local update; (2) Clients perform $\tau$ steps of local updates according to $\boldsymbol{\theta}_{i,t}^{j+1} = \boldsymbol{\theta}_{i,t}^j - \eta_t \nabla F_i(\boldsymbol{\theta}_{i,t}^j, \xi_{i,t}^j)$ where $\eta_t$ is the learning rate and $\xi_{i,t}^j$ is the local batch randomly chosen for the stochastic gradient descent step. Then, each client $i$ uploads the model update $\boldsymbol{g}_{i,t} = \boldsymbol{\theta}_{i,t}^{\tau+1} - \boldsymbol{\theta}_{i,t}^1 = -\eta_t \sum_{j=1}^{\tau} \nabla F_i(\boldsymbol{\theta}_{i,t}^j, \xi_{i,t}^j)$ to the server.

**Game-theoretic formulation for incentivization.** Instead of assuming benevolent clients as in the standard FL, we consider a more practical setting where a client $i$ has the freedom to strategically decide his resource contribution $p_i \in \mathbb{R}_{\geq 0}$. Let $\boldsymbol{p}_{-i} = [p_1, p_2, \ldots, p_{i-1}, p_{i+1}, \ldots, p_N] \in \mathbb{R}_{\geq 0}^{N-1}$, we define $P_{-i} \in \mathbb{R}_{\geq 0}$ as the aggregate contribution of clients other than $i$. We define a rewarding function $\nu(p_i, P_{-i})$ that is continuous, non-decreasing and concave on both arguments, reflecting the diminishing marginal return of contribution. It is usually observed in machine learning that the marginal increase in model accuracy diminishes with the increase in training dataset size (Wang et al., 2021; De & Chakrabarti, 2022). Furthermore, a client $i$ incurs a non-trivial constant cost $c_i > 0$ for offering some marginal contribution (Sim et al., 2020) and the cost is essential for the clients to decide an optimal amount of contribution that yields the highest utility (i.e., benefit minus cost). Practically, $c_i$ could account for the cost associated with collecting an additional data sample, computing the gradients for a larger dataset, or increasing the rate of partial federated participation.

# 4  INCENTIVE-AWARE FEDERATED LEARNING

Standard FL focuses on training a globally shared model and neglects the incentives for clients to join the FL procedure. We rethink the server-centric objective of FL and aim to develop a client-centric procedure to incentivize participation. The main idea is to offer training-time model rewards that are commensurate with client contributions through our proposed mechanism implementation.

## 4.1  INCENTIVE MECHANISM

When multiple clients are free to decide their contributions, clients' behaviors depend on the rewarding strategy formalized as a mechanism $\mathcal{M}_\nu(\boldsymbol{p}) : \mathbb{R}_{\geq 0}^N \to [0, 1]^N$. The rewarding mechanism $\mathcal{M}_\nu(\boldsymbol{p}) = \nu(p_i, P_{-i})$ maps the contribution vector $\boldsymbol{p}$ to the corresponding reward values using the rewarding function $\nu$. In standard FL, the global model is synced with all the clients at each FL iteration. Clients lose their incentives to increase $p_i$ since they receive the best aggregate model regardless of contributions. Therefore, we have $P_{-i} = h(\boldsymbol{p}_{-i})$ where the aggregate function $h$ only depends on $\boldsymbol{p}_{-i}$. Our idea is to make $P_{-i}$ dependent on $p_i$ by defining a new aggregate function $\tilde{h}(p_i, \boldsymbol{p}_{-i})$.

**Proposition 1.** *If $\tilde{h}$ satisfies $\frac{d\tilde{h}(p_i, \boldsymbol{p}_{-i})}{dp_i} > 0$ and $\frac{\partial \nu(p_i, \tilde{h}(p_i, \boldsymbol{p}_{-i}))}{\partial \tilde{h}(p_i, \boldsymbol{p}_{-i})} > 0$, then a mechanism $[\mathcal{M}_\nu(\boldsymbol{p})]_i = \nu(p_i, \tilde{h}(p_i, \boldsymbol{p}_{-i}))$ incentivizes a client to contribute more than that in the standard FL mechanism.*

The proof is in Appendix B.1. Specifically, $d\tilde{h}(p_i, \boldsymbol{p}_{-i})/dp_i > 0$ implies that a client $i$ should benefit more from other clients if his contribution $p_i$ is larger, and $\partial \nu(p_i, \tilde{h}(p_i, \boldsymbol{p}_{-i}))/\partial \tilde{h}(p_i, \boldsymbol{p}_{-i}) > 0$ indicates the feasibility of improving the reward from increasing the aggregate contribution of other clients. Proposition 1 reveals possible better designs of the mechanism that incentivize clients to contribute more resources. This result aligns with Karimireddy et al. (2022), which showed that the standard FL mechanism leads to catastrophic free-riding where only the lowest cost client contributes.

**Mechanism design.** Following Proposition 1, we propose to design a function $\tilde{h}$ and a mechanism to reward clients based on contributions. Intuitively, we allow clients with higher contribution $p_i$ to be rewarded based on a larger proportion of the aggregate contribution of other clients $h(\boldsymbol{p}_{-i})$. We first propose $[\mathcal{M}_\nu(\boldsymbol{p})]_i = \nu(p_i, (p_i / \max_j p_j) h(\boldsymbol{p}_{-i}))$. However, this mechanism with a relative rewarding strategy may result in an undesirable equilibrium. It is possible that clients converge to an equilibrium where all clients give similar but little contributions. Such equilibria are undesirable as little contribution may translate to a bad model. The proof for the existence of undesirable equilibria is provided in Proposition 3 in Appendix B.2. To circumvent this problem, we propose to replace

$\max_j p_j$ with a pre-defined contribution ceiling $p_{\text{ceil}}$ and a tunable sharing coefficient $\kappa \in [0, 1]$,

$$[\mathcal{M}_\nu(\boldsymbol{p})]_i = \nu \left( p_i, (\min\{p_i/p_{\text{ceil}}, 1\})^{1-\kappa} h(\boldsymbol{p}_{-i}) \right). \tag{1}$$

Here, $p_{\text{ceil}}$ can be viewed as a hypothetical upper limit for a client's contribution imposed by the mechanism. For example, $p_{\text{ceil}} = 1$ implies full client participation in all FL iterations if $p_i$ measures the rate of client participation. As for the sharing coefficient, $\kappa$ determines the extent of sharing enforced by the server. A larger $\kappa$ makes the rewards more equitable across clients. One limitation of this design is that the clients lose incentive for $p_i > p_{\text{ceil}}$. The server can mitigate this by setting a high $p_{\text{ceil}}$ upper limit (to keep incentivizing high contributors) while setting a larger $\kappa$ (to even out the fractions $(p_i/p_{\text{ceil}})^{1-\kappa}$ towards 1 to motivate low contributors).

To interpret, $h(\boldsymbol{p}_{-i})$ can be seen as local model updates from other clients (i.e., other than client $i$) in FL and a client with higher contribution $p_i$ is rewarded a model trained with a broader range of local model updates. If an averaging aggregation strategy is adopted, the model of a higher-contributing is updated using an aggregated average from a larger proportion of local model updates.

**Definition 1** (Individual Rationality (IR) in FL). *A mechanism $\mathcal{M}_\nu(\boldsymbol{p})$ satisfies individual rationality if $[\mathcal{M}_\nu(\boldsymbol{p})]_i \geq \nu(p_i, h(\mathbf{0}))$ for any $i$ and $\boldsymbol{p}$. That is, any client will receive a reward at least as good as what they can achieve on their own.*

**Remark 1.** *Our mechanism (1) satisfies IR (Definition 1), an established concept for player incentivization now adapted to FL. In our FL process, a client $i$ always utilizes the resource $p_i$ that he contributes. This essential property of our mechanism ensures the participation of all rational agents.*

## 4.2 IMPLEMENTATION OF THE MECHANISM IN FL

We present an implementation of mechanism (1) in the FL algorithm. We extend the setting in Section 4.1 to allow a client $i$ to vary its contribution $p_{i,t}$ across iterations $t$ (instead of a fixed $p_i$). Aligning with the conditions in Proposition 1, we propose the idea that, instead of sharing the global model in each iteration, the server shares updates of varying qualities with the clients. Accordingly, a client with a higher contribution receives a higher quality model. This can be achieved by first computing a client reward rate $\gamma_{i,t} = (\min\{p_{i,t}/p_{\text{ceil}}, 1\})^{1-\kappa} = \min\{(p_{i,t}/p_{\text{ceil}})^{1-\kappa}, 1\}$ and then using it to vary the proportion of local model updates $\{\boldsymbol{g}_{i,t}\}_{i=1}^N$ that a client can average from in iteration $t$. To ensure convergence to the global optimum, we additionally introduce a stochastic recovery (with probability $q$) of client models using a flexible user-defined reference model $\boldsymbol{\theta}_{\text{ref},t}$ trained with custom induced reward rates $\gamma'_{\text{ref},t}$. We defer the justification and details of the stochastic recovery technique to Section 5.1. The details of the *incentive-aware federated learning* (IAFL) algorithm are in Algorithm 1, where $[N] := \{1, 2, ..., N\}$ and $\lceil \cdot \rceil$ denotes the ceiling.

---

**Algorithm 1:** Incentive-Aware Federated Learning

1   Initialize $\boldsymbol{\theta}_{i,0} = \boldsymbol{\theta}_{\text{ref},0} = \boldsymbol{\theta}_0$ and $\boldsymbol{g}_{i,0} = \mathbf{0}$;
2   **for** $t = 1$ **to** $T$ **do**
3     Perform Procedure 2 such that $\{\Delta\boldsymbol{\theta}_{i,t-1}\}_{i=1}^N, \boldsymbol{\theta}_{\text{ref},t-1} = \text{ServerAgg}(t-1)$;
4     **foreach** *client $i$* **do**
5       **with probability $q$ do**
6         Server shares $\boldsymbol{\theta}_{\text{ref},t-1}$ with client $i$;
7         $\boldsymbol{\theta}_{i,t} = \boldsymbol{\theta}_{\text{ref},t-1}$;
8       **else**
9         Server shares $\Delta\boldsymbol{\theta}_{i,t-1}$ with client $i$;
10        $\boldsymbol{\theta}_{i,t} = \boldsymbol{\theta}_{i,t-1} + \Delta\boldsymbol{\theta}_{i,t-1}$;
11       $\boldsymbol{\theta}_{i,t}^{\tau+1} = \boldsymbol{\theta}_{i,t} - \eta_t \sum_{j=1}^\tau \nabla F_i(\boldsymbol{\theta}_{i,t}^j, \xi_{i,t}^j)$;
12       Upload the local update $\boldsymbol{g}_{i,t} = \boldsymbol{\theta}_{i,t}^{\tau+1} - \boldsymbol{\theta}_{i,t}$;

**Procedure 2:** ServerAgg($t$)

1   **foreach** *client $i$* **do**
2     Compute $\gamma_{i,t} = \min\left\{ (p_{i,t}/p_{\text{ceil}})^{1-\kappa}, 1 \right\}$;
3     Sample $\mathcal{S}'_{i,t} \subseteq \{j : j \in [N], j \neq i\}$ randomly s.t. $|\mathcal{S}'_{i,t}| = \lceil \gamma_{i,t}(N-1) \rceil$;
4     Include client $i$'s update $\mathcal{S}_{i,t} = \{i\} \cup \mathcal{S}'_{i,t}$;
5     $\Delta\boldsymbol{\theta}_{i,t} = \frac{1}{|\mathcal{S}_{i,t}|} \sum_{j \in \mathcal{S}_{i,t}} \boldsymbol{g}_{j,t}$;
6   Sample $\mathcal{S}_{\text{ref},t} \subseteq [N]$ randomly such that $|\mathcal{S}_{\text{ref},t}| = \lceil \gamma'_{\text{ref},t} N \rceil$;
7   $\boldsymbol{\theta}_{\text{ref},t} = \boldsymbol{\theta}_{\text{ref},t-1} + \frac{1}{|\mathcal{S}_{\text{ref},t}|} \sum_{j \in \mathcal{S}_{\text{ref},t}} \boldsymbol{g}_{j,t}$;
8   **return** $\{\Delta\boldsymbol{\theta}_{i,t}\}_{i=1}^N, \boldsymbol{\theta}_{\text{ref},t}$

---

An intuition for the connection between the mechanism in (1) and the IAFL algorithm is provided in Appendix C. Overall, clients keep different versions of the model and update their respective models using the partially averaged model updates rewarded to them over the FL iterations. The models have varying qualities as the updates rewarded to the clients are crafted to match their contributions. This approach incentivizes clients to increase their contributions to the learning process.

**Contribution measurement.** Our IAFL is agnostic to contribution measures and distributes training-time model rewards that are commensurate with client contribution $p_{i,t}$. This is beneficial because clients have the option to define a custom contribution measure collaboratively. Besides the conventional measures depending on dataset size or participation rate, it is possible to utilize external knowledge to value contributions. For example, meta attributes such as the timeliness of the data, reliability of the client network and closeness of clients' relationships are all possible considerations of the contribution measure. Importantly, our method also gives the flexibility to vary $p_i$ with FL iteration $t$ through $p_{i,t}$. Our IAFL is also plug-and-play with existing FL contribution evaluation methods capturing the quality of local model updates, such as FedSV (Wang et al., 2020b), ComFedSV (Fan et al., 2022), CGSV (Xu et al., 2021), FedFAIM (Shi et al., 2022), R-RCCE (Zhao et al., 2021), etc.

## 5 THEORETICAL ANALYSIS

This section provides theoretical justification for IAFL via convergence analysis. Our training-time local reward scheme ensures that a higher-contributing client receives a final model with a better performance guarantee. We further propose a stochastic reference-model recovery strategy to mitigate the theoretically observed *error propagation* problem and ensure the asymptotic convergence of client models to the global optimum as $T \to \infty$. For ease of analysis, we consider full client participation, where the results can be extended to partial client participation following Li et al. (2020d). We define an induced reward rate $\gamma'_{i,t} = \gamma_{i,t} - (\gamma_{i,t} - 1)/N$, which represents the fraction of the local model updates used by client $i$ at FL iteration $t$.

### 5.1 PERFORMANCE GUARANTEE

Formalized in Assumptions 1-4 (Appendix D), we assume that the loss functions $F_i(\cdot)$ are $L$-smooth and $\mu$-strongly convex. The average expected variance of stochastic gradients is $\tilde{\Sigma}^2$, and the squared $l_2$-norm of stochastic gradients is bounded by $G$. We let each client $i$ have the equal weight $w_i = 1/N$ in the global objective $F$ and further use $\Gamma = F^* - \sum_{i=1}^{N} w_i F_i^*$ to denote the extent of data heterogeneity where $F^*$ and $F_i^*$ are the optimum values of $F$ and $F_i$, respectively.

**Theorem 1** (Performance bound). *With full clients participation and a decreasing learning rate* $\eta_t = \frac{1}{\mu\tau(t+\alpha)}$ *where* $\alpha \geq \frac{4L(\tau+1)}{\mu\tau}$, *define* $B = 2L\tau(2\tau+3)\Gamma + 2\tau^3 G^2 + \tau^2\tilde{\Sigma}^2 + (\alpha+1)\mu^2\tau^2\mathbb{E}[\|\boldsymbol{\theta}_1 - \boldsymbol{\theta}^*\|^2]$ *and* $H_T = (2 - 2\mu\eta_T)\tau^3 G^2$, *then the performance of the client model* $\boldsymbol{\theta}_{i,T}$ *trained over $T$ FL iterations using IAFL is bounded by*

$$\mathbb{E}\left[F(\boldsymbol{\theta}_{i,T})\right] - F^* \leq \frac{L}{2\mu^2\tau^2} \frac{B + C_T}{\alpha + T} \text{ where } C_T = \frac{H_T}{N} \sum_{m=1}^{N} \sum_{t=1}^{T} \left(\frac{T+\alpha}{t+\alpha}\right)^2 \left(\frac{1}{\gamma'_{m,t}} + \frac{1}{\gamma'_{i,t}} - 2\right).$$

The proof is given in Appendix D.1. Here, $B$ is constant w.r.t $T$ and the client contributions only affect $\gamma'_{i,t}$ and thus $C_T$. To interpret, our IAFL ensures that a higher-contributing client (i.e., larger $\gamma'_{i,t}$) receives a model with a better performance guarantee measured in terms of the expected performance gap to the global optimum. This aligns with our goal of rewarding higher-contributing clients more and hence incentivizing client participation and contributions. However, we observe that $C_T$ grows with FL iterations $T$ and causes non-convergence of client model $\boldsymbol{\theta}_{i,T}$. This undesirable behavior is caused by the lack of model synchronization and we next propose a remedy to the problem.

**Error propagation and non-convergence.** The non-convergence is seen in Theorem 1 as $T \to \infty$. Asymptotically, we require $C_T/T \to 0$ for the algorithm to converge to $F^*$. Since we already have $(H_T/T)\sum_{t=1}^{T} T^2/t^2 = O(T)$, we need $1/\gamma'_{i,T} = o(1/T)$ for the convergence to hold, where the big-$O$ and small-$o$ notations are defined in Definition D.1 and Definition D.2. However, this cannot be achieved as $\gamma'_{i,t} \leq 1, \forall t$. Consequently, we instead require that $\gamma'_{i,t} = 1$ for all $i \in [N], t \in [T]$, which corresponds to the situation where all the clients contribute at their maximum capacity.

**Corollary 1.** *When all the clients contribute at their maximum capacity, i.e.,* $p_{i,t} = p_{ceil}$ *for all* $i \in [N], t \in [T]$, *we have* $C_T = 0$ *and achieve asymptotic convergence* $\lim_{T\to\infty} \mathbb{E}\left[F(\boldsymbol{\theta}_{i,T})\right] - F^* = 0$.

Corollary 1 states a very strong assumption that recovers the FedAvg algorithm: Even when only one client fails to contribute at the maximum, all client models cannot be guaranteed to converge to the

optimum, including for the highest-contributing client. We call this phenomenon *error propagation* because low-quality model updates from low-contributing clients deteriorate the high-quality models.

**Stochastic recovery strategy.** To ensure convergence to the global optimum, we propose a stochastic recovery strategy, which shares the current reference model with a client at probability $q$. The reference model is allowed to have a custom induced reference reward rate $\gamma'_{\text{ref},t}$. Specifically, the reference model is trained by aggregating local model updates $\{\boldsymbol{g}_{i,t}\}_{i \in \mathcal{S}_{\text{ref},t}}$ where $\mathcal{S}_{\text{ref},t}$ is randomly sampled from $[N]$ such that $|\mathcal{S}_{\text{ref},t}| = \lceil \gamma'_{\text{ref},t} N \rceil$. Therefore, every participating client at each FL iteration has an equal chance $q$ of receiving the current reference model. With probability $1 - q$, the client still receives from the server an update that is aggregated with the corresponding proportion of local model updates. Algorithm 1 outlines the implementations. As we present in the following theorem, this method prevents persistent error propagation and enables the convergence of the model.

**Theorem 2** (Improved bound). *Let $\eta_t = \frac{\beta}{t+\alpha}$ and $B, H_T$ be defined in Theorem 1. With a stochastic recovery rate of $q$, the performance of client model $\boldsymbol{\theta}_{i,T}$ trained over $T$ FL iterations is bounded by*

$$\mathbb{E}\left[F(\boldsymbol{\theta}_{i,T})\right] - F^* \leq \frac{L}{2} \left( \prod_{t=1}^{T} (1 - 2\mu\eta_t\tau) \right) \|\boldsymbol{\theta}_1 - \boldsymbol{\theta}^*\|^2 + \frac{L}{2} \sum_{t=1}^{T} \eta_t^2 (Q + D_t + E_t) \prod_{l=t+1}^{T} (1 - 2\mu\eta_t\tau)$$

*where*
$$D_T = \frac{H_T}{N} \sum_{m=1}^{N} \sum_{t=1}^{T} \left( \frac{T+\alpha}{t+\alpha} \right)^2 \left( \frac{1}{\gamma'_{m,t}} + \frac{1}{\gamma'_{\text{ref},t}} - 2 \right) (1-q)^{T-t+1}$$

$$E_T = H_T \sum_{t=1}^{T} \left( \frac{T+\alpha}{t+\alpha} \right)^2 \left( \frac{1}{\gamma'_{i,t}} + \frac{1}{\gamma'_{\text{ref},t}} - 2 \right) (1-q)^{T-t+1} .$$

**Proposition 2** (Asymptotic convergence). *If $\frac{1}{\gamma'_{i,t}} = o\left( \frac{t^2}{\log t} \right)$ for all $i \in [N]$ and $\frac{1}{\gamma'_{\text{ref},t}} = o\left( \frac{t^2}{\log t} \right)$,*

$$\lim_{T \to \infty} \mathbb{E}\left[F(\boldsymbol{\theta}_{i,T})\right] - F^* = 0 .$$

The proofs are in Appendix D.3 and Appendix D.4. Asymptotically, all clients are expected to achieve the optimal model regardless of their contributions. To interpret the conditions for convergence, the contributions of all clients $i \in [N]$ and the induced reference reward rate do not decrease too quickly over time. The first condition is satisfied as $1/\gamma'_{i,t} \leq N$ by definition whereas the second is user-defined. While offering the same model in the limit may seem to work against client incentives at first glance, clients who exit the collaboration early will not obtain the global optimal model. Theorem 2 shows that the impact of contribution $\gamma'_{i,t}$ is reflected on $E_T$, where $E_T$ decreases with $\gamma'_{i,t}$. Therefore, a client that contributes more receives a better model with a smaller performance gap to the global optimum. This improvement is more highlighted for limited FL iterations $T$ (see Section 6.2).

## 6 EXPERIMENTS

In this section, we show the effectiveness of IAFL and the impacts of its hyperparameters. We also show that IAFL is suited for custom contribution measure definitions and partial client participation.

**Datasets & partition strategies.** We use the datasets and heterogeneous data partition pipelines in the non-IID FL Benchmark (Li et al., 2022). We simulate 50 clients for all experiments. We perform extensive experiments on various vision datasets like MNIST (LeCun et al., 1989), FMNIST (Xiao et al., 2017), SVHN (Netzer et al., 2011), CIFAR-10/100 (Krizhevsky, 2009), Tiny-ImageNet (Deng et al., 2009) and language datasets *Stanford Sentiment Treebank* (SST) (Socher et al., 2013), Sentiment140 (Go et al., 2009). The datasets are partitioned among the clients following five partitioning strategies: (1) ***Distribution-based label distribution skew*** simulates label imbalance by sampling proportions $\boldsymbol{\pi}_k$ from a Dirichlet distribution $Dir(\boldsymbol{\beta})$ where $\boldsymbol{\beta}$ is the concentration parameter. For each label class $k$, we sample $\boldsymbol{\pi}_k$ and allocate $\pi_{k,i}$ of label-$k$ samples to client $i$; (2) ***Quantity-based label distribution skew***, denoted by $\#C = k$, creates another type of label imbalance by sampling $k$ label classes for each client and then randomly and equally distributing samples from class $k$ among eligible clients; (3) ***Noise-based feature distribution skew*** only applies to vision datasets and adds noises with distribution $\mathcal{N}(\sigma \cdot i/N)$ to the data of client $i$; (4) ***Quantity skew*** allocates $\pi_i$ proportion of total data randomly to client $i$ from a Dirichlet sample $\boldsymbol{\pi} \sim Dir(\boldsymbol{\beta})$; (5) ***Homogeneous partition***, denoted by IID, partitions the samples from each class randomly and equally among the clients.

Table 1: Comparison of $\text{IPR}_{\text{accu}}$ among IAFL and baselines using different dataset partitions. Each value reports the mean and the standard error of 10 independent evaluations and partition seedings.

| Category | Dataset | Partitioning | FedAvg Finetune | LG-FedAvg | CGSV | Rank | IAFL |
|---|---|---|---|---|---|---|---|
| **Label Distribution Skew** | MNIST | $Dir(0.5)$ | 0.85±0.01 | 0.96±0.01 | 0.95±0.01 | 0.79±0.08 | **1.00±0.00** |
| | | #C = 3 | 0.73±0.06 | **1.00±0.00** | **1.00±0.00** | 0.84±0.02 | **1.00±0.00** |
| | FMNIST | $Dir(0.5)$ | 0.65±0.03 | 0.89±0.01 | 0.95±0.01 | 0.80±0.02 | **0.99±0.01** |
| | | #C = 3 | 0.41±0.04 | 0.99±0.01 | 0.99±0.01 | 0.71±0.02 | **1.00±0.00** |
| | SVHN | $Dir(0.5)$ | 0.99±0.00 | **1.00±0.00** | 0.90±0.02 | 0.92±0.02 | **1.00±0.00** |
| | | #C = 3 | 0.35±0.05 | **0.97±0.01** | 0.56±0.09 | 0.46±0.02 | 0.82±0.02 |
| | CIFAR-10 | $Dir(0.5)$ | 0.26±0.09 | 0.45±0.09 | 0.12±0.06 | 0.33±0.10 | **0.67±0.05** |
| | | #C = 3 | 0.00±0.00 | 0.01±0.01 | 0.01±0.00 | 0.02±0.01 | **0.46±0.03** |
| | CIFAR-100 | $Dir(0.5)$ | 0.37±0.05 | 0.96±0.02 | 0.00±0.00 | 0.83±0.02 | **1.00±0.00** |
| | | #C = 30 | 0.02±0.01 | 0.14±0.03 | 0.00±0.00 | 0.40±0.05 | **0.64±0.06** |
| | SST | $Dir(0.5)$ | 0.47±0.04 | 0.65±0.02 | 0.47±0.03 | 0.75±0.03 | **0.79±0.03** |
| | | #C = 3 | **0.96±0.01** | 0.58±0.02 | 0.51±0.05 | 0.85±0.01 | 0.89±0.01 |
| **Feature Distribution Skew** | MNIST | | 0.19±0.03 | 0.70±0.04 | **0.98±0.01** | 0.21±0.04 | 0.71±0.09 |
| | FMNIST | | 0.66±0.03 | 0.16±0.05 | 0.02±0.01 | 0.68±0.03 | **1.00±0.00** |
| | SVHN | $\mathcal{N}(0.1)$ | **1.00±0.00** | **1.00±0.00** | **1.00±0.00** | 0.97±0.01 | **1.00±0.00** |
| | CIFAR-10 | | **1.00±0.00** | **1.00±0.00** | 0.16±0.07 | **1.00±0.00** | **1.00±0.00** |
| | CIFAR-100 | | **1.00±0.00** | **1.00±0.00** | 0.46±0.10 | 0.98±0.01 | **1.00±0.00** |
| **Quantity Skew** | MNIST | | 0.75±0.02 | 0.89±0.01 | 0.95±0.01 | 0.97±0.01 | **0.99±0.01** |
| | FMNIST | | 0.88±0.02 | 0.80±0.03 | 0.52±0.02 | 0.90±0.02 | **1.00±0.00** |
| | SVHN | $Dir(0.5)$ | **1.00±0.00** | **1.00±0.00** | 0.99±0.01 | 0.63±0.06 | **1.00±0.00** |
| | CIFAR-10 | | 0.91±0.01 | 0.96±0.01 | 0.65±0.02 | 0.54±0.04 | **0.96±0.00** |
| | CIFAR-100 | | 0.98±0.01 | **0.99±0.00** | 0.68±0.01 | 0.68±0.04 | 0.98±0.01 |
| | SST | | **1.00±0.00** | 0.57±0.05 | **1.00±0.00** | 0.94±0.01 | **1.00±0.00** |
| **Homogeneous Partition** | MNIST | | 0.23±0.02 | 0.58±0.04 | **0.99±0.01** | 0.25±0.06 | 0.71±0.07 |
| | FMNIST | | 0.60±0.05 | 0.37±0.03 | 0.01±0.01 | 0.68±0.04 | **1.00±0.00** |
| | SVHN | IID | **1.00±0.00** | **1.00±0.00** | **1.00±0.00** | **1.00±0.00** | **1.00±0.00** |
| | CIFAR-10 | | **1.00±0.00** | **1.00±0.00** | 0.26±0.09 | 0.99±0.00 | **1.00±0.00** |
| | CIFAR-100 | | **1.00±0.00** | **1.00±0.00** | 0.64±0.08 | **1.00±0.00** | **1.00±0.00** |
| | SST | | **1.00±0.00** | 0.51±0.07 | **1.00±0.00** | 0.96±0.01 | **1.00±0.00** |
| Number of times that performs the best | | | 10 | 11 | 7 | 3 | **24** |

**Baselines.** We compare to a FedAvg (McMahan et al., 2017) baseline with local finetuning, a personalization method LG-FedAvg (Liang et al., 2020) and two FL training-time model incentives methods, CGSV (Xu et al., 2021) and Rank (Kong et al., 2022). In particular, CGSV has demonstrated empirical superiority over other popular techniques such as q-FFL (Li et al., 2020c), CFFL (Lyu et al., 2020) and ECI (Song et al., 2019). We refer to Appendix E.1 for the implementation details.

## 6.1 INCENTIVIZATION PERFORMANCE

**Incentivized participation rate (IPR).** Cho et al. (2022) proposed using IPR to measure the incentivization performance of an FL algorithm. We let $\text{IPR}_{\text{loss}}$ be the percentage of clients receiving a model not worse than his standalone model in terms of the test loss. Likewise, We define $\text{IPR}_{\text{accu}}$ but in terms of the test accuracy. Here, a standalone model refers to the model trained only with the client's local dataset. IPR therefore indicates IR (Definition 1). Table 1 summarizes the $\text{IPR}_{\text{accu}}$ for different methods, datasets and partitions. Notably, our IAFL achieves the highest $\text{IPR}_{\text{accu}}$ 24 times out of the 29 experimental settings, out of which 17 of them have a 100% $\text{IPR}_{\text{accu}}$. Although FedAvg Finetune and LF-FedAvg achieve comparable performance as IAFL, they are significantly worse than IAFL under the heterogeneous learning setting with label distribution skew, which may be the most challenging regime and the regime of interest for FL. Therefore, we have demonstrated IAFL's ability to facilitate effective collaboration to achieve better learning performance for federated clients. Additionally, we attribute the imperfect IR (Remark 1) to the violation of non-decreasing and concave assumptions for the rewarding function $\nu$ in practice, which is defined as the test accuracy in this experiment. Alternatively choosing $\nu$ as the test loss, IAFL achieves perfect IR for almost all settings. We show the results for $\text{IPR}_{\text{loss}}$ in Table 4 of Appendix E.2. Additional results on complex large-scale datasets, Tiny-ImageNet and Sent140, are in Appendix E.9.

**Correlation to contribution.** Another way to quantify the incentivization performance is by measuring the Pearson correlation coefficient $\rho$ between the client model accuracies achieved by the algorithm after $T$ FL iterations and their standalone accuracies (Xu et al., 2021). We use standalone accuracies as a surrogate for the client contributions in FL, hence a good incentivizing method should produce client models with accuracies highly correlated to the respective client contributions. Table 2 presents the effectiveness of various methods under different types of heterogeneity. Most methods, including LG-FedAvg, CGSV and Rank, could only perform well when the quality of client datasets

Table 2: The incentivization performance under different dataset partitions, measured using the Pearson correlation coefficient $\rho$ between the final client model accuracies and standalone accuracies. Each value reports the mean and the standard error over 10 independent evaluations.

| Category | Dataset | Partitioning | FedAvg Finetune | LG-FedAvg | CGSV | Rank | IAFL |
|---|---|---|---|---|---|---|---|
| **Label Distribution Skew** | MNIST | $Dir(0.5)$ | 0.74±0.03 | 0.66±0.06 | -0.80±0.02 | 0.71±0.08 | **0.81±0.02** |
| | | $\#C=3$ | 0.27±0.06 | 0.12±0.03 | -0.01±0.11 | 0.56±0.03 | **0.59±0.02** |
| | FMNIST | $Dir(0.5)$ | 0.85±0.01 | 0.82±0.02 | 0.23±0.14 | **0.92±0.01** | 0.84±0.02 |
| | | $\#C=3$ | -0.05±0.09 | 0.35±0.05 | 0.28±0.04 | 0.65±0.01 | **0.86±0.02** |
| | SVHN | $Dir(0.5)$ | **0.87±0.01** | 0.75±0.01 | 0.57±0.03 | 0.85±0.01 | 0.81±0.01 |
| | | $\#C=3$ | **1.00±0.00** | 0.81±0.02 | 0.49±0.05 | 0.79±0.02 | 0.84±0.01 |
| | CIFAR-10 | $Dir(0.5)$ | 0.76±0.04 | **0.84±0.03** | 0.46±0.04 | 0.79±0.03 | 0.79±0.02 |
| | | $\#C=3$ | 0.83±0.01 | **0.88±0.01** | 0.31±0.05 | 0.56±0.06 | 0.63±0.02 |
| | CIFAR-100 | $Dir(0.5)$ | 0.58±0.04 | 0.43±0.04 | 0.20±0.07 | 0.60±0.03 | **0.89±0.01** |
| | | $\#C=30$ | 0.70±0.02 | 0.55±0.03 | 0.01±0.07 | 0.53±0.03 | **0.87±0.02** |
| | SST | $Dir(0.5)$ | 0.83±0.01 | **0.97±0.00** | 0.73±0.04 | 0.89±0.02 | 0.72±0.02 |
| | | $\#C=3$ | 0.86±0.01 | **0.93±0.01** | 0.53±0.09 | 0.84±0.01 | 0.66±0.02 |
| **Feature Distribution Skew** | MNIST | | 0.01±0.03 | 0.09±0.04 | -0.02±0.04 | 0.03±0.04 | **0.39±0.09** |
| | FMNIST | | 0.04±0.03 | 0.06±0.04 | 0.01±0.05 | 0.03±0.04 | **0.39±0.05** |
| | SVHN | $\mathcal{N}(0.1)$ | 0.00±0.05 | 0.14±0.03 | -0.01±0.07 | 0.13±0.05 | **0.41±0.07** |
| | CIFAR-10 | | 0.08±0.05 | 0.08±0.04 | -0.02±0.04 | 0.20±0.04 | **0.57±0.03** |
| | CIFAR-100 | | 0.12±0.04 | 0.16±0.04 | -0.05±0.05 | 0.20±0.03 | **0.72±0.02** |
| **Quantity Skew** | MNIST | | -0.56±0.03 | 0.82±0.04 | 0.37±0.16 | **0.88±0.03** | 0.77±0.03 |
| | FMNIST | | -0.37±0.03 | 0.90±0.01 | 0.86±0.04 | **0.94±0.01** | 0.78±0.02 |
| | SVHN | $Dir(0.5)$ | -0.20±0.03 | 0.83±0.02 | 0.75±0.04 | **0.94±0.01** | 0.77±0.03 |
| | CIFAR-10 | | 0.25±0.06 | **0.95±0.00** | 0.73±0.04 | 0.95±0.01 | 0.80±0.01 |
| | CIFAR-100 | | -0.24±0.04 | 0.80±0.01 | 0.66±0.04 | **0.93±0.01** | 0.83±0.01 |
| | SST | | 0.20±0.06 | **0.81±0.01** | 0.49±0.05 | 0.70±0.02 | 0.47±0.06 |
| **Homogeneous Partition** | MNIST | | -0.03±0.05 | 0.02±0.05 | 0.08±0.04 | 0.01±0.04 | **0.39±0.12** |
| | FMNIST | | 0.04±0.05 | -0.06±0.05 | 0.01±0.03 | 0.12±0.04 | **0.39±0.05** |
| | SVHN | IID | 0.04±0.03 | 0.20±0.07 | -0.04±0.04 | 0.11±0.08 | **0.40±0.06** |
| | CIFAR-10 | | 0.00±0.05 | 0.08±0.04 | -0.03±0.05 | 0.09±0.03 | **0.59±0.05** |
| | CIFAR-100 | | -0.02±0.03 | 0.08±0.04 | 0.05±0.07 | 0.18±0.04 | **0.71±0.02** |
| | SST | | 0.27±0.04 | **0.69±0.02** | 0.22±0.04 | 0.56±0.02 | 0.44±0.02 |
| Number of times that performs the best | | | 2 | 7 | 0 | 5 | **15** |

exhibits distinct differences under severe label and quantity skew. They cannot handle the more challenging scenarios under feature skew or homogeneous partition, where client data only present minor differences in standalone accuracies. In particular, Rank further assumes access to a good global validation set to rank local models during training, which could be infeasible in practice. FedAvg Finetune fails the quantity skew data partition, showing even negative correlations. In contrast, IAFL shows superior incentivization performance across extensive settings. It can achieve relatively high $\rho$ for all settings and perform the best for 16 out of 29 experimental settings. Therefore, IAFL has demonstrated its ability to distribute model rewards that are commensurate with client contributions.

**Predictive performance.** We compare the average and highest client model test accuracies achieved by IAFL and the baseline methods. Presented in Table 5 of Appendix E.3 and Table 10 of Appendix E.9, our IAFL outperforms other baselines in nearly all the settings. The high average client model performance illustrates the effectiveness of collaboration following IAFL. Additionally, IAFL is able to maintain the model performance of high-contributing clients by updating their local models using data from a larger proportion of clients, which is important in a collaborative training setting. Additional results showing the average increase in test accuracies are in Table 6 of Appendix E.4.

## 6.2 EFFECT OF HYPERPARAMETERS ON EMPIRICAL CONVERGENCE

**Sharing coefficient $\kappa$.** As introduced in (1), $\kappa \in [0, 1]$ determines the extent of sharing in IAFL and reflects an essential practical trade-off between model performance and client incentivization (more in Appendix E.7). A larger $\kappa$ results in more equitable and better client models, but this can hurt the incentives of high-contributing clients as they receive models similar to those received by low-contributing clients. This is reflected in Figure 1 where we plot the model performance of 6 clients with different $\lambda'_i$ (see Section 5). A larger $\kappa$ better utilizes local model updates and achieves higher accuracies. However, it results in models with more similar model performance and thus worse client incentivization measured by $\rho$. For convergence, Figure 1 confirms the results in Theorem 1 that less contributing clients converge to worse models, further away from the global optimum.

**Stochastic recovery rate $q$.** Stochastic recovery is introduced in IAFL to ensure asymptotic convergence to the global optimum for all client models. In a stochastic manner, each client has an equal probability $q$ to be recovered with a reference model in each FL iteration. In practice, we recommend

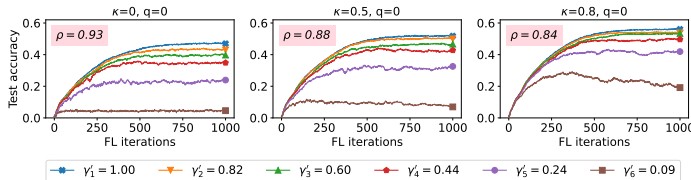

Figure 1: Effects of sharing coefficient $\kappa$ with a fixed stochastic recovery rate. We use CIFAR-100 with quantity skew and $N = 50$.

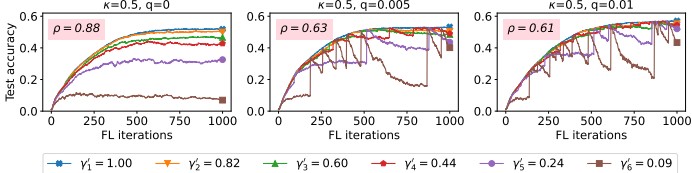

Figure 2: Effects of stochastic recovery rate $q$ with a fixed sharing coefficient. We use CIFAR-100 with quantity skew and $N = 50$.

Table 3: Performance comparison to baselines when $p_i$ corresponds to the participation rate.

| Method | $\rho$ | IPR$_{\text{accu}}$ | Acc. (highest) |
|---|---|---|---|
| FedAvg Finetune | 0.06 | 0.28 | 0.12 (0.16) |
| LG-FedAvg | -0.25 | 0.60 | 0.13 (0.15) |
| CGSV | 0.90 | 0.00 | 0.06 (0.07) |
| Rank | -0.74 | 0.94 | 0.24 (0.31) |
| IAFL | **0.94** | **1.00** | **0.44 (0.53)** |

Figure 3: Visualization of correlation $\rho$ when $p_i$ corresponds to the participation rate.

the reference model to be updated with the best reward (largest $\lambda'_{i,t}$) in each FL iteration, whereas another choice of the "median" reference model is discussed in Appendix E.5. As shown in Figure 2, stochastic recovery presents a trade-off between incentivization performance measured by $\rho$ and the test accuracies of client models. A small $q$ is sufficient as $q = 0.01$ results in higher test accuracies while maintaining a relatively high incentivization performance of $\rho = 0.61$ in finite FL iterations. The empirical convergence observed in Figure 2 corroborates the theoretical guarantee in Theorem 2.

**Free riders behavior.** With differentiated training-time model rewards, IAFL can offer worsened or delayed rewards to free riders. In Figures 1 and 2, we identify client 6 with $\lambda'_6 = 0.09$ as the free rider as it contributes very little compared to other clients. With $q = 0$, the free rider only achieves $4.6\%$ accuracy when $\kappa = 0$ and $19.2\%$ accuracy when the $\kappa = 0.8$. After the stochastic recovery is incorporated, we observe the free rider receives a good recovered reward at a delayed time stochastically and continues to worsen due to the poor model updates received in each FL iteration.

### 6.3 ALTERNATIVE CONTRIBUTION MEASURES

IAFL is flexible to plug-and-play with various contribution evaluation techniques out of the box (i.e., no modification is needed). To demonstrate the effectiveness with partial client participation, we let the client participation rate be the measure. Specifically, we define $p_i = 0.5 \times (1 + i/N)$ as illustrated in Figure 3. There is a high correlation of $0.94$ between the test accuracies of client models and client contributions measured by the participation rate $p_i$. In Table 3, IAFL also achieves the best average (highest) client model accuracy of $44\%$ ($53\%$) and IPR$_{\text{accu}}$ across all methods. In Appendix E.8, we further show IAFL's superior effectiveness when using the established CGSV contribution measure.

## 7 CONCLUSION & DISCUSSION

In this paper, we have introduced the IAFL, a novel algorithm that distributes training-time model rewards to incentive client contributions for FL. In particular, we propose a proportional aggregation scheme for local training-time rewards that achieves the global incentivization objective. By analyzing the theoretical guarantees of the method, we observe the error propagation phenomenon that creates tension between client incentives and achieving optimality on client models. We then mitigate the problem with a stochastic reference-model recovery strategy, which enables us to distribute globally optimal models in the limit while preserving client incentives. Our method is highly versatile and can be applied to a wider range of domains (e.g., financial, biomedical, etc.) as it allows novel definitions of the contribution measure according to the specific resources that are valuable in the domain.

Our work also opens up several directions for future investigation. It is important to study the roles of the server and clients in determining the contribution measure and rewarding strategy. Additionally, there can be tensions and mismatches between the contribution measure and the model rewards because clients can continuously contribute, while model performance cannot grow infinitely and suffers from diminishing returns. We further elaborate on the broader societal impacts in Appendix A.

## REPRODUCIBILITY STATEMENT

We have carefully discussed all the assumptions for the theoretical results and included the complete proofs for all theorems, lemmas and propositions in Appendix B. To further improve the reproducibility of our work, we elaborate in Appendix E.1 the details for dataset preprocessing, models architectures, hyperparameters for training using IAFL as well as training using all the baseline methods that we compare to. The code has been submitted as supplementary material.

## ACKNOWLEDGEMENT

This research/project is supported by the National Research Foundation Singapore and DSO National Laboratories under the AI Singapore Programme (AISG Award No: AISG2-RP-2020-018). The authors thank Fusheng Liu for many interesting discussions.

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

## A  BROADER IMPACTS AND ETHICS

Designed to promote client contribution among potentially competitive clients in FL, our incentive-aware federated learning algorithm with training-time model reward offers versatile applications in various fields such as finance, healthcare, the Internet of things, etc. While our proposed method is capable of shaping the pipeline of collaboration between the server and clients in distributed learning, we also see potential societal impacts associated with the development of such incentive methods.

Firstly, the use of highly customizable contribution measures presents both advantages and challenges. While it is technically feasible to employ any contribution measure, ethical considerations arise when certain choices of the contribution measure do not align well with the training-time model reward. For instance, the rewards based on model performance (e.g., using the test performance as a utility measure), cannot infinitely improve and should exhibit diminishing returns with respect to the contribution of learning resources. This raises questions about the ethical implications of incentivizing client contributions in these later stages of model training, especially when the marginal utility is minimal. Is it ethical to continue incentivizing client contributions at later FL iterations (e.g., near convergence), knowing that the marginal utility of contribution is small? Is it reasonable to reward a client less at a later FL iteration, even though the client contributes more in this FL iteration as compared to some earlier iterations? These concerns need to be addressed with specific application scenarios, taking into account the characteristics of the server and clients. In order to mitigate these ethical concerns, it may be necessary to enforce regulations on the protocol to ensure the adoption of an accumulative contribution measure.

Secondly, the ethical considerations regarding the distribution of rewards require closer examination. In this paper, we adopt the perspective of collaborative fairness (Xu & Lyu, 2021), where our incentive-aware algorithm encourages client contributions by providing rewards commensurate with their level of contribution. However, some may argue that this approach violates equitable fairness (Li et al., 2020c) among clients. Interestingly, our stochastic reference-model recovery with a tunable probability $q$ ensures equitable fairness in the long run (i.e., as the number of FL iterations approaches infinity). By adjusting the value of $q$, a balance can be struck between collaborative fairness and equitable fairness. Smaller $q$ values prioritize collaborative fairness, while larger $q$ values emphasize equitable fairness. Therefore, guidelines should be established to assist the server and clients in collaboratively determining the algorithm's hyperparameters, thereby minimizing potential ethical concerns.

Thirdly, reward distribution within the broader spectrum of collaborative learning may require careful ethical considerations. In this paper, we give out training-time model rewards due to the difficulties of timeliness and feasibility (discussed in Section 1) associated with post-training monetary rewards. We believe future investigations could potentially open up new opportunities if we can incorporate both monetary and non-monetary (e.g., model) rewards when considering client incentives in FL training. However, the challenge lies in determining the denomination between monetary incentives and model incentives both technically and ethically. Nguyen et al. (2022) offer some insights on dealing with the following scenario: How can one rectify a situation where a client's contribution does not correspond to a good model reward, but the client desires to obtain a better model by offering additional monetary payments? Addressing these interesting questions could bring us a more holistic view of incentivization and collaboration in learning. At the same time, apart from the technical perspective, more in-depth ethical studies may be necessary to regulate such denominations between monetary and non-monetary rewards for client incentivization in practice.

## B  PROOFS

### B.1  PROOF OF PROPOSITION 1

**Proposition 1.** *If $\tilde{h}$ satisfies $\frac{d\tilde{h}(p_i, \boldsymbol{p}_{-i})}{dp_i} > 0$ and $\frac{\partial \nu(p_i, \tilde{h}(p_i, \boldsymbol{p}_{-i}))}{\partial \tilde{h}(p_i, \boldsymbol{p}_{-i})} > 0$, then a mechanism $[\mathcal{M}_\nu(\boldsymbol{p})]_i = \nu(p_i, \tilde{h}(p_i, \boldsymbol{p}_{-i}))$ incentivizes a client to contribute more than that in the standard FL mechanism.*

*Proof.* The utility of client $i$ with $p_i$ contribution can be written as

$$u_i = [\mathcal{M}_\nu(\boldsymbol{p})]_i - c_i p_i$$
$$= \nu(p_i, \tilde{h}(p_i, \boldsymbol{p}_{-i})) - c_i p_i .$$

Thus, taking the derivative with respect to $p_i$, we have

$$u_i' = \frac{\partial \nu}{\partial p_i} \frac{dp_i}{dp_i} + \frac{\partial \nu}{\partial \tilde{h}(p_i, \boldsymbol{p}_{-i})} \frac{d\tilde{h}(p_i, \boldsymbol{p}_{-i})}{dp_i} - c_i$$

$$= \frac{\partial \nu}{\partial p_i} + \frac{\partial \nu}{\partial \tilde{h}(p_i, \boldsymbol{p}_{-i})} \frac{d\tilde{h}(p_i, \boldsymbol{p}_{-i})}{dp_i} - c_i . \qquad (2)$$

In the standard FL mechanism, we have $\tilde{h}(p_i, \boldsymbol{p}_{-i}) = h(\boldsymbol{p}_{-i})$ where $h(\boldsymbol{p}_{-i})$ is independent of $p_i$. This implies $\frac{d\tilde{h}(p_i, \boldsymbol{p}_{-i})}{dp_i} = 0$. The derivative of the utility is

$$u_i'^{(FL)} = \frac{\partial \nu}{\partial p_i} - c_i .$$

If the following conditions are satified: ① $\frac{\partial \nu}{\partial \tilde{h}(p_i, \boldsymbol{p}_{-i})} > 0$ and ② $\frac{d\tilde{h}(p_i, \boldsymbol{p}_{-i})}{dp_i} > 0$, according to (2) we will have $u_i' > u_i'^{(FL)}$ for any $p_i$.

To maximize the utility, a rational client $i$ will contribute $p_i^*$ such that $u_i' = 0$. Since $u_i$ is in general a concave function and $u_i' > u_i'^{(FL)}$ for any $p_i$ with ① $\frac{\partial \nu}{\partial \tilde{h}(p_i, \boldsymbol{p}_{-i})} > 0$ and ② $\frac{d\tilde{h}(p_i, \boldsymbol{p}_{-i})}{dp_i} > 0$, we have $p_i^* > p_i^{*(FL)}$. That is, the mechanism with such a $\tilde{h}$ incentivizes a client to contribute more than that in the standard FL mechanism.

$\square$

## B.2 PROOF OF PROPOSITION 3

**Proposition 3.** *There exists an undesirable equilibrium in the relative reward mechanism* $[\mathcal{M}_\nu(\boldsymbol{p})]_i = \nu\left(p_i, \frac{p_i}{\max_j p_j} h(\boldsymbol{p}_{-i})\right)$. *Specifically, the equilibrium elicits contributions from all client that sums to what a single client would contribute without the federation.*

*Proof.* This is a constructive proof for existence.

We consider $N$ identical clients each with a common cost $c_i = c, \forall i \in \{1, \ldots, N\}$. Let the rewarding function $\nu$ be a composition function such that $\nu(p_i, P_{-i}) = g(f(p_i, P_{-i}))$ where $g$ is a continuous non-decreasing concave function and $f(p_i, P_{-i}) = p_i + P_{-i}$. Furthermore, we define the aggregate function as $h(\boldsymbol{p}_{-i}) = \sum_{j:j \in [N] \wedge j \neq i} p_j$.

Define an optimal contribution $p^*$ for an individual client, such that $\nu'(p^*, P_{-i}) = g'(p^*) = c$. This is because $P_{-i} = h(\boldsymbol{p}_{-i}) = 0$ when there is only an individual client.

Now, if every client contributes $p^*/N$ (i.e., $\boldsymbol{p} = [p^*/N, \ldots, p^*/N]$), we have $\frac{p_i}{\max_j p_j} = 1$ for all $i$ and thus

$$[\mathcal{M}_\nu(\boldsymbol{p})]_i = \nu(p^*/N, h(\underbrace{[p^*/N, \ldots, p^*/N]}_{N-1 \text{ terms}}))$$

$$= g(f(p^*/N, \sum_{j:j \in [N] \wedge j \neq i} p_j))$$

$$= g\left(\frac{p^*}{N} + (N-1)\frac{p^*}{N}\right)$$

$$= g(p^*) .$$

In this case, clients reach a bad Nash equilibrium as the total contribution from all the federated clients is the same as a single client's contribution in the non-federated environment. Here, in the

Nash equilibrium, each client is assumed to know the equilibrium strategies of the other participating clients, and no client can increase one's own utility by deviating to other strategies. In other words, the mechanism does not incentivize clients to contribute more. □

## C  CONNECTION BETWEEN THE MECHANISM AND THE IAFL ALGORITHM

We draw the connections between the IAFL algorithm in Algorithm 1 and the mechanism in (1) through a simplifying scenario. Consider $N$ homogeneous clients producing updates $\{\boldsymbol{g}_{i,t}\}_{i=1}^N$ that follow a population with the same mean $\boldsymbol{\mu}$. The update $\Delta\boldsymbol{\theta}_{i,t}$ follows the sampling distribution of the mean with $Var(\Delta\boldsymbol{\theta}_{i,t}) = \sigma_t^2/|\mathcal{S}_{i,t}|$ where $\sigma_t$ is the population variance. Note that step 4 of Procedure 2 aggregates the contribution of client $i$, $\Delta\boldsymbol{\theta}_{i,t}$, with the sampled contributions of other clients. Taking the reciprocal of the variance as the quality of the training-time model update reward, then

$$\frac{1}{Var(\Delta\boldsymbol{\theta}_{i,t})} = \frac{1}{\sigma_t^2} + \left(\min\left\{\frac{p_{i,t}}{p_{\text{ceil}}}, 1\right\}\right)^{1-\kappa}\frac{N-1}{\sigma_t^2}\,,$$

which matches the mechanism in (1).

## D  PROOF OF THE CONVERGENCE

Denote the optimal solution of $F(\boldsymbol{\theta})$ by $\boldsymbol{\theta}^*$ such that $F^* = F(\boldsymbol{\theta}^*)$. Similarly, we denote $F_i^* = F_i(\boldsymbol{\theta}_i^*)$ where $\boldsymbol{\theta}_i^*$ is the optimal solution of $F_i(\boldsymbol{\theta})$. We define the data bias as $\Gamma \triangleq F^* - \frac{1}{N}\sum_{i=1}^N F_m^*$. We set the learning rate for client $i$ at FL iteration $t$ to common rate $\eta_t = \eta_{i,t}, \forall i$. Recall that the $j$-th step of stochastic gradient decent at client $i$ is $\boldsymbol{\theta}_{i,t}^{j+1} = \boldsymbol{\theta}_{i,t}^j - \eta_t\nabla F_i(\boldsymbol{\theta}_{i,t}^j, \xi_{i,t}^j)$ where $\xi_{i,t}^j$ is the local batch randomly chosen for the SGD step. Note that $\boldsymbol{\theta}_{i,t}^1 = \boldsymbol{\theta}_{i,t}$ is the model at client $i$ at the beginning of a FL iteration $t$. At the end of each FL iteration, a client $i$ updates its model to

$$\boldsymbol{\theta}_{i,t+1} = \boldsymbol{\theta}_{i,t} - \eta_t\frac{1}{|\mathcal{S}_{i,t}|}\sum_{m\in\mathcal{S}_{i,t}}\sum_{j=1}^\tau\nabla F_m(\boldsymbol{\theta}_{m,t}^j, \xi_{m,t}^j)$$

where $\mathcal{S}_{i,t}$ is defined in Algorithm 1 as the set of indices that a client $i$ can aggregate from in a given FL iteration $t$. Without loss of generality, we assume that $\gamma_{i,t}'N$ is an integer. In practice, the ceiling $\lceil\gamma_{i,t}'N\rceil$ can be used. Note that $|\mathcal{S}_{i,t}| = \gamma_{i,t}'N$.

For simplicity, we denote the update for client $i$ in iteration $t$ as

$$R_{i,t} = -\eta_t\frac{1}{|\mathcal{S}_{i,t}|}\sum_{m\in\mathcal{S}_{i,t}}\sum_{j=1}^\tau\nabla F_m(\boldsymbol{\theta}_{m,t}^j, \xi_{m,t}^j)\,.$$

Since $R_{i,t}$ essentially computes the i.i.d. sample mean from the population $\{\boldsymbol{g}_{m,t}\}_{m=1}^N = \{-\eta_t\nabla F_m(\boldsymbol{\theta}_{m,t}^j, \xi_{m,t}^j)\}_{m=1}^N$ for each FL iteration $t$, we have $\mathbb{E}[R_{i,t}] = \mathbb{E}[R_{j,t}]$ for all $i, j \in \{1, \ldots, N\}$.

For the analysis in this section, we make the following assumptions on the functions $F_1, \ldots, F_N$:

**Assumption 1** ($L$-smoothness). *The loss function $F_m$ is $L$-smooth, then for all $\boldsymbol{u}, \boldsymbol{v} \in \mathbb{R}^d$,*

$$2(F_m(\boldsymbol{u}) - F_m(\boldsymbol{v})) \leq 2\langle\boldsymbol{u} - \boldsymbol{v}, \nabla F_m(\boldsymbol{v})\rangle + L\|\boldsymbol{u} - \boldsymbol{v}\|^2\,, \quad \forall m \in [N]\,.$$

**Assumption 2** ($\mu$-strong convexity). *The loss functions $F_1, \ldots, F_N$ for all $N$ clients are $\mu$-strongly convex, then for all $\boldsymbol{u}, \boldsymbol{v} \in \mathbb{R}^d$,*

$$2(F_m(\boldsymbol{u}) - F_m(\boldsymbol{v})) \geq 2\langle\boldsymbol{u} - \boldsymbol{v}, \nabla F_m(\boldsymbol{v})\rangle + \mu\|\boldsymbol{u} - \boldsymbol{v}\|^2\,, \quad \forall m \in [N]\,.$$

**Assumption 3** (Expected stochastic gradient variance). *The variance of stochastic gradients in a client is bounded,*

$$\mathbb{E}\left[\left\|\nabla F_m(\boldsymbol{\theta}_{m,t}^j, \xi_{m,t}^j) - \nabla F_m(\boldsymbol{\theta}_{m,t}^j)\right\|^2\right] \leq \tilde{\sigma}_m^2, \quad \forall m \in [N], \forall j \in [\tau], \forall t \in [T]\,.$$

*We further define $\tilde{\Sigma}^2 = \frac{1}{N}\sum_{i=1}^N\tilde{\sigma}_m^2$ as the average expected stochastic gradient variance among clients.*

**Assumption 4** (Expected squared $l_2$-norm). *The expected squared $l_2$-norm of the stochastic gradients is uniformly bounded,*

$$\mathbb{E}\left[\left\|\nabla F_m(\boldsymbol{\theta}_{m,t}^j, \xi_{m,t}^j)\right\|^2\right] \leq G^2, \quad \forall m \in [N], \forall j \in [\tau], \forall t \in [T] .$$

### D.1 PROOF OF THEOREM 1

**Theorem 1** (Performance bound). *With full clients participation and a decreasing learning rate $\eta_t = \frac{1}{\mu\tau(t+\alpha)}$ where $\alpha \geq \frac{4L(\tau+1)}{\mu\tau}$, define $B = 2L\tau(2\tau+3)\Gamma + 2\tau^3 G^2 + \tau^2\tilde{\Sigma}^2 + (\alpha+1)\mu^2\tau^2\mathbb{E}[\|\boldsymbol{\theta}_1 - \boldsymbol{\theta}^*\|^2]$ and $H_T = (2 - 2\mu\eta_T)\tau^3 G^2$, then the performance of the client model $\boldsymbol{\theta}_{i,T}$ trained over $T$ FL iterations using IAFL is bounded by*

$$\mathbb{E}\left[F(\boldsymbol{\theta}_{i,T})\right] - F^* \leq \frac{L}{2\mu^2\tau^2} \cdot \frac{B + C_T}{\alpha + T}$$

*where*

$$C_T = \frac{H_T}{N} \sum_{m=1}^{N} \sum_{t=1}^{T} \left(\frac{T+\alpha}{t+\alpha}\right)^2 \left(\frac{1}{\gamma'_{m,t}} + \frac{1}{\gamma'_{i,t}} - 2\right) .$$

*Proof.* Define a decreasing learning rate $\eta_t = \frac{\beta}{t+\alpha}$ for some $\beta > \frac{1}{2\mu\tau}$ and $\alpha \geq \frac{4L(\tau+1)}{\mu\tau}$.

From Lemma D.1, we have the following for any client $i$ after we train for $T$ FL iterations,

$$\mathbb{E}\left[\|\boldsymbol{\theta}_{i,T+1} - \boldsymbol{\theta}^*\|^2\right]$$

$$\leq (1 - 2\mu\eta_T\tau)\mathbb{E}\left[\|\boldsymbol{\theta}_{i,T} - \boldsymbol{\theta}^*\|^2\right] + 2L\tau(2\tau+3)\eta_T^2\Gamma \tag{3}$$

$$+ (2 - 2\mu\eta_T)\frac{1}{N}\sum_{m=1}^{N}\sum_{j=1}^{\tau}\mathbb{E}\left[\left\|\boldsymbol{\theta}_{m,T}^j - \boldsymbol{\theta}_{i,T}\right\|^2\right] + \eta_T^2\tau^2\tilde{\Sigma}^2 .$$

From Lemma D.2, we have the following for any pair of clients $w$ and $v$ after we train for $T$ FL iterations,

$$\mathbb{E}\left[\left\|\boldsymbol{\theta}_{w,T}^j - \boldsymbol{\theta}_{v,T}^1\right\|^2\right] \leq \tau^2 G^2 \sum_{t=1}^{T} \eta_t^2 \left(\frac{1}{\gamma'_{w,t}} + \frac{1}{\gamma'_{v,t}} - 2\right) + \eta_T^2(j-1)^2 G^2 . \tag{4}$$

Then,

$$\mathbb{E}\left[\|\boldsymbol{\theta}_{i,T+1} - \boldsymbol{\theta}^*\|^2\right]$$

$$\leq (1 - 2\mu\eta_T\tau)\mathbb{E}\left[\|\boldsymbol{\theta}_{i,T} - \boldsymbol{\theta}^*\|^2\right] + 2L\tau(2\tau+3)\eta_T^2\Gamma + \eta_T^2\tau^2\tilde{\Sigma}^2$$

$$+ (2 - 2\mu\eta_T)\frac{1}{N}\sum_{m=1}^{N}\sum_{j=1}^{\tau}\left(\tau^2 G^2 \sum_{t=1}^{T} \eta_t^2 \left(\frac{1}{\gamma'_{m,t}} + \frac{1}{\gamma'_{i,t}} - 2\right) + \eta_T^2(j-1)^2 G^2\right)$$

$$= (1 - 2\mu\eta_T\tau)\mathbb{E}\left[\|\boldsymbol{\theta}_{i,T} - \boldsymbol{\theta}^*\|^2\right]$$

$$+ \eta_T^2\left(2L\tau(2\tau+3)\Gamma + (2 - 2\mu\eta_T)G^2\sum_{j=1}^{\tau}(j-1)^2 + \tau^2\tilde{\Sigma}^2\right) \tag{5}$$

$$+ (2 - 2\mu\eta_T)\frac{1}{N}\sum_{m=1}^{N}\sum_{j=1}^{\tau}\left(\tau^2 G^2 \sum_{t=1}^{T} \eta_t^2 \left(\frac{1}{\gamma'_{m,t}} + \frac{1}{\gamma'_{i,t}} - 2\right)\right)$$

$$\leq (1 - 2\mu\eta_T\tau)\mathbb{E}\left[\|\boldsymbol{\theta}_{i,T} - \boldsymbol{\theta}^*\|^2\right] + \eta_T^2\left(2L\tau(2\tau+3)\Gamma + 2\tau^3 G^2 + \tau^2\tilde{\Sigma}^2\right)$$

$$+ \eta_T^2(2 - 2\mu\eta_T)\tau^3 G^2 \frac{1}{N}\sum_{m=1}^{N}\sum_{t=1}^{T}\left(\frac{T+\alpha}{t+\alpha}\right)^2\left(\frac{1}{\gamma'_{m,t}} + \frac{1}{\gamma'_{i,t}} - 2\right)$$

$$= (1 - 2\mu\eta_T\tau)\mathbb{E}\left[\|\boldsymbol{\theta}_{i,T} - \boldsymbol{\theta}^*\|^2\right] + \eta_T^2(Q + C_T)$$

where for simplicity we define shorthands $Q = 2L\tau(2\tau + 3)\Gamma + 2\tau^3 G^2 + \tau^2 \tilde{\Sigma}^2$, $H_T = (2 - 2\mu\eta_T)\tau^3 G^2$ and $C_T = \frac{H_T}{N} \sum_{m=1}^{N} \sum_{t=1}^{T} \left(\frac{T+\alpha}{t+\alpha}\right)^2 \left(\frac{1}{\gamma'_{m,t}} + \frac{1}{\gamma'_{i,t}} - 2\right)$. Note that $Q$ is a constant whereas $C_T$ is dependent on $T$.

Let $\Delta_t = \mathbb{E}\left[\|\boldsymbol{\theta}_{i,t} - \boldsymbol{\theta}^*\|^2\right]$. We now aim to prove $\Delta_t \leq \frac{v_t}{\alpha+t}$ by induction where $v_t = \max\left\{\frac{\beta^2(Q+C_t)}{2\beta\mu\tau-1}, (\alpha+1)\Delta_1\right\}$. When $t = 1$, we have $\Delta_1 \leq \frac{v_1}{\alpha+1}$ by definition. Now assume $\Delta_t \leq \frac{v_t}{\alpha+t}$ for some $t$, then from (5) we have

$$
\begin{aligned}
\Delta_{t+1} &\leq \left(1 - \frac{2\beta\mu\tau}{t+\alpha}\right)\frac{v_t}{t+\alpha} + \frac{\beta^2(Q+C_t)}{(t+\alpha)^2} \\
&\leq \left(1 - \frac{2\beta\mu\tau}{t+\alpha}\right)\frac{v_t}{t+\alpha} + \frac{2\beta\mu\tau - 1}{(t+\alpha)^2}v_t \\
&= \frac{v_t}{t+\alpha} - \frac{v_t}{(t+\alpha)^2} \\
&\leq \frac{v_t}{t+\alpha} - \frac{v_t}{(t+\alpha)(t+\alpha+1)} \\
&= \frac{v_t}{t+\alpha+1} \\
&\overset{(a)}{\leq} \frac{v_{t+1}}{t+\alpha+1}
\end{aligned}
\tag{6}
$$

where (a) follows from $C_t \leq C_{t+1}$ and $H_t \leq H_{t+1}$.

From the $L$-smoothness of $F$,

$$
\mathbb{E}\left[F(\boldsymbol{\theta}_{i,T})\right] - F^* \leq \frac{L}{2}\Delta_T \leq \frac{L}{2} \cdot \frac{v_T}{\alpha+T} \, .
\tag{7}
$$

Specifically, we let $\beta = \frac{1}{\mu\tau}$ and train for $t = T$ iterations, then

$$
\begin{aligned}
v_T &= \max\left\{\frac{\beta^2(Q+C_T)}{2\beta\mu\tau-1}, (\alpha+1)\Delta_1\right\} \\
&\leq \frac{\beta^2(Q+C_T)}{2\beta\mu\tau-1} + (\alpha+1)\Delta_1 \\
&= \frac{Q+C_T}{\mu^2\tau^2} + (\alpha+1)\Delta_1 \, .
\end{aligned}
\tag{8}
$$

Finally, substituting (8) into (7), we get

$$
\mathbb{E}\left[F(\boldsymbol{\theta}_{i,T})\right] - F^* \leq \frac{L}{2(\alpha+T)}\left(\frac{Q+C_T}{\mu^2\tau^2} + (\alpha+1)\Delta_1\right) \, .
\tag{9}
$$

Define $B = Q + (\alpha+1)\mu^2\tau^2\mathbb{E}\left[\|\boldsymbol{\theta}_1 - \boldsymbol{\theta}^*\|^2\right]$, we simplify the inequality to

$$
\begin{aligned}
\mathbb{E}\left[F(\boldsymbol{\theta}_{i,T})\right] - F^* &\leq \frac{L}{2\mu^2\tau^2(\alpha+T)}\left(Q + C_T + (\alpha+1)\mu^2\tau^2\Delta_1\right) \\
&= \frac{L}{2\mu^2\tau^2} \cdot \frac{B+C_T}{\alpha+T} \, .
\end{aligned}
\tag{10}
$$

$\square$

### D.2 DEFERRED PROOFS OF LEMMAS

**Lemma D.1.** *Given $\eta_t \le \frac{1}{4L(\tau+1)}$ and Assumptions 1-3,*

$$
\mathbb{E}\left[\|\boldsymbol{\theta}_{i,t+1} - \boldsymbol{\theta}^*\|^2\right]
$$

$$
\le (1 - 2\mu\eta_t\tau)\mathbb{E}\left[\|\boldsymbol{\theta}_{i,t} - \boldsymbol{\theta}^*\|^2\right] + 2L\tau(2\tau+3)\eta_t^2\Gamma
$$

$$
+ (2 - 2\mu\eta_t)\frac{1}{N}\sum_{m=1}^{N}\sum_{j=1}^{\tau}\mathbb{E}\left[\left\|\boldsymbol{\theta}_{m,t}^j - \boldsymbol{\theta}_{i,t}\right\|^2\right] + \eta_t^2\tau^2\tilde{\Sigma}^2 .
$$

(11)

*Proof.* In this proof, we define

$$
\bar{R}_{i,t} = -\eta_t\frac{1}{\gamma_{i,t}'N}\sum_{m \in \mathcal{S}_{i,t}}\sum_{j=1}^{\tau}\nabla F_m(\boldsymbol{\theta}_{m,t}^j) .
$$

(12)

To simplify the notation in this proof, we use $\boldsymbol{\theta}_{i,t}$ interchangeably with $\boldsymbol{\theta}_{i,t}^1$ to denote the model of client $i$ at FL iteration $t$ without performing any local updates.

We write

$$
\|\boldsymbol{\theta}_{i,t+1} - \boldsymbol{\theta}^*\|^2
$$

$$
= \|\boldsymbol{\theta}_{i,t} + R_{i,t} - \boldsymbol{\theta}^* - \bar{R}_{i,t} + \bar{R}_{i,t}\|^2
$$

(13)

$$
= \|\boldsymbol{\theta}_{i,t} - \boldsymbol{\theta}^* + \bar{R}_{i,t}\|^2 + \|R_{i,t} - \bar{R}_{i,t}\|^2 + 2\langle\boldsymbol{\theta}_{i,t} - \boldsymbol{\theta}^* + \bar{R}_{i,t}, R_{i,t} - \bar{R}_{i,t}\rangle .
$$

**Step 1: Bounding the second and third term in (13)**

Bound the second term in (13) in expectation with respect to $\xi_{m,t}^j$ and $\mathcal{S}_{i,t}$,

$$
\mathbb{E}_{\mathcal{S},\xi}\left[\|R_{i,t} - \bar{R}_{i,t}\|^2\right]
$$

$$
= \mathbb{E}_{\mathcal{S},\xi}\left[\left\|-\eta_t\frac{1}{\gamma_{i,t}'N}\sum_{m \in \mathcal{S}_{i,t}}\sum_{j=1}^{\tau}(\nabla F_m(\boldsymbol{\theta}_{m,t}^j, \xi_{m,t}^j) - \nabla F_m(\boldsymbol{\theta}_{m,t}^j))\right\|^2\right]
$$

$$
\le \eta_t^2\tau\mathbb{E}_{\mathcal{S}}\left[\frac{1}{\gamma_{i,t}'N}\sum_{m \in \mathcal{S}_{i,t}}\sum_{j=1}^{\tau}\mathbb{E}_\xi\left[\left\|\nabla F_m(\boldsymbol{\theta}_{m,t}^j, \xi_{m,t}^j) - \nabla F_m(\boldsymbol{\theta}_{m,t}^j))\right\|^2\right]\right]
$$

(14)

$$
\stackrel{(a)}{\le} \eta_t^2\tau^2\mathbb{E}_{\mathcal{S}}\left[\frac{1}{\gamma_{i,t}'N}\sum_{m \in \mathcal{S}_{i,t}}^{N}\tilde{\sigma}_m^2\right]
$$

$$
\stackrel{(b)}{=} \eta_t^2\tau^2\tilde{\Sigma}^2
$$

where (a) and (b) both follow from Assumption 3.

The expectation of the third term in (13) is zero. That is,

$$
\mathbb{E}\left[2\langle\boldsymbol{\theta}_{i,t} - \boldsymbol{\theta}^* + \bar{R}_{i,t}, R_{i,t} - \bar{R}_{i,t}\rangle\right] = 0
$$

(15)

because $\mathbb{E}\left[R_{i,t}\right] = \bar{R}_{i,t}$.

**Step 2: Bounding the first term in (13)**

We write

$$
\|\boldsymbol{\theta}_{i,t} - \boldsymbol{\theta}^* + \bar{R}_{i,t}\|^2 = \|\boldsymbol{\theta}_{i,t} - \boldsymbol{\theta}^*\|^2 + \|\bar{R}_{i,t}\|^2 + 2\langle\boldsymbol{\theta}_{i,t} - \boldsymbol{\theta}^*, \bar{R}_{i,t}\rangle .
$$

(16)

We will next bound $\|\bar{R}_{i,t}\|^2$ and $2\langle\boldsymbol{\theta}_{i,t} - \boldsymbol{\theta}^*, \bar{R}_{i,t}\rangle$, respectively.

**Step 2.1: Bounding $\left\|\bar{R}_{i,t}\right\|^2$**

From the Assumption 1 of $L$-smoothness for $F_m$, due to Lemma 4 of (Zhou, 2018), we have

$$\left\|\nabla F_m(\boldsymbol{\theta}_{m,t}^j)\right\|^2 \leq 2L(F_m(\boldsymbol{\theta}_{m,t}^j)) - F_m^*) . \tag{17}$$

Bounding the second term in the RHS of (16), we have

$$
\begin{aligned}
\left\|\bar{R}_{i,t}\right\|^2 &= \eta_t^2 \left\| \frac{1}{\gamma_{i,t}' N} \sum_{m \in \mathcal{S}_{i,t}} \sum_{j=1}^{\tau} \nabla F_m(\boldsymbol{\theta}_{m,t}^j) \right\|^2 \\
&\leq \eta_t^2 \frac{1}{\gamma_{i,t}' N} \sum_{m \in \mathcal{S}_{i,t}} \left\| \sum_{j=1}^{\tau} \nabla F_m(\boldsymbol{\theta}_{m,t}^j) \right\|^2 \\
&\leq \eta_t^2 \tau \frac{1}{\gamma_{i,t}' N} \sum_{m \in \mathcal{S}_{i,t}} \sum_{j=1}^{\tau} \left\| \nabla F_m(\boldsymbol{\theta}_{m,t}^j) \right\|^2 \\
&\leq 2L \eta_t^2 \tau \frac{1}{\gamma_{i,t}' N} \sum_{m \in \mathcal{S}_{i,t}} \sum_{j=1}^{\tau} (F_m(\boldsymbol{\theta}_{m,t}^j) - F_m^*) .
\end{aligned}
\tag{18}
$$

**Step 2.2: Bounding $2\langle \boldsymbol{\theta}_{i,t} - \boldsymbol{\theta}^*, \bar{R}_{i,t} \rangle$**

Bounding the last term in the RHS of (16), we rewrite

$$
\begin{aligned}
2\langle \boldsymbol{\theta}_{i,t} - \boldsymbol{\theta}^*, \bar{R}_{i,t} \rangle &= 2\langle \boldsymbol{\theta}_{i,t} - \boldsymbol{\theta}^*, -\eta_t \frac{1}{\gamma_{i,t}' N} \sum_{m \in \mathcal{S}_{i,t}} \sum_{j=1}^{\tau} \nabla F_m(\boldsymbol{\theta}_{m,t}^j) \rangle \\
&= 2\eta_t \frac{1}{\gamma_{i,t}' N} \sum_{m \in \mathcal{S}_{i,t}} \sum_{j=1}^{\tau} \langle \boldsymbol{\theta}^* - \boldsymbol{\theta}_{i,t}, \nabla F_m(\boldsymbol{\theta}_{m,t}^j) \rangle \\
&= 2\eta_t \frac{1}{\gamma_{i,t}' N} \sum_{m \in \mathcal{S}_{i,t}} \sum_{j=1}^{\tau} \langle \boldsymbol{\theta}_{m,t}^j - \boldsymbol{\theta}_{i,t}, \nabla F_m(\boldsymbol{\theta}_{m,t}^j) \rangle \\
&\quad + 2\eta_t \frac{1}{\gamma_{i,t}' N} \sum_{m \in \mathcal{S}_{i,t}} \sum_{j=1}^{\tau} \langle \boldsymbol{\theta}^* - \boldsymbol{\theta}_{m,t}^j, \nabla F_m(\boldsymbol{\theta}_{m,t}^j) \rangle .
\end{aligned}
\tag{19}
$$

Consider the first term in (19),

$$
\begin{aligned}
&2\eta_t \frac{1}{\gamma_{i,t}' N} \sum_{m \in \mathcal{S}_{i,t}} \sum_{j=1}^{\tau} \langle \boldsymbol{\theta}_{m,t}^j - \boldsymbol{\theta}_{i,t}, \nabla F_m(\boldsymbol{\theta}_{m,t}^j) \rangle \\
&\leq \eta_t \frac{1}{\gamma_{i,t}' N} \sum_{m \in \mathcal{S}_{i,t}} \sum_{j=1}^{\tau} \left[ \frac{1}{\eta_t} \left\| \boldsymbol{\theta}_{m,t}^j - \boldsymbol{\theta}_{i,t} \right\|^2 + \eta_t \left\| \nabla F_m(\boldsymbol{\theta}_{m,t}^j) \right\|^2 \right] \\
&= \frac{1}{\gamma_{i,t}' N} \sum_{m \in \mathcal{S}_{i,t}} \sum_{j=1}^{\tau} \left\| \boldsymbol{\theta}_{m,t}^j - \boldsymbol{\theta}_{i,t} \right\|^2 + \eta_t^2 \frac{1}{\gamma_{i,t}' N} \sum_{m \in \mathcal{S}_{i,t}} \sum_{j=1}^{\tau} \left\| \nabla F_m(\boldsymbol{\theta}_{m,t}^j) \right\|^2 .
\end{aligned}
\tag{20}
$$

Consider the second term in (19),

$$
\begin{aligned}
&2\eta_t \frac{1}{\gamma_{i,t}' N} \sum_{m \in \mathcal{S}_{i,t}} \sum_{j=1}^{\tau} \langle \boldsymbol{\theta}^* - \boldsymbol{\theta}_{m,t}^j, \nabla F_m(\boldsymbol{\theta}_{m,t}^j) \rangle \\
&\stackrel{(a)}{\leq} 2\eta_t \frac{1}{\gamma_{i,t}' N} \sum_{m \in \mathcal{S}_{i,t}} \sum_{j=1}^{\tau} \left[ F_m(\boldsymbol{\theta}^*) - F_m(\boldsymbol{\theta}_{m,t}^j) - \frac{\mu}{2} \left\| \boldsymbol{\theta}_{m,t}^j - \boldsymbol{\theta}^* \right\|^2 \right]
\end{aligned}
\tag{21}
$$

where (a) follows from Assumption 2.

### Step 3: Putting the results together

In the previous step, we want to bound (16). Applying (18), (20) and (21), we have

$$\left\|\boldsymbol{\theta}_{i,t} - \boldsymbol{\theta}^* + \bar{R}_{i,t}\right\|^2$$

$$= \left\|\boldsymbol{\theta}_{i,t} - \boldsymbol{\theta}^*\right\|^2 + \left\|\bar{R}_{i,t}\right\|^2 + 2\langle\boldsymbol{\theta}_{i,t} - \boldsymbol{\theta}^*, \bar{R}_{i,t}\rangle$$

$$\leq \left\|\boldsymbol{\theta}_{i,t} - \boldsymbol{\theta}^*\right\|^2 + 2L\eta_t^2\tau\frac{1}{\gamma'_{i,t}N}\sum_{m\in\mathcal{S}_{i,t}}\sum_{j=1}^{\tau}(F_m(\boldsymbol{\theta}_{m,t}^j) - F_m^*)$$

$$+ \frac{1}{\gamma'_{i,t}N}\sum_{m\in\mathcal{S}_{i,t}}\sum_{j=1}^{\tau}\left\|\boldsymbol{\theta}_{m,t}^j - \boldsymbol{\theta}_{i,t}\right\|^2 + \eta_t^2\frac{1}{\gamma'_{i,t}N}\sum_{m\in\mathcal{S}_{i,t}}\sum_{j=1}^{\tau}\left\|\nabla F_m(\boldsymbol{\theta}_{m,t}^j)\right\|^2$$

$$+ 2\eta(t)\frac{1}{\gamma'_{i,t}N}\sum_{m\in\mathcal{S}_{i,t}}\sum_{j=1}^{\tau}\left[F_m(\boldsymbol{\theta}^*) - F_m(\boldsymbol{\theta}_{m,t}^j) - \frac{\mu}{2}\left\|\boldsymbol{\theta}_{m,t}^j - \boldsymbol{\theta}^*\right\|^2\right]$$

$$\overset{(a)}{=} \left\|\boldsymbol{\theta}_{i,t} - \boldsymbol{\theta}^*\right\|^2 - \mu\eta_t\frac{1}{\gamma'_{i,t}N}\sum_{m\in\mathcal{S}_{i,t}}\sum_{j=1}^{\tau}\left\|\boldsymbol{\theta}_{m,t}^j - \boldsymbol{\theta}^*\right\|^2 + \frac{1}{\gamma'_{i,t}N}\sum_{m\in\mathcal{S}_{i,t}}\sum_{j=1}^{\tau}\left\|\boldsymbol{\theta}_{m,t}^j - \boldsymbol{\theta}_{i,t}\right\|^2$$

$$+ \underbrace{2L(\tau+1)\eta_t^2\frac{1}{\gamma'_{i,t}N}\sum_{m\in\mathcal{S}_{i,t}}\sum_{j=1}^{\tau}(F_m(\boldsymbol{\theta}_{m,t}^j) - F_m^*) + 2\eta_t\frac{1}{\gamma'_{i,t}N}\sum_{m\in\mathcal{S}_{i,t}}\sum_{j=1}^{\tau}\left(F_m(\boldsymbol{\theta}^*) - F_m(\boldsymbol{\theta}_{m,t}^j)\right)}_{A_t}$$

$$\overset{(b)}{\leq} \left\|\boldsymbol{\theta}_{i,t} - \boldsymbol{\theta}^*\right\|^2 - 2\mu\eta_t\frac{1}{\gamma'_{i,t}N}\sum_{m\in\mathcal{S}_{i,t}}\sum_{j=1}^{\tau}\left\|\boldsymbol{\theta}_{i,t} - \boldsymbol{\theta}^*\right\|^2$$

$$+ (1 - 2\mu\eta_t)\frac{1}{\gamma'_{i,t}N}\sum_{m\in\mathcal{S}_{i,t}}\sum_{j=1}^{\tau}\left\|\boldsymbol{\theta}_{m,t}^j - \boldsymbol{\theta}_{i,t}\right\|^2 + A_t$$

$$= (1 - 2\mu\eta_t\tau)\left\|\boldsymbol{\theta}_{i,t} - \boldsymbol{\theta}^*\right\|^2 + (1 - 2\mu\eta_t)\frac{1}{\gamma'_{i,t}N}\sum_{m\in\mathcal{S}_{i,t}}\sum_{j=1}^{\tau}\left\|\boldsymbol{\theta}_{m,t}^j - \boldsymbol{\theta}_{i,t}\right\|^2 + A_t \tag{22}$$

where (a) follows from $L$-smoothness in (17) and rearranging and (b) follows from $\left\|\boldsymbol{\theta}_{m,t}^j - \boldsymbol{\theta}^*\right\|^2 \leq 2\left\|\boldsymbol{\theta}_{m,t}^j - \boldsymbol{\theta}_{i,t}\right\|^2 + 2\left\|\boldsymbol{\theta}_{i,t} - \boldsymbol{\theta}^*\right\|^2$.

We next bound $A_t$ in expectation. Define $W_t = 2\eta_t(1 - 2L(\tau+1)\eta_t)$. Note that we define $\eta_t$ such that $\eta_t \leq \frac{1}{4L(\tau+1)}$. Hence, $W_t > 0$.

From the definition of $A$ above, we have

$$
\begin{aligned}
A_t &= 2L(\tau+1)\eta_t^2 \frac{1}{\gamma'_{i,t}N} \sum_{m\in\mathcal{S}_{i,t}} \sum_{j=1}^{\tau}(F_m(\boldsymbol{\theta}_{m,t}^j) - F_m^*) \\
&\quad + 2\eta_t \frac{1}{\gamma'_{i,t}N} \sum_{m\in\mathcal{S}_{i,t}} \sum_{j=1}^{\tau}\Big(F_m(\boldsymbol{\theta}^*) - F_m^* + F_m^* - F_m(\boldsymbol{\theta}_{m,t}^j)\Big) \\
&= -2\eta_t(1 - 2L(\tau+1)\eta_t)\frac{1}{\gamma'_{i,t}N} \sum_{m\in\mathcal{S}_{i,t}} \sum_{j=1}^{\tau}(F_m(\boldsymbol{\theta}_{m,t}^j) - F_m^*) \\
&\quad + 2\eta_t \frac{1}{\gamma'_{i,t}N} \sum_{m\in\mathcal{S}_{i,t}} \sum_{j=1}^{\tau}(F_m(\boldsymbol{\theta}^*) - F_m^*) \\
&= -W_t \frac{1}{\gamma'_{i,t}N} \sum_{m\in\mathcal{S}_{i,t}} \sum_{j=1}^{\tau}(F_m(\boldsymbol{\theta}_{m,t}^j) - F^*) - W_t\tau\frac{1}{\gamma'_{i,t}N} \sum_{m\in\mathcal{S}_{i,t}} (F^* - F_m^*) \\
&\quad + 2\eta_t\tau \frac{1}{\gamma'_{i,t}N} \sum_{m\in\mathcal{S}_{i,t}} (F_m(\boldsymbol{\theta}^*) - F_m^*) \ .
\end{aligned}
\tag{23}
$$

Therefore,

$$
\begin{aligned}
\mathbb{E}[A_t] &= \mathbb{E}\left[-W_t\frac{1}{\gamma'_{i,t}N} \sum_{m\in\mathcal{S}_{i,t}} \sum_{j=1}^{\tau}(F_m(\boldsymbol{\theta}_{m,t}^j) - F^*)\right] + \\
&\quad - W_t\tau\frac{1}{\gamma'_{i,t}N}\mathbb{E}\left[\sum_{m\in\mathcal{S}_{i,t}}(F^* - F_m^*)\right] + 2\eta_t\tau\frac{1}{\gamma'_{i,t}N}\mathbb{E}\left[\sum_{m\in\mathcal{S}_{i,t}}(F_m(\boldsymbol{\theta}^*) - F_m^*)\right] \\
&\overset{(a)}{=} \mathbb{E}\left[-W_t\frac{1}{\gamma'_{i,t}N} \sum_{m\in\mathcal{S}_{i,t}} \sum_{j=1}^{\tau}(F_m(\boldsymbol{\theta}_{m,t}^j) - F^*)\right] + 4L\tau(\tau+1)\eta_t^2\Gamma
\end{aligned}
\tag{24}
$$

where (a) follows from $\mathbb{E}\left[\frac{1}{\gamma'_{i,t}N}\sum_{m\in\mathcal{S}_{i,t}}(F^* - F_m^*)\right] = \frac{1}{N}\sum_{m=1}^{N}(F^* - F_m^*) = \Gamma$.

Note that in the equation above, we have

$$
\begin{aligned}
&\frac{1}{\gamma'_{i,t}N} \sum_{m\in\mathcal{S}_{i,t}} \sum_{j=1}^{\tau}(F_m(\boldsymbol{\theta}_{m,t}^j) - F^*) \\
&= \frac{1}{\gamma'_{i,t}N} \sum_{m\in\mathcal{S}_{i,t}} \sum_{j=1}^{\tau}(F_m(\boldsymbol{\theta}_{m,t}^j) - F_m(\boldsymbol{\theta}_{i,t})) + \frac{1}{\gamma'_{i,t}N} \sum_{m\in\mathcal{S}_{i,t}} \sum_{j=1}^{\tau}(F_m(\boldsymbol{\theta}_{i,t}) - F^*) \\
&\overset{(a)}{\geq} \frac{1}{\gamma'_{i,t}N} \sum_{m\in\mathcal{S}_{i,t}} \sum_{j=1}^{\tau}\langle\nabla F_m(\boldsymbol{\theta}_{i,t}), \boldsymbol{\theta}_{m,t}^j - \boldsymbol{\theta}_{i,t}\rangle + \frac{1}{\gamma'_{i,t}N} \sum_{m\in\mathcal{S}_{i,t}} \sum_{j=1}^{\tau}(F_m(\boldsymbol{\theta}_{i,t}) - F^*) \\
&\geq -\frac{1}{2\gamma'_{i,t}N} \sum_{m\in\mathcal{S}_{i,t}} \sum_{j=1}^{\tau}\left[\eta_t\|\nabla F_m(\boldsymbol{\theta}_{i,t})\|^2 + \frac{1}{\eta_t}\left\|\boldsymbol{\theta}_{m,t}^j - \boldsymbol{\theta}_{i,t}\right\|^2\right] \\
&\quad + \frac{1}{\gamma'_{i,t}N} \sum_{m\in\mathcal{S}_{i,t}} \sum_{j=1}^{\tau}(F_m(\boldsymbol{\theta}_{i,t}) - F^*) \\
&\overset{(b)}{\geq} -\frac{1}{\gamma'_{i,t}N} \sum_{m\in\mathcal{S}_{i,t}} \sum_{j=1}^{\tau}\left[\eta_t L(F_m(\boldsymbol{\theta}_{i,t}) - F_m^*) + \frac{1}{2\eta_t}\left\|\boldsymbol{\theta}_{m,t}^j - \boldsymbol{\theta}_{i,t}\right\|^2\right] \\
&\quad + \frac{1}{\gamma'_{i,t}N} \sum_{m\in\mathcal{S}_{i,t}} \sum_{j=1}^{\tau}(F_m(\boldsymbol{\theta}_{i,t}) - F^*)
\end{aligned}
\tag{25}
$$

where (a) follows from Assumption 2 and (b) follows from (17). Therefore,

$$
\begin{aligned}
&- W_t \frac{1}{\gamma'_{i,t} N} \sum_{m \in \mathcal{S}_{i,t}} \sum_{j=1}^{\tau} (F_m(\boldsymbol{\theta}^j_{m,t}) - F^*) \\
&\leq W_t \frac{1}{\gamma'_{i,t} N} \sum_{m \in \mathcal{S}_{i,t}} \sum_{j=1}^{\tau} \left[ \eta_t L (F_m(\boldsymbol{\theta}_{i,t}) - F^*_m) + \frac{1}{2\eta_t} \left\| \boldsymbol{\theta}^j_{m,t} - \boldsymbol{\theta}_{i,t} \right\|^2 \right] \\
&\quad - W_t \frac{1}{\gamma'_{i,t} N} \sum_{m \in \mathcal{S}_{i,t}} \sum_{j=1}^{\tau} (F_m(\boldsymbol{\theta}_{i,t}) - F^*) \\
&= W_t(\eta_t L - 1) \frac{1}{\gamma'_{i,t} N} \sum_{m \in \mathcal{S}_{i,t}} \sum_{j=1}^{\tau} (F_m(\boldsymbol{\theta}_{i,t}) - F^*) \\
&\quad + W_t \eta_t L \frac{1}{\gamma'_{i,t} N} \sum_{m \in \mathcal{S}_{i,t}} \sum_{j=1}^{\tau} (F^* - F^*_m) + \frac{W_t}{2\eta_t} \frac{1}{\gamma'_{i,t} N} \sum_{m \in \mathcal{S}_{i,t}} \sum_{j=1}^{\tau} \left\| \boldsymbol{\theta}^j_{m,t} - \boldsymbol{\theta}_{i,t} \right\|^2 \\
&\overset{(a)}{\leq} W_t \eta_t L \frac{1}{\gamma'_{i,t} N} \sum_{m \in \mathcal{S}_{i,t}} \sum_{j=1}^{\tau} (F^* - F^*_m) + \frac{1}{\gamma'_{i,t} N} \sum_{m \in \mathcal{S}_{i,t}} \sum_{j=1}^{\tau} \left\| \boldsymbol{\theta}^j_{m,t} - \boldsymbol{\theta}_{i,t} \right\|^2
\end{aligned}
\tag{26}
$$

where (a) follows from $\eta_t L - 1 \leq 0$ and $\frac{1}{\gamma'_{i,t} N} \sum_{m \in \mathcal{S}_{i,t}} \sum_{j=1}^{\tau} (F_m(\boldsymbol{\theta}_{i,t}) - F^*) \geq 0$, and also from the fact that $\eta_t \leq W_t \leq 2\eta_t$ given that $\eta_t \leq \frac{1}{4L(\tau+1)}$.

Then, substituting (26) into (24),

$$
\begin{aligned}
\mathbb{E}[A_t] &\leq W_t \eta_t L \tau \frac{1}{\gamma'_{i,t} N} \mathbb{E} \left[ \sum_{m \in \mathcal{S}_{i,t}} (F^* - F^*_m) \right] \\
&\quad + \mathbb{E} \left[ \frac{1}{\gamma'_{i,t} N} \sum_{m \in \mathcal{S}_{i,t}} \sum_{j=1}^{\tau} \left\| \boldsymbol{\theta}^j_{m,t} - \boldsymbol{\theta}_{i,t} \right\|^2 \right] + 4L\tau(\tau+1)\eta_t^2 \Gamma \\
&\overset{(a)}{\leq} 2L\tau \eta_t^2 (2\tau+3)\Gamma + \mathbb{E} \left[ \frac{1}{\gamma'_{i,t} N} \sum_{m \in \mathcal{S}_{i,t}} \sum_{j=1}^{\tau} \left\| \boldsymbol{\theta}^j_{m,t} - \boldsymbol{\theta}_{i,t} \right\|^2 \right]
\end{aligned}
\tag{27}
$$

where (a) again follows from $\mathbb{E} \left[ \frac{1}{\gamma'_{i,t} N} \sum_{m \in \mathcal{S}_{i,t}} (F^* - F^*_m) \right] = \frac{1}{N} \sum_{m=1}^{N} (F^* - F^*_m) = \Gamma$.

Finally, taking expectation of (22) and then substituting (27), we have

$$
\begin{aligned}
&\mathbb{E} \left[ \left\| \boldsymbol{\theta}_{i,t} - \boldsymbol{\theta}^* + \bar{R}_{i,t} \right\|^2 \right] \\
&\leq (1 - 2\mu\eta_t\tau) \mathbb{E} \left[ \left\| \boldsymbol{\theta}_{i,t} - \boldsymbol{\theta}^* \right\|^2 \right] + (1 - 2\mu\eta_t) \mathbb{E} \left[ \frac{1}{\gamma'_{i,t} N} \sum_{m \in \mathcal{S}_{i,t}} \sum_{j=1}^{\tau} \left\| \boldsymbol{\theta}^j_{m,t} - \boldsymbol{\theta}_{i,t} \right\|^2 \right] \\
&\quad + 2L\tau(2\tau+3)\eta_t^2 \Gamma + \mathbb{E} \left[ \frac{1}{\gamma'_{i,t} N} \sum_{m \in \mathcal{S}_{i,t}} \sum_{j=1}^{\tau} \left\| \boldsymbol{\theta}^j_{m,t} - \boldsymbol{\theta}_{i,t} \right\|^2 \right] \\
&= (1 - 2\mu\eta_t\tau) \mathbb{E} \left[ \left\| \boldsymbol{\theta}_{i,t} - \boldsymbol{\theta}^* \right\|^2 \right] + 2L\tau(2\tau+3)\eta_t^2 \Gamma \\
&\quad + (2 - 2\mu\eta_t) \frac{1}{N} \sum_{m=1}^{N} \sum_{j=1}^{\tau} \mathbb{E} \left[ \left\| \boldsymbol{\theta}^j_{m,t} - \boldsymbol{\theta}_{i,t} \right\|^2 \right].
\end{aligned}
\tag{28}
$$

**Step 4: Concluding the proof**

Lastly, we take expectation of (13) and substitute (28), (14), (15),

$$
\begin{aligned}
&\mathbb{E}\left[\left\|\boldsymbol{\theta}_{i,t+1} - \boldsymbol{\theta}^*\right\|^2\right]\\
&= \mathbb{E}\left[\left\|\boldsymbol{\theta}_{i,t} - \boldsymbol{\theta}^* + \bar{R}_{i,t}\right\|^2\right] + \mathbb{E}\left[\left\|R_{i,t} - \bar{R}_{i,t}\right\|^2\right]\\
&\overset{(a)}{\leq} (1 - 2\mu\eta_t\tau)\mathbb{E}\left[\left\|\boldsymbol{\theta}_{i,t} - \boldsymbol{\theta}^*\right\|^2\right] + 2L\tau(2\tau+3)\eta_t^2\Gamma\\
&\quad + (2 - 2\mu\eta_t)\frac{1}{N}\sum_{m=1}^{N}\sum_{j=1}^{\tau}\mathbb{E}\left[\left\|\boldsymbol{\theta}_{m,t}^j - \boldsymbol{\theta}_{i,t}\right\|^2\right] + \eta_t^2\tau^2\tilde{\Sigma}^2 .
\end{aligned}
\tag{29}
$$

Note that we have two sources of uncertainties above. They are 1) the random selection of updates from $\mathcal{S}_{i,t}$ in $R_{i,t}$ and 2) the stochastic gradients $\xi_{m,t}^j$. We have taken expectation with respect to both of them above.

$\square$

**Lemma D.2.** *Given Assumption 4,*

$$
\mathbb{E}\left[\left\|\boldsymbol{\theta}_{w,t}^j - \boldsymbol{\theta}_{v,t}^1\right\|^2\right] \leq \tau^2 G^2 \sum_{z=1}^{t} \eta_z^2\left(\frac{1}{\gamma_{w,z}'} + \frac{1}{\gamma_{v,z}'} - 2\right) + \eta_t^2(j-1)^2 G^2 .
\tag{30}
$$

*Proof.* Using the Cauchy-Schwarz inequality, we first bound

$$
\begin{aligned}
&\mathbb{E}\left[\left\|\boldsymbol{\theta}_{w,t}^1 - \boldsymbol{\theta}_{v,t}^1\right\|^2\right]\\
&= \mathbb{E}\left[\left\|\sum_{z=1}^{t}(R_{w,z} - R_{v,z})\right\|^2\right]\\
&\overset{(a)}{=} \sum_{z=1}^{t}\mathbb{E}\left[\left\|R_{w,z} - R_{v,z}\right\|^2\right]\\
&= \sum_{z=1}^{t}\mathbb{E}\left[\eta_z^2\left\|\frac{1}{|\mathcal{S}_{w,z}|}\sum_{m\in\mathcal{S}_{w,z}}\sum_{j=1}^{\tau}\nabla F_m(\boldsymbol{\theta}_{m,z}^j, \xi_{m,z}^j) - \frac{1}{|\mathcal{S}_{v,z}|}\sum_{m\in\mathcal{S}_{v,z}}\sum_{j=1}^{\tau}\nabla F_m(\boldsymbol{\theta}_{m,z}^j, \xi_{m,z}^j)\right\|^2\right]\\
&\overset{(b)}{=} \sum_{z=1}^{t}\eta_z^2\mathbb{E}\left[\left\|\left(\frac{1}{\gamma_{w,z}'N} - \frac{1}{\gamma_{v,z}'N}\right)\sum_{\substack{m\in\mathcal{S}_{w,z}\\\wedge m\in\mathcal{S}_{v,z}}}\sum_{j=1}^{\tau}\nabla F_m(\boldsymbol{\theta}_{m,z}^j, \xi_{m,z}^j)\right.\right.\\
&\quad \left.\left. + \frac{1}{\gamma_{w,z}'N}\sum_{\substack{m\in\mathcal{S}_{w,z}\\\wedge m\notin\mathcal{S}_{v,z}}}\sum_{j=1}^{\tau}\nabla F_m(\boldsymbol{\theta}_{m,z}^j, \xi_{m,z}^j) - \frac{1}{\gamma_{v,z}'N}\sum_{\substack{m\in\mathcal{S}_{v,z}\\\wedge m\notin\mathcal{S}_{w,z}}}\sum_{j=1}^{\tau}\nabla F_m(\boldsymbol{\theta}_{m,z}^j, \xi_{m,z}^j)\right\|^2\right]\\
&\overset{(c)}{\leq} \sum_{z=1}^{t}\eta_z^2\left[\gamma_{w,z}'\gamma_{v,z}'N\left(\frac{1}{\gamma_{w,z}'N} - \frac{1}{\gamma_{v,z}'N}\right)^2 + \gamma_{w,z}'(1 - \gamma_{v,z}')N\left(\frac{1}{\gamma_{w,z}'N}\right)^2\right.\\
&\quad \left. + \gamma_{v,z}'(1 - \gamma_{w,z}')N\left(\frac{1}{\gamma_{v,z}'N}\right)^2\right]\mathbb{E}\left[\sum_{m=1}^{N}\left\|\sum_{j=1}^{\tau}\nabla F_m(\boldsymbol{\theta}_{m,z}^j, \xi_{m,z}^j)\right\|^2\right]\\
&= \sum_{z=1}^{t}\eta_z^2\left(\frac{1}{\gamma_{w,z}'} + \frac{1}{\gamma_{v,z}'} - 2\right)\frac{1}{N}\left(\sum_{m=1}^{N}\mathbb{E}\left[\left\|\sum_{j=1}^{\tau}\nabla F_m(\boldsymbol{\theta}_{m,z}^j, \xi_{m,z}^j)\right\|^2\right]\right)
\end{aligned}
$$

$$\overset{(d)}{\leq} \tau^2 G^2 \sum_{z=1}^{t} \eta_z^2 \left( \frac{1}{\gamma'_{w,z}} + \frac{1}{\gamma'_{v,z}} - 2 \right) \tag{31}$$

where (a) follows from the independent nature of $R_{i,z}$'s at different $z$, $\forall i$ and $\mathbb{E}\left[R_{w,z} - R_{s,z}\right] = 0$, (b) follows from the algorithm such that $|\mathcal{S}_{i,z}| = \gamma_{i,z}(N-1)+1 = \gamma'_{i,z}N$ for all $i \in [N]$, (c) follows from expected number of party indices in $\mathcal{S}_{w,z}, \mathcal{S}_{v,z}$ and (d) follows from Assumption 4. Note that here we assume $\gamma'_{i,z}N$ is a positive integer w.l.o.g.

Then, we bound the model that has performed $j-1$ local gradient steps,

$$\mathbb{E}\left[\left\|\boldsymbol{\theta}_{w,t}^j - \boldsymbol{\theta}_{v,t}^1\right\|^2\right] \tag{32}$$

$$= \mathbb{E}\left[\left\|\boldsymbol{\theta}_{w,t}^1 - \eta_t \sum_{b=1}^{j-1} \nabla F_w(\boldsymbol{\theta}_{w,t}^b, \xi_{w,t}^b) - \boldsymbol{\theta}_{v,t}^1\right\|^2\right] \tag{33}$$

$$= \mathbb{E}\left[\left\|\boldsymbol{\theta}_{w,t}^1 - \boldsymbol{\theta}_{v,t}^1\right\|^2\right] + \mathbb{E}\left[\left\|-\eta_t \sum_{b=1}^{j-1} \nabla F_w(\boldsymbol{\theta}_{w,t}^b, \xi_{w,t}^b)\right\|^2\right] \tag{34}$$

$$+ \mathbb{E}\left[2\langle \boldsymbol{\theta}_{w,t}^1 - \boldsymbol{\theta}_{v,t}^1, -\eta_t \sum_{b=1}^{j-1} \nabla F_w(\boldsymbol{\theta}_{w,t}^b, \xi_{w,t}^b) \rangle\right] \tag{35}$$

$$\overset{(a)}{=} \mathbb{E}\left[\left\|\boldsymbol{\theta}_{w,t}^1 - \boldsymbol{\theta}_{v,t}^1\right\|^2\right] + \mathbb{E}\left[\left\|-\eta_t \sum_{b=1}^{j-1} \nabla F_w(\boldsymbol{\theta}_{w,t}^b, \xi_{w,t}^b)\right\|^2\right] \tag{36}$$

$$\leq \mathbb{E}\left[\left\|\boldsymbol{\theta}_{w,t}^1 - \boldsymbol{\theta}_{v,t}^1\right\|^2\right] + \eta_t^2(j-1) \sum_{b=1}^{j-1} \mathbb{E}\left[\left\|\nabla F_i(\boldsymbol{\theta}_{i,t}^b, \xi_{i,t}^b)\right\|^2\right] \tag{37}$$

$$\overset{(b)}{\leq} \mathbb{E}\left[\left\|\boldsymbol{\theta}_{w,t}^1 - \boldsymbol{\theta}_{v,t}^1\right\|^2\right] + \eta_t^2(j-1)^2 G^2 \tag{38}$$

$$\overset{(c)}{\leq} \tau^2 G^2 \sum_{z=1}^{t} \eta_z^2 \left( \frac{1}{\gamma'_{w,z}} + \frac{1}{\gamma'_{v,z}} - 2 \right) + \eta_t^2(j-1)^2 G^2 \tag{39}$$

where (a) follows from $\mathbb{E}\left[\boldsymbol{\theta}_{w,t}^1 - \boldsymbol{\theta}_{v,t}^1\right] = \sum_{z=0}^{t-1} \mathbb{E}\left[R_{w,z} - R_{v,z}\right] = 0$, (b) follows from Assumption 4 and (c) follows from (31). □

### D.3 PROOF OF THEOREM 2

**Theorem 2** (Improved bound). *Let $\eta_t = \frac{\beta}{t+\alpha}$ and $B, H_T$ be defined in Theorem 1. With a stochastic recovery rate of $q$, the performance of client model $\boldsymbol{\theta}_{i,T}$ trained over $T$ FL iterations is bounded by*

$$\mathbb{E}\left[F(\boldsymbol{\theta}_{i,T})\right] - F^* \leq \frac{L}{2} \left( \prod_{t=1}^{T}(1 - 2\mu\eta_t\tau) \right) \|\boldsymbol{\theta}_1 - \boldsymbol{\theta}^*\|^2 + \frac{L}{2} \sum_{t=1}^{T} \eta_t^2 (Q + D_t + E_t) \prod_{l=t+1}^{T}(1 - 2\mu\eta_t\tau)$$

*where*
$$D_T = \frac{H_T}{N} \sum_{m=1}^{N} \sum_{t=1}^{T} \left(\frac{T+\alpha}{t+\alpha}\right)^2 \left( \frac{1}{\gamma'_{m,t}} + \frac{1}{\gamma'_{\text{ref},t}} - 2 \right) (1-q)^{T-t+1}$$

$$E_T = H_T \sum_{t=1}^{T} \left(\frac{T+\alpha}{t+\alpha}\right)^2 \left( \frac{1}{\gamma'_{i,t}} + \frac{1}{\gamma'_{\text{ref},t}} - 2 \right) (1-q)^{T-t+1} .$$

*Proof.* Let $0 \leq q \leq 1$ be the probability of stochastic recovery. During stochastic recovery, a client recovers a reference model. We denote $\boldsymbol{\theta}_{\text{ref}}$ as the reference model parameter and $R_{\text{ref},t}$ as the reward for the reference model at iteration $t$, which is determined by the induced reference reward rate $\gamma'_{\text{ref},t}$. Let $h$ be the number of iterations the client has not synchronized for.

With the introduction of the reference model, the difference $\|\boldsymbol{\theta}_{m,t} - \boldsymbol{\theta}_{i,t}\|^2$ between two models at an iteration $t$ can be broken down into $\|\boldsymbol{\theta}_{m,t} - \boldsymbol{\theta}_{\text{ref},t}\|^2$ and $\|\boldsymbol{\theta}_{\text{ref},t} - \boldsymbol{\theta}_{i,t}\|^2$, which can be better bounded using the stochastic recovery rate $0 \le q \le 1$.

We now derive a new bound for the expected difference between a client model $\boldsymbol{\theta}_{w,T}$ and the reference model $\boldsymbol{\theta}_{\text{ref},T}$ at an iteration $T$,

$$
\mathbb{E}\left[\left\|\boldsymbol{\theta}_{w,T}^1 - \boldsymbol{\theta}_{\text{ref},T}^1\right\|^2\right]
$$

$$
= \sum_{h=1}^{T} q(1-q)^h \mathbb{E}\left[\left\|\sum_{z=T+1-h}^{T} (R_{w,z} - R_{\text{ref},z})\right\|^2\right]
$$

$$
\stackrel{(a)}{=} \sum_{h=1}^{T} q(1-q)^h \left[\sum_{z=T+1-h}^{T} \mathbb{E}\left[\|(R_{w,z} - R_{\text{ref},z})\|^2\right]\right]
$$

$$
= q\left[\left[\sum_{h=1}^{T}(1-q)^h\right]\mathbb{E}\left[\|R_{w,T} - R_{\text{ref},T}\|^2\right] + \dots + \left[\sum_{h=T}^{T}(1-q)^h\right]\mathbb{E}\left[\|R_{w,1} - R_{\text{ref},1}\|^2\right]\right]
$$

$$
= q\sum_{l=1}^{T}\left[\left[\sum_{h=l}^{T}(1-q)^h\right]\mathbb{E}\left[\|R_{w,T+1-l} - R_{\text{ref},T+1-l}\|^2\right]\right]
$$

$$
\stackrel{(b)}{\le} q\tau^2 G^2 \sum_{l=1}^{T}\left(\frac{1}{\gamma'_{w,T+1-l}} + \frac{1}{\gamma'_{\text{ref},T+1-l}} - 2\right)\eta_{T+1-l}^2\left[\sum_{h=l}^{T}(1-q)^h\right]
$$

$$
\stackrel{(c)}{=} \tau^2 G^2 \sum_{l=1}^{T}\left(\frac{1}{\gamma'_{w,T+1-l}} + \frac{1}{\gamma'_{\text{ref},T+1-l}} - 2\right)\eta_{T+1-l}^2\left((1-q)^l - (1-q)^{T+1}\right)
$$

$$
\stackrel{(d)}{=} \tau^2 G^2 \sum_{t=1}^{T}\left(\frac{1}{\gamma'_{w,t}} + \frac{1}{\gamma'_{\text{ref},t}} - 2\right)\eta_t^2\left((1-q)^{T-t+1} - (1-q)^{T+1}\right)
$$

$$
\le \tau^2 G^2 \sum_{t=1}^{T}\left(\frac{1}{\gamma'_{w,t}} + \frac{1}{\gamma'_{\text{ref},t}} - 2\right)\eta_t^2(1-q)^{T-t+1} \tag{40}
$$

where (a) follows from the independent nature of $R_{i,t}$'s at different $t$, $\forall i$ and $\mathbb{E}\left[R_{w,z} - R_{s,z}\right] = 0$, (b) follows from (31), (c) follows from the closed form of the summation $\sum_{h=l}^{T}(1-q)^h = \frac{(1-q)^l - (1-q)^{T+1}}{q}$ and (d) follows from replacing $l$ with $t$ such that $t = T + 1 - l$.

From Lemma D.1, we have

$$
\mathbb{E}\left[\|\boldsymbol{\theta}_{i,T+1} - \boldsymbol{\theta}^*\|^2\right]
$$

$$
\le (1 - 2\mu\eta_T\tau)\mathbb{E}\left[\|\boldsymbol{\theta}_{i,T} - \boldsymbol{\theta}^*\|^2\right] + 2L\tau(2\tau+3)\eta_T^2\Gamma + \eta_t^2\tau^2\tilde{\Sigma}^2
$$

$$
+ (2 - 2\mu\eta_t)\frac{1}{N}\sum_{m=1}^{N}\sum_{j=1}^{\tau}\mathbb{E}\left[\left\|\boldsymbol{\theta}_{m,T}^j - \boldsymbol{\theta}_{i,T}\right\|^2\right]
$$

$$
\stackrel{(a)}{\le} (1 - 2\mu\eta_T\tau)\mathbb{E}\left[\|\boldsymbol{\theta}_{i,T} - \boldsymbol{\theta}^*\|^2\right] + 2L\tau(2\tau+3)\eta_T^2\Gamma + \eta_T^2\tau^2\tilde{\Sigma}^2
$$

$$
+ (2 - 2\mu\eta_T)\frac{1}{N}\sum_{m=1}^{N}\sum_{j=1}^{\tau}\left(\mathbb{E}\left[\left\|\boldsymbol{\theta}_{m,T}^1 - \boldsymbol{\theta}_{i,T}^1\right\|^2\right] + \eta_t^2(j-1)^2 G^2\right) \tag{41}
$$

$$
\stackrel{(b)}{\le} (1 - 2\mu\eta_T\tau)\mathbb{E}\left[\|\boldsymbol{\theta}_{i,T} - \boldsymbol{\theta}^*\|^2\right] + 2L\tau(2\tau+3)\eta_T^2\Gamma + 2\eta_T^2\tau^3 G^2 + \eta_T^2\tau^2\tilde{\Sigma}^2
$$

$$
+ (2 - 2\mu\eta_T)\tau\frac{1}{N}\sum_{m=1}^{N}\mathbb{E}\left[\left\|\boldsymbol{\theta}_{m,T}^1 - \boldsymbol{\theta}_{\text{ref},T}^1\right\|^2\right] + (2 - 2\mu\eta_T)\tau\mathbb{E}\left[\left\|\boldsymbol{\theta}_{\text{ref},T}^1 - \boldsymbol{\theta}_{i,T}^1\right\|^2\right]
$$

where (a) follows from (38) and (b) follows from $\mathbb{E}\left[\boldsymbol{\theta}_{i,T}^1\right] = \mathbb{E}\left[\boldsymbol{\theta}_{\text{ref},T}^1\right]$.

Then, we substitute (40),

$$
\mathbb{E}\left[\|\boldsymbol{\theta}_{i,T+1} - \boldsymbol{\theta}^*\|^2\right]
$$

$$
\leq (1 - 2\mu\eta_T\tau)\mathbb{E}\left[\|\boldsymbol{\theta}_{i,T} - \boldsymbol{\theta}^*\|^2\right] + 2L\tau(2\tau+3)\eta_T^2\Gamma + 2\eta_T^2\tau^3 G^2 + \eta_T^2\tau^2\tilde{\Sigma}^2
$$

$$
+ \eta_T^2(2 - 2\mu\eta_T)\tau^3 G^2 \frac{1}{N}\sum_{m=1}^{N}\sum_{t=1}^{T}\left(\frac{T+\alpha}{t+\alpha}\right)^2\left(\frac{1}{\gamma'_{m,t}} + \frac{1}{\gamma'_{\mathrm{ref},t}} - 2\right)(1-q)^{T-t+1} \tag{42}
$$

$$
+ \eta_T^2(2 - 2\mu\eta_T)\tau^3 G^2 \sum_{t=1}^{T}\left(\frac{T+\alpha}{t+\alpha}\right)^2\left(\frac{1}{\gamma'_{i,t}} + \frac{1}{\gamma'_{\mathrm{ref},t}} - 2\right)(1-q)^{T-t+1}
$$

$$
= (1 - 2\mu\eta_T\tau)\mathbb{E}\left[\|\boldsymbol{\theta}_{i,T} - \boldsymbol{\theta}^*\|^2\right] + \eta_T^2(Q + D_T + E_T)\,.
$$

Note that in the above, we have defined $Q = 2L\tau(2\tau+3)\Gamma + 2\tau^3 G^2 + \tau^2\tilde{\Sigma}^2$, $H_T = (2-2\mu\eta_T)\tau^3 G^2$ in Theorem 1 and we further define

$$
D_T = \frac{H_T}{N}\sum_{m=1}^{N}\sum_{t=1}^{T}\left(\frac{T+\alpha}{t+\alpha}\right)^2\left(\frac{1}{\gamma'_{m,t}} + \frac{1}{\gamma'_{\mathrm{ref},t}} - 2\right)(1-q)^{T-t+1}
$$

$$
E_T = H_T\sum_{t=1}^{T}\left(\frac{T+\alpha}{t+\alpha}\right)^2\left(\frac{1}{\gamma'_{i,t}} + \frac{1}{\gamma'_{\mathrm{ref},t}} - 2\right)(1-q)^{T-t+1}\,. \tag{43}
$$

From the $L$-smoothness of $F$,

$$
\mathbb{E}\left[F(\boldsymbol{\theta}_{i,T})\right] - F^*
$$

$$
\leq \frac{L}{2}\mathbb{E}\left[\|\boldsymbol{\theta}_{i,T} - \boldsymbol{\theta}^*\|^2\right]
$$

$$
\leq \frac{L}{2}\left(\prod_{t=1}^{T}(1 - 2\mu\eta_t\tau)\right)\|\boldsymbol{\theta}_1 - \boldsymbol{\theta}^*\|^2 + \frac{L}{2}\sum_{t=1}^{T}\eta_t^2(Q + D_t + E_t)\prod_{l=t+1}^{T}(1 - 2\mu\eta_t\tau)\,. \tag{44}
$$

$\square$

## D.4    Proof of Proposition 2

**Definition D.1.** *We write*
$$
f(x) = o(g(x)) \qquad \text{as } x \to \infty
$$
*if for all positive real number $M$, there exist a real number $x_0$ such that*
$$
|f(x)| < Mg(x) \qquad \forall x \geq x_0\,.
$$
*Or equivalently,*
$$
\lim_{x\to\infty}\frac{f(x)}{g(x)} = 0\,.
$$

**Definition D.2.** *We write*
$$
f(x) = O(g(x)) \qquad \text{as } x \to \infty
$$
*if there exists positive real numbers $M$ and $\delta$ such that for all defined $x$ with $0 < |x - a| < \delta$,*
$$
|f(x)| \leq Mg(x) \qquad \forall x \geq x_0\,.
$$
*Or equivalently,*
$$
\limsup_{x\to\infty}\frac{|f(x)|}{g(x)} < \infty\,.
$$

**Proposition 2** (Asymptotic convergence). *If $\frac{1}{\gamma'_{i,t}} = o\left(\frac{t^2}{\log t}\right)$ for all $i \in [N]$ and $\frac{1}{\gamma'_{\mathrm{ref},t}} = o\left(\frac{t^2}{\log t}\right)$, we achieve*
$$
\lim_{T\to\infty}\mathbb{E}\left[F(\boldsymbol{\theta}_{i,T})\right] - F^* = 0\,.
$$

*Proof.* Firstly, we define new functions $a_i(t) = \frac{1}{\gamma'_{i,t}}$ for all $i \in [N]$ such that $t \in \mathbb{Z}^+$. Define a non-decreasing function $\bar{a}_i(t)$ such that $\bar{a}_i(t) \geq a_i(t)$ for all $t$, specifically,

$$\bar{a}_i(t) = \max_{x \in [t]} a_i(x) . \tag{45}$$

Note that $\bar{a}_i(t) = o\left(\frac{t^2}{\log t}\right)$ and $\bar{a}_{\text{ref}}(t) = o\left(\frac{t^2}{\log t}\right)$ still hold.

Consider the term $E_T$ in (42),

$$
\begin{aligned}
E_T &= H_T \sum_{t=1}^{T} \left(\frac{T+\alpha}{t+\alpha}\right)^2 \left(\frac{1}{\gamma'_{i,t}} + \frac{1}{\gamma'_{\text{ref},t}} - 2\right) (1-q)^{T-t+1} \\
&\leq 2\tau^3 G^2 \sum_{t=1}^{T} (a_i(t) + a_{\text{ref}}(t)) \left(\frac{T+\alpha}{t+\alpha}\right)^2 (1-q)^{T-t+1} \\
&\overset{(a)}{\leq} 2\tau^3 G^2 \sum_{l=0}^{T-1} (a_i(T-l) + a_{\text{ref}}(T-l)) \left(\frac{T+\alpha}{T-l+\alpha}\right)^2 (1-q)^{l+1} \\
&\overset{(b))}{\leq} 2\tau^3 G^2 \sum_{l=0}^{T-1} (\bar{a}_i(T) + \bar{a}_{\text{ref}}(T)) \left(\frac{T+\alpha}{T-l+\alpha}\right)^2 (1-q)^{l+1} \\
&\leq 2\tau^3 G^2 (\bar{a}_i(T) + \bar{a}_{\text{ref}}(T)) \underbrace{\sum_{l=0}^{T-1} (l+1+\alpha)^2 (1-q)^{l+1}}_{S_T}
\end{aligned}
\tag{46}
$$

where (a) follows from $t = T - l$, (b) follows from the definition of $\bar{a}_i(t)$ and the non-decreasing nature of it.

Notice in (46), $S_T$ is non-decreasing in $T$, we can then use the limit of $S_T$ to bound it. The limit has a closed form,

$$\lim_{T \to \infty} S_T = \frac{(q-1)(-\alpha^2 q^2 - 2\alpha q + q - 2)}{q^3} . \tag{47}$$

Therefore, we bound the term $E_T$,

$$E_T \leq M A_{i,\text{ref}}(T) \tag{48}$$

where $M$ is a constant

$$M = 2\tau^3 G^2 \frac{(q-1)(-\alpha^2 q^2 - 2\alpha q + q - 2)}{q^3} \tag{49}$$

and

$$A_{i,\text{ref}}(T) = \bar{a}_i(T) + \bar{a}_{\text{ref}}(T) . \tag{50}$$

Similarly, we bound the term $D_T$ in (42),

$$
\begin{aligned}
D_T &\leq M \frac{1}{N} \sum_{m=1}^{N} (\bar{a}_m(T) + \bar{a}_{\text{ref}}(T)) \\
&= \frac{M}{N} \sum_{m=1}^{N} A_{m,\text{ref}}(T) .
\end{aligned}
\tag{51}
$$

Notice that we have $A_{i,\text{ref}}(T) = o\left(\frac{T^2}{\log T}\right)$ for all $i \in [N]$.

Now, we define $\Delta_t = \mathbb{E}\left[\|\boldsymbol{\theta}_{i,t} - \boldsymbol{\theta}^*\|^2\right]$, $Y_t = \beta^2(Q + MA_{i,\mathrm{ref}}(t) + \frac{M}{N}\sum_{m=1}^{N} A_{m,\mathrm{ref}}(t))$ and let $\beta = \frac{1}{2\mu\tau}$. From (42), we can write

$$\Delta_{t+1} \leq (1 - 2\mu\eta_t\tau)\Delta_t + \eta_T^2(Q + D_t + E_t)$$

$$\Delta_{t+1} \leq \left(1 - \frac{1}{\alpha + t}\right)\Delta_t + \frac{Y_t}{(\alpha + t)^2} \tag{52}$$

$$(\alpha + t)\Delta_{t+1} \leq (\alpha + t - 1)\Delta_t + \frac{Y_t}{\alpha + t} .$$

Let $\tilde{\Delta}_t = (\alpha + t - 1)\Delta_t$ where $\tilde{\Delta}_1 = \alpha\Delta_1$, we have

$$\tilde{\Delta}_{t+1} \leq \tilde{\Delta}_t + \frac{Y_t}{\alpha + t} . \tag{53}$$

Therefore,

$$\tilde{\Delta}_T \leq \alpha\Delta_1 + Y_T \sum_{l=1}^{T-1} \frac{1}{\alpha + l} \tag{54}$$

due to the non-decreasing nature of $A_{i,\mathrm{ref}}(t)$ with $t$.

Now,

$$
\begin{aligned}
\lim_{T \to \infty} \Delta_T &= \lim_{T \to \infty} \frac{\tilde{\Delta}_T}{T + \alpha - 1} \\
&\leq \lim_{T \to \infty} \frac{\alpha\Delta_1 + \beta^2(Q + MA_{i,\mathrm{ref}}(T) + \frac{M}{N}\sum_{m=1}^{N} A_{m,\mathrm{ref}}(T))\sum_{l=1}^{T-1} \frac{1}{\alpha + l}}{T} \\
&= \lim_{T \to \infty} \frac{A_{i,\mathrm{ref}}(T) + \frac{1}{N}\sum_{m=1}^{N} A_{m,\mathrm{ref}}(T)}{T} \lim_{T \to \infty} \frac{\sum_{l=1}^{T-1} \frac{1}{\alpha + l}}{T} \\
&= \lim_{T \to \infty} \frac{A_{i,\mathrm{ref}}(T) + \frac{1}{N}\sum_{m=1}^{N} A_{m,\mathrm{ref}}(T)}{\frac{T^2}{\log T}} \lim_{T \to \infty} \frac{\sum_{l=1}^{T-1} \frac{1}{\alpha + l}}{\log T} .
\end{aligned}
\tag{55}
$$

Note that $A_{i,\mathrm{ref}}(T) = o\left(\frac{T^2}{\log T}\right), \forall i \in [N]$, hence by definition

$$\lim_{T \to \infty} \frac{A_{i,\mathrm{ref}}(T) + \frac{1}{N}\sum_{m=1}^{N} A_{m,\mathrm{ref}}(T)}{\frac{T^2}{\log T}} = 0 . \tag{56}$$

Also, since $\sum_{l=1}^{T-1} \frac{1}{\alpha + l} = O(\log T)$,

$$\lim_{T \to \infty} \frac{\sum_{l=1}^{T-1} \frac{1}{\alpha + l}}{\log T} = Z \tag{57}$$

where $Z$ is a constant.

Therefore, substituting (56) and (57) into (55) to obtain $\lim_{T \to \infty} \Delta_T \leq 0$, and further considering that $\Delta_T \geq 0$,

$$\lim_{T \to \infty} \Delta_T = 0 . \tag{58}$$

From the $L$-smoothness of $F$, we conclude that

$$\lim_{T \to \infty} \mathbb{E}\left[F(\boldsymbol{\theta}_{i,T})\right] - F^* \leq \lim_{T \to \infty} \frac{L}{2}\Delta_T = 0 . \tag{59}$$

$\square$

# E ADDITIONAL EXPERIMENTS

## E.1 DATASETS AND IMPLEMENTATION DETAILS

We provide a comprehensive comparison between our method and the existing baselines using widely-used benchmark datasets, following a heterogeneous data partitioning benchmark (Li et al., 2022). All experiments were carried out on a server with Intel(R) Xeon(R)@2.70GHz processors and 1.5TB RAM. We utilized one Tesla V100 GPU for the experiments.

**Dataset preprocessing.** For all vision datasets, we standardized the pixel values of the images. We additionally applied data augmentation techniques of random cropping and random horizontal flipping to the CIFAR-10 and CIFAR-100 datasets. For natural language datasets, we performed standard tokenization and vocabulary building. We simulated the federated environment with 50 clients for all experiments. We employed five different heterogeneous partitioning strategies, namely: 1) Distribution-based label distribution skew, 2) quantity-based label distribution skew, 3) noise-based feature distribution skew, 4) quantity skew and 5) homogeneous partition. More detailed information on the implementation of these partitioning strategies can be found in Section 6.

**FL models**. To ensure the reproducibility of the experiments carried out in this paper, we used common and well-established model architectures. Specifically, we utilized the following model architectures:

1. Convolutional neural networks (CNN) following that of (Li et al., 2022). The CNN consists of 2 convolutional layers followed by 3 fully connected layers.

2. ResNet18 (He et al., 2016). We replaced the batch normalization layers in ResNet18 with group normalization to ensure stable training with highly heterogeneous clients (Wu & He, 2018; Hsieh et al., 2020; Wang et al., 2022). The number of groups used for the four layers of ResNet18 were 4, 8, 16 and 32, respectively.

3. Long short-term memory networks (LSTM). The network comprises an embedding layer of a dimension 300, an LSTM layer and 3 fully connected layers.

In Section 6.1, Appendix E.2, Appendix E.3 and Appendix E.4, we conducted fair evaluations using architectures 1 and 3. In Section 6.2, we utilized the more complex architecture 2 to demonstrate the ability of IAFL to achieve high accuracies on the challenging CIFAR-100 dataset. Architecture 2 was also employed in experiments conducted in Section 6.3 and Appendix E.5.

**Hyperparameters and options for training.** We used a default initial learning rate of $\eta_0 = 0.001$, unless otherwise specified. We used an exponential learning rate decay with a rate of 0.977, causing the learning rate to decrease by 10 folds in 100 iterations. For MNIST, we used $\eta_0 = 0.01$. For CIFAR-100 with ResNet18, we used $\eta_0 = 0.005$ with a decay rate of 0.988. The total number of FL training iterations used was 50 for MNIST, FMNIST, SVHN and 100 for CIFAR-10, CIFAR-100 and SST. Each FL iteration involved a client training for 1 local epoch. For all classification tasks, we employed the cross-entropy loss.

**FedAvg finetune.** Standard FedAvg returns the same model to all clients in each FL iteration. In this case, the correlation metric $\rho$ in Section 6.1 is undefined and uncomputable. To address this and create a personalized model for each client, we consider a simple modification that allows clients to train for an additional epoch at the end of the FL algorithm. Therefore, this variation, named FedAvg finetune, serves as a straightforward FL personalization baseline.

**LG-FedAvg (Liang et al., 2020).** Local global federated averaging (LG-FedAvg) achieves personalized models by personalizing the layers or structures of the shared model. Specifically, each client learns a personalized local feature extractor and the local representations of client data are federated to train shared global layers. To achieve this, the model updates averaged by the server are exclusively used to update the shared global layers. In our experiments, we specify the last 3 fully connected layers of the model architecture as the shared global layers.

**CGSV (Xu et al., 2021).** Cosine gradient Shapley value (CGSV) evaluates client contributions based on the cosine similarity between a client's gradients and the average of all clients' gradients. During the rewarding phase (i.e., when the server sends respective updates), CGSV masks out (i.e., zeroes out) portions of the gradient update aggregate according to the client contributions. This masking is

Table 4: Comparison of IPR$_{\text{loss}}$ among IAFL and baselines using differents datasets and partitions. Each value reports the mean and standard error of 10 independent evaluations and partition seedings.

| Category | Dataset | Partitioning | FedAvg Finetune | LG-FedAvg | CGSV | Rank | IAFL |
|----------|---------|--------------|-----------------|-----------|------|------|------|
| **Label Distribution Skew** | MNIST | $Dir(0.5)$ | 0.97±0.01 | 0.93±0.02 | **1.00±0.00** | 0.80±0.02 | **1.00±0.00** |
| | | $\#C=3$ | 0.89±0.01 | 0.90±0.02 | **1.00±0.00** | 0.68±0.02 | **1.00±0.00** |
| | FMNIST | $Dir(0.5)$ | **1.00±0.00** | 0.98±0.01 | **1.00±0.00** | 0.85±0.01 | **1.00±0.00** |
| | | $\#C=3$ | 0.92±0.01 | 0.95±0.01 | **1.00±0.00** | 0.81±0.02 | **1.00±0.00** |
| | SVHN | $Dir(0.5)$ | **1.00±0.00** | **1.00±0.00** | **1.00±0.00** | 0.95±0.01 | **1.00±0.00** |
| | | $\#C=3$ | **1.00±0.00** | **1.00±0.00** | **1.00±0.00** | 0.94±0.01 | **1.00±0.00** |
| | CIFAR-10 | $Dir(0.5)$ | 0.99±0.01 | **1.00±0.00** | **1.00±0.00** | 0.92±0.01 | **1.00±0.00** |
| | | $\#C=3$ | 0.91±0.01 | **1.00±0.00** | **1.00±0.00** | 0.85±0.02 | **1.00±0.00** |
| | CIFAR-100 | $Dir(0.5)$ | **1.00±0.00** | **1.00±0.00** | **1.00±0.00** | **1.00±0.00** | **1.00±0.00** |
| | | $\#C=30$ | **1.00±0.00** | **1.00±0.00** | **1.00±0.00** | 0.92±0.01 | **1.00±0.00** |
| | SST | $Dir(0.5)$ | **1.00±0.00** | 0.00±0.00 | 0.99±0.01 | 0.74±0.01 | **1.00±0.00** |
| | | $\#C=3$ | **1.00±0.00** | 0.00±0.00 | **1.00±0.00** | 0.78±0.02 | **1.00±0.00** |
| **Feature Distribution Skew** | MNIST | $\mathcal{N}(0.1)$ | 0.66±0.02 | 0.24±0.01 | 1.00±0.00 | 0.76±0.04 | **1.00±0.00** |
| | FMNIST | | **1.00±0.00** | 0.92±0.02 | **1.00±0.00** | 0.96±0.01 | **1.00±0.00** |
| | SVHN | | **1.00±0.00** | **1.00±0.00** | **1.00±0.00** | 0.97±0.00 | **1.00±0.00** |
| | CIFAR-10 | | **1.00±0.00** | **1.00±0.00** | **1.00±0.00** | **1.00±0.00** | **1.00±0.00** |
| | CIFAR-100 | | **1.00±0.00** | **1.00±0.00** | **1.00±0.00** | **1.00±0.00** | **1.00±0.00** |
| **Quantity Skew** | MNIST | $Dir(0.5)$ | 0.96±0.01 | 0.56±0.02 | **1.00±0.00** | 0.92±0.02 | **1.00±0.00** |
| | FMNIST | | **1.00±0.00** | 0.74±0.02 | **1.00±0.00** | 0.99±0.00 | **1.00±0.00** |
| | SVHN | | **1.00±0.00** | **1.00±0.00** | **1.00±0.00** | 0.99±0.00 | **1.00±0.00** |
| | CIFAR-10 | | 0.96±0.01 | **0.98±0.00** | 0.85±0.01 | 0.90±0.02 | **0.98±0.01** |
| | CIFAR-100 | | **1.00±0.00** | **1.00±0.00** | 0.99±0.00 | 0.99±0.00 | **1.00±0.00** |
| | SST | | **1.00±0.00** | 0.00±0.00 | **1.00±0.00** | 0.77±0.02 | **1.00±0.00** |
| **Homogeneous Partition** | MNIST | IID | 0.81±0.02 | 0.41±0.03 | **1.00±0.00** | 0.76±0.04 | **1.00±0.00** |
| | FMNIST | | **1.00±0.00** | 0.98±0.01 | **1.00±0.00** | **1.00±0.00** | **1.00±0.00** |
| | SVHN | | **1.00±0.00** | **1.00±0.00** | **1.00±0.00** | **1.00±0.00** | **1.00±0.00** |
| | CIFAR-10 | | **1.00±0.00** | **1.00±0.00** | **1.00±0.00** | **1.00±0.00** | **1.00±0.00** |
| | CIFAR-100 | | **1.00±0.00** | **1.00±0.00** | **1.00±0.00** | **1.00±0.00** | **1.00±0.00** |
| | SST | | **1.00±0.00** | 0.00±0.00 | **1.00±0.00** | 0.67±0.02 | **1.00±0.00** |

performed layer-wise, where the contribution determines the proportion of parameter values being masked out in each model layer's gradient updates. For our experiments, we utilized the original implementation of Xu et al. (2021).

**Rank (Kong et al., 2022).** Rank rewards clients in each iteration based on the accuracy of their respective models on a validation dataset. The method ranks the clients by their validation accuracies before rewarding them. Client $i$ can aggregate model updates from other clients that have a lower validation accuracy than that of client $i$. Since no code has been released for this work, we followed the original paper for our own implementation.

**IAFL.** The implementation follows that of Algorithm 1. For baseline comparisons in Section 4.1, we set the sharing parameter $\kappa = 0$ and stochastic recovery probability $q = 0$. Subsequently, we investigated the effects of these hyperparameters in Section 6.2. In all experiments except Section 6.3, we used the standalone accuracies of clients as their contributions $p_{i,t}$, which remain fixed across iterations. In Section 6.3, we instead used the participation rates of clients as the contribution measure.

## E.2 INCENTIVIZED PARTICIPATION RATE USING TEST LOSS

We expect IAFL to satisfy individual rationality (IR) if the mechanism strictly follows $[\mathcal{M}_\nu(\boldsymbol{p})]_i = \nu\left(p_i, (\min\{p_i/p_{\text{ceil}}, 1\})^{1-\kappa} h(\boldsymbol{p}_{-i})\right)$ and the rewarding function $\nu(p_i, P_{-i})$ is continuous, non-decreasing, concave and exchangeable with respect to both arguments. However, the assumptions on $\nu$ are often violated, as shown by the results of Table 1 in Section 6.1. Here we show that using the test loss as the rewarding function $\nu$ better fulfills the assumptions needed for $\nu$. The comparison of IPR$_{\text{loss}}$ (i.e., using the test loss) among IAFL and baselines is shown in Table 4. Our IAFL has excellent incentivization performance as it achieves the highest IPR$_{\text{loss}}$ in all settings and perfect IR in all but one setting.

Table 5: The average (highest) accuracies achieved by the client models. The values show the top-1 test accuracy measured in percentage (i.e., %). Each value reports the mean of 10 independent evaluations and partition seedings.

| Category | Dataset | Partitioning | FedAvg Finetune | LG-FedAvg | CGSV | Rank | IAFL |
|---|---|---|---|---|---|---|---|
| **Label Distribution Skew** | MNIST | $Dir(0.5)$ | 79 (95) | 86 (94) | 89 (93) | 71 (94) | **90 (96)** |
| | | $\#C = 3$ | 32 (46) | 47 (63) | 64 (72) | 38 (66) | **93 (96)** |
| | FMNIST | $Dir(0.5)$ | 60 (77) | 64 (75) | 71 (73) | 62 (80) | **74 (83)** |
| | | $\#C = 3$ | 30 (46) | 35 (46) | 47 (54) | 32 (47) | **71 (78)** |
| | SVHN | $Dir(0.5)$ | 57 (75) | **70** (77) | 59 (68) | 56 (75) | 67 **(79)** |
| | | $\#C = 3$ | 26 (41) | 35 (52) | 30 (44) | 25 (41) | **38 (69)** |
| | CIFAR-10 | $Dir(0.5)$ | 28 (39) | 30 (39) | 24 (30) | 29 (40) | **32 (45)** |
| | | $\#C = 3$ | 21 (24) | 22 (26) | 16 (23) | 22 (25) | **24 (36)** |
| | CIFAR-100 | $Dir(0.5)$ | 8 (11) | 11 (13) | 6 (6) | 10 (12) | **13 (17)** |
| | | $\#C = 30$ | 7 (9) | 8 (10) | 4 (5) | 8 (9) | **9 (13)** |
| | SST | $Dir(0.5)$ | 23 (33) | 23 (30) | 23 (29) | 24 (31) | **26 (34)** |
| | | $\#C = 3$ | 29 **(37)** | 25 (30) | 24 (30) | 27 (34) | **30** (37) |
| **Feature Distribution Skew** | MNIST | | 92 (96) | 95 **(97)** | **96** (96) | 91 (96) | 95 (96) |
| | FMNIST | | 79 (82) | 76 (79) | 75 (75) | 79 (83) | **84 (84)** |
| | SVHN | $\mathcal{N}(0.1)$ | 83 (85) | 79 (81) | 83 (83) | 82 (85) | **86 (86)** |
| | CIFAR-10 | | 52 (54) | 49 (51) | 42 (42) | 51 (54) | **56 (56)** |
| | CIFAR-100 | | 17 (18) | 16 (17) | 9 (9) | 15 (18) | **19 (20)** |
| **Quantity Skew** | MNIST | | 95 **(98)** | 93 (97) | 96 (96) | 91 (97) | **97** (97) |
| | FMNIST | | 82 (85) | 76 (83) | 75 (76) | 75 (83) | **85 (86)** |
| | SVHN | $Dir(0.5)$ | **84 (86)** | 77 (85) | 79 (84) | 60 (81) | **84 (86)** |
| | CIFAR-10 | | **53** (55) | 46 **(57)** | 41 (44) | 37 (50) | 52 (56) |
| | CIFAR-100 | | **18 (20)** | 15 **(20)** | 8 (9) | 8 (15) | 15 **(20)** |
| | SST | | 35 **(38)** | 25 (29) | 33 (35) | 29 (35) | **37 (38)** |
| **Homogeneous Partition** | MNIST | | 93 **(96)** | 94 **(96)** | **96 (96)** | 91 **(96)** | 95 **(96)** |
| | FMNIST | | 79 (82) | 78 (80) | 75 (75) | 79 (83) | **84 (84)** |
| | SVHN | IID | 84 (85) | 82 (83) | 83 (83) | 83 (85) | **86 (86)** |
| | CIFAR-10 | | 53 (55) | 51 (53) | 43 (43) | 52 (54) | **55 (56)** |
| | CIFAR-100 | | 17 (18) | 18 (18) | 9 (9) | 16 (18) | **19 (20)** |
| | SST | | 36 **(38)** | 25 (28) | 35 (36) | 30 (35) | **37 (38)** |

## E.3    AVERAGE AND HIGHEST TEST ACCURACY

It is important to achieve high overall performance for client models as obtaining good models serves as a strong motivation for clients to actively participate in the federated learning process. To assess the effectiveness of IAFL and baseline methods in this regard, we compare the average and highest client model test accuracies. As presented in Table 5, our IAFL consistently outperforms the other baselines in almost all settings. These results indicate that IAFL successfully incentivizes clients to actively contribute and enables them to benefit from the collaborative process. The higher highest accuracies achieved by IAFL further demonstrated its ability to effectively leverage the decentralized data of clients to achieve superior predictive performance. We further note that the low accuracies obtained for CIFAR-100 can be attributed to the CNN model architecture used in this experiment. Further results in Section 6.2 show that IAFL achieves competitive accuracies on CIFAR-100 using a more complex ResNet18 model.

## E.4    AVERAGE INCREASE IN ACCURACIES

Following (Pillutla et al., 2022), we compare the average increase in client model performance measured by test accuracy. Table 6 demonstrates that IAFL generally yields the most significant improvements in test accuracies across all clients. Intuitively speaking, such significant increases in test accuracies before and after the adoption of federated collaboration via IAFL serve as strong incentives for clients to actively participate and use the IAFL algorithm. These findings further complement the earlier experiments on $IPR_{accu}$ and $IPR_{loss}$, which only provided a percentage for binary outcomes (i.e., whether a client model improved after collaboration or not). Through this additional experiment, we have clearly shown the extent of client model improvements on the individual standalone models that our IAFL brings.

Table 6: Average accuracy increase of clients. The unit for this table is %, and we report the mean of 10 independent evaluations.

| Category | Dataset | Partitioning | FedAvg Finetune | LG-FedAvg | CGSV | Rank | IAFL |
|---|---|---|---|---|---|---|---|
| **Label Distribution Skew** | MNIST | $Dir(0.5)$ | 8.6±0.5 | 15.0±0.7 | 18.2±0.6 | 0.9±4.5 | **19.1±0.7** |
| | | #C = 3 | 2.7±0.5 | 17.2±0.9 | 34.0±2.4 | 8.4±0.3 | **63.2±0.4** |
| | FMNIST | $Dir(0.5)$ | 3.0±0.5 | 7.2±0.3 | 14.2±0.5 | 5.5±0.4 | **17.5±0.4** |
| | | #C = 3 | 1.9±0.3 | 6.8±0.4 | 18.3±1.5 | 3.5±0.4 | **42.7±1.1** |
| | SVHN | $Dir(0.5)$ | 9.2±0.5 | **21.5±0.6** | 10.8±0.7 | 7.8±0.6 | 18.6±0.5 |
| | | #C = 3 | -0.3±0.1 | 7.9±0.5 | 3.0±2.8 | -1.2±0.2 | **11.1±0.9** |
| | CIFAR-10 | $Dir(0.5)$ | -1.7±0.9 | 0.1±0.9 | -5.7±0.9 | -1.0±1.0 | **2.5±0.9** |
| | | #C = 3 | -3.3±0.1 | -2.0±0.1 | -8.0±0.3 | -2.1±0.1 | **-0.4±0.4** |
| | CIFAR-100 | $Dir(0.5)$ | -0.2±0.1 | 2.8±0.3 | -2.6±0.1 | 1.0±0.1 | **4.5±0.2** |
| | | #C = 30 | -1.3±0.1 | -0.9±0.1 | -4.2±0.2 | -0.2±0.1 | **0.7±0.3** |
| | SST | $Dir(0.5)$ | -0.0±0.3 | 0.3±0.0 | -0.0±0.3 | 1.1±0.1 | **3.3±0.3** |
| | | #C = 3 | 4.6±0.2 | 0.2±0.0 | -1.0±0.4 | 2.0±0.2 | **5.1±0.3** |
| **Feature Distribution Skew** | MNIST | | -1.9±0.5 | 1.0±0.5 | **2.2±0.4** | -3.3±0.6 | 1.4±0.4 |
| | FMNIST | | 0.6±0.2 | -2.6±0.4 | -3.6±0.3 | 1.1±0.2 | **5.9±0.1** |
| | SVHN | $\mathcal{N}(0.1)$ | 10.8±0.3 | 6.8±0.4 | 10.7±0.5 | 9.1±0.5 | **13.1±0.5** |
| | CIFAR-10 | | 8.9±0.3 | 5.4±0.3 | -1.6±0.4 | 7.8±0.2 | **12.3±0.2** |
| | CIFAR-100 | | 8.6±0.2 | 7.3±0.2 | -0.1±0.2 | 6.1±0.1 | **10.2±0.2** |
| **Quantity Skew** | MNIST | | 10.1±0.8 | 7.7±0.5 | 10.8±0.6 | 6.3±0.4 | **11.3±0.6** |
| | FMNIST | | 10.6±0.6 | 5.2±0.4 | 4.1±0.6 | 3.7±0.4 | **13.5±0.5** |
| | SVHN | $Dir(0.5)$ | 26.9±1.0 | 19.3±0.8 | 21.6±0.9 | 2.9±1.0 | **27.1±0.9** |
| | CIFAR-10 | | **15.8±0.3** | 8.7±0.3 | 4.1±0.3 | -0.0±0.4 | 15.6±0.3 |
| | CIFAR-100 | | **10.7±0.1** | 7.8±0.1 | 0.8±0.1 | 0.4±0.2 | 7.5±0.2 |
| | SST | | 10.7±0.1 | 0.2±0.2 | 8.6±0.2 | 4.2±0.2 | **11.8±0.2** |
| **Homogeneous Partition** | MNIST | | -1.2±0.4 | 0.5±0.4 | **2.3±0.4** | -2.9±0.5 | 1.5±0.4 |
| | FMNIST | | 0.3±0.2 | -0.7±0.1 | -3.6±0.2 | 0.9±0.2 | **5.2±0.2** |
| | SVHN | IID | 10.9±0.5 | 9.2±0.5 | 10.2±0.4 | 10.2±0.5 | **13.3±0.3** |
| | CIFAR-10 | | 9.1±0.3 | 7.4±0.4 | -1.3±0.4 | 7.8±0.5 | **11.4±0.4** |
| | CIFAR-100 | | 8.3±0.2 | 8.5±0.1 | 0.3±0.2 | 7.2±0.2 | **10.0±0.1** |
| | SST | | 10.3±0.2 | -0.1±0.2 | 9.6±0.3 | 4.4±0.2 | **11.6±0.2** |

## E.5 CHANGING THE REFERENCE MODEL

IAFL is flexible with the reference model definition, as long as the reference model at the server is also updated using a portion of the client model updates in each FL iteration, similar to any other federated client. The reference model, of course, does not contribute any learning resources (e.g., data). The proportion of client model updates used by the reference model is determined by the user-defined custom induced reference reward rate $\gamma'_{\text{ref},t}$.

In the experiments of Section 6.2, we set the induced reference reward rate as $\gamma'_{\text{ref},t} = \max_{i\in[N]} \gamma'_{i,t}$. In other words, the induced reference reward rate is equal to the highest induced reward rate among all clients in a given iteration $t$. In this section, we explore the possibility of changing the reference model to a median reward, i.e., $\gamma'_{\text{ref},t} = median_{i\in[N]}\gamma'_{i,t}$. Intuitively, the clients are now stochastically recovered with a "roughly median" model among all clients in any given iteration. The results are shown in Figure 4 and should be viewed in comparison to Figure 1, where the highest induced reference reward rates are used to update the reference models. Overall, the incentivization performances are comparable under the two reference model definitions. We observe that lower

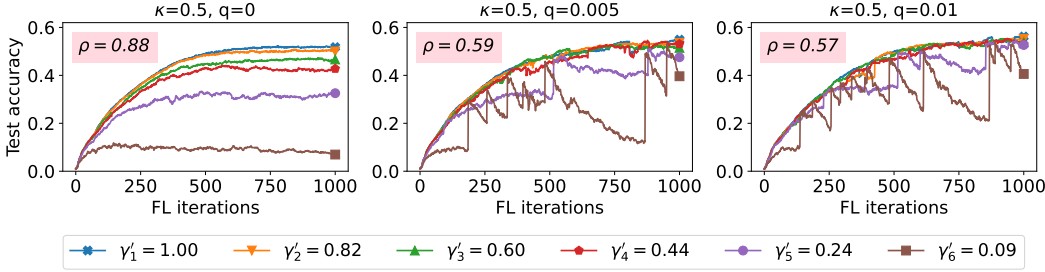

Figure 4: Effects of the stochastic recovery rate $q$ with a fixed sharing coefficient and a reference model trained with median induced reward rates. The dataset is CIFAR-100 with quantity skew and 50 clients. The figures should be viewed in comparison to Figure 2.

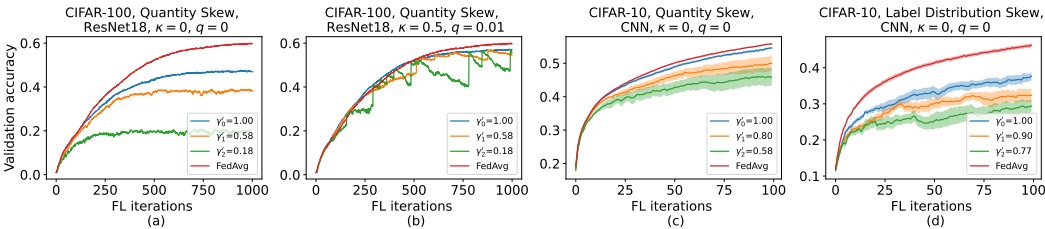

Figure 5: Comparing IAFL and FedAvg for the speed of convergence. Generally, clients with different levels of contribution induce different reward rates $\gamma_i'$ and result in different speeds for convergence. Notably, the stochastic reference model recovery is vital to the convergence to the global optimum.

induced reference reward rates $\gamma_{\mathrm{ref},t}'$ may hinder convergence to the global optimum and affect the overall average accuracy. However, lower $\gamma_{\mathrm{ref},t}'$ values have the advantage of better incentivizing top-contributors because it becomes more difficult for the lower-contributing clients to obtain the highest-performing models within a limited number of iterations. Therefore, the choice of the reference model definition becomes an important hyperparameter of IAFL, as it determines the extent of sharing and collaboration among clients. In practical settings where clients are more altruistic in their contributions (e.g., collaboration among non-profit organizations or government agencies), having a high $\gamma_{\mathrm{ref},t}'$ value is recommended.

### E.6 EMPIRICAL CONVERGENCE

In this section, we investigate the effect of IAFL on the empirical convergence speed of different clients contributing at different levels. Figure 5 illustrates the empirical convergence curves of selected representative clients. Based on Figure 5(a), when $q = 0$ (i.e., the stochastic reference model recovery is deactivated), even the client with the highest induced reward rate $\gamma_i'$ experiences slowed convergence, eventually to a suboptimal model. Therefore, the proposed stochastic reference model recovery strategy is essential to the convergence to the global optimum. Upon the activation of the stochastic recovery at $q = 0.01$, as shown in Figure 5(b), we observe that the convergence rates of client models trained using IAFL are comparable to FedAvg. In Figure 5(c) and (d), we observe a similar behavior to that in Figure 5(a) using different datasets and data partitions: The empirical convergence is slower than FedAvg as expected when stochastic recovery is deactivated.

### E.7 PRACTICAL TRADE-OFF BETWEEN PERFORMANCE AND INCENTIVIZATION

In Section 6.2, we discuss the controllable hyperparameters of the IAFL algorithm and their effects on the practical trade-off between model performance and our goal of client incentivization. Generally, larger values of $\kappa$, $q$ and setting $\gamma_{\mathrm{ref},\,t}'$ closer to $\max_{i \in [N]} \gamma_{i,t}'$ favor overall client model performances (at the same time promoting equality among clients) while trading-off client incentivization (i.e., compromising collaborative fairness and distributing less distinctive client models). In this paper, we have presented recommended values for easy usage that strike a good balance between these goals. The following set of values

- $\kappa = 0.5$ ,
- $q = 0.01$ ,
- $\gamma_{\mathrm{ref},\,t}' = \max_{i \in [N]} \gamma_{i,t}'$ ,

generally worked well in our experiments. Consequently, we recommend using them as the default configuration.

If practitioners wish to achieve a different degree of trade-off with our IAFL framework, they could calibrate these hyperparameters to favor either client incentivization or model performance, from different perspectives depending on the hyperparameter. Our proposed IAFL algorithm provides a high degree of flexibility for the server and clients (e.g., users, practitioners) to adjust the hyperparameters such that they align with users' perceptions and preferences. However, we highlight that this flexibility is not without constraint: It must trade off between fairness and equality, implying that there

are certain scenarios infeasible for accommodation. For example, the '*winner takes all*' perspective falls outside the spectrum of our trade-off as it violates both fairness and equality. Similarly, it is also unrealistic to expect to achieve both fairness and equality together in the IAFL's training-time model rewards distributed to the clients.

## E.8 ALTERNATIVE CHOICES OF THE CONTRIBUTION MEASURE

As discussed in Section 4.2, IAFL as an incentive mechanism can be applied to a vast range of contribution measures. In this section, we adopt the cosine gradient Shapley value (CGSV) discussed in Section 2 as the contribution measure to determine $p_{i,t}$ and conduct IAFL training. We denote this specific method as CGSV-IAFL and compare it with the original CGSV implementation with gradient masking (Xu et al., 2021). Table 7 shows the superior incentivization performance of CGSV-IAFL across various data partitioning strategies on CIFAR-10. Therefore, IAFL works with alternative choices of contribution measures. Additionally, we demonstrate through this experiment that our rewarding scheme detailed in Procedure 2 is more effective than the gradient masking technique of the original CGSV.

Table 7: Comparison of the incentivization performance between CGSV and IAFL after adopting a common contribution measure defined by CGSV. The incentivization performance is measured using the Pearson correlation coefficient $\rho$ between the final client model accuracies and standalone accuracies. Each value reports the mean and the standard error over 10 independent evaluations.

| Category | Partitioning | CGSV | CGSV-IAFL |
|---|---|---|---|
| **Label** 
 **Distribution Skew** | $Dir(0.5)$ 
 $\#C = 3$ | 0.46±0.04 
 0.31±0.05 | **0.69±0.03** 
 **0.62±0.03** |
| **Feature Distribution Skew** | $\mathcal{N}(0.1)$ | -0.02±0.04 | **0.06±0.05** |
| **Quantity Skew** | $Dir(0.5)$ | 0.73±0.04 | **0.81±0.01** |
| **Homogeneous Partition** | IID | -0.03±0.05 | **-0.03±0.04** |

## E.9 EXPERIMENTS ON COMPLEX LARGE-SCALE DATASETS

We additionally conduct experiments on complex large-scale datasets: Tiny-ImageNet for vision tasks and Sent140 for language tasks. The results are shown in Table 8.

Table 8: The incentivization performance under different dataset partitions, measured using the Pearson correlation coefficient $\rho$ between the final client model accuracies and standalone accuracies. Each value reports the mean and the standard error over 3 independent evaluations.

| Category | Dataset | Partitioning | FedAvg Finetune | LG-FedAvg | CGSV | Rank | IAFL |
|---|---|---|---|---|---|---|---|
| **Label** 
 **Distribution** 
 **Skew** | Tiny-ImageNet 

 Sent140 | $Dir(0.5)$ 
 $\#C = 60$ 
 $Dir(0.5)$ 
 $\#C = 1$ | 0.27±0.06 
 0.38±0.06 
 0.93±0.01 
 **1.00±0.00** | 0.10±0.07 
 0.37±0.11 
 **0.99±0.00** 
 **1.00±0.00** | -0.02±0.00 
 0.10±0.08 
 0.23±0.32 
 **1.00±0.00** | 0.34±0.11 
 0.22±0.10 
 0.98±0.00 
 **1.00±0.00** | **0.81±0.07** 
 **0.82±0.01** 
 0.84±0.03 
 0.07±0.07 |
| **Feature** 
 **Distribution Skew** | Tiny-ImageNet | $\mathcal{N}(0.1)$ | undefined | -0.04±0.07 | -0.04±0.10 | 0.04±0.10 | **0.77±0.06** |
| **Quantity** 
 **Skew** | Tiny-ImageNet 
 Sent140 | $Dir(0.5)$ | undefined 
 -0.74±0.04 | **0.95±0.00** 
 **1.00±0.00** | 0.38±0.11 
 0.80±0.02 | 0.85±0.02 
 0.99±0.00 | 0.83±0.02 
 0.90±0.01 |
| **Homogeneous** 
 **Partition** | Tiny-ImageNet 
 Sent140 | IID | undefined 
 0.08±0.03 | -0.02±0.06 
 **0.27±0.02** | 0.07±0.10 
 -0.20±0.08 | 0.02±0.01 
 0.06±0.04 | **0.78±0.08** 
 -0.06±0.11 |
| Number of times that performs the best | | | 1 | 5 | 1 | 1 | 4 |

For the complex Tiny-ImageNet dataset, IAFL achieves the best incentivization performance in the majority of the data partition settings. The correlations $\rho$ between the final client model accuracies and standalone accuracies are generally above $0.75$. Note that finetuning the network trained by FedAvg with the local client dataset for a small number of epochs may not affect the performance of the networks. Therefore, in some cases, the final models received by the clients have the same accuracy (as the global FedAvg model) and thus the correlation $\rho$ is undefined.

For the Sent140 dataset, we notice that other methods such as FedAvg Finetune and LG-FedAvg outperform IAFL in terms of incentivization performance measured by the correlation $\rho$ in many

cases. Further investigation reveals that using $\rho$ alone to assess the effectiveness of collaborative learning is insufficient. Comparing the $\text{IPR}_{\text{loss}}$ in Table 9, the high $\rho$ of baselines on Sent140 $\#C = 1$ corresponds to $0\%$ client incentivization measured by $\text{IPR}_{\text{loss}}$. This is because collaboration through baseline methods does not result in any improvement of the client's model. We further provide the results of the average (and highest) accuracies achieved by clients on Sent140 in Table 10. The results illustrate that IAFL rewards better models to clients as compared to other baselines. Overall, IAFL achieves the best incentivization performance considering both metrics $\rho$ and $\text{IPR}_{\text{loss}}$ while outputting client models with the highest model accuracies.

Table 9: Comparison of $\text{IPR}_{\text{loss}}$ among IAFL and baselines on Sent140. Each value reports the mean and standard error of 3 independent evaluations and partition seedings.

| Category | Partitioning | FedAvg Finetune | LG-FedAvg | CGSV | Rank | IAFL |
|---|---|---|---|---|---|---|
| Label
Distribution Skew | $Dir(0.5)$
$\#C = 1$ | 0.84±0.02
0.00±0.00 | 0.01±0.01
0.00±0.00 | **0.93±0.07**
0.00±0.00 | 0.62±0.00
0.00±0.00 | 0.87±0.02
**0.83±0.04** |
| Quantity Skew | $Dir(0.5)$ | **1.00±0.00** | 0.00±0.00 | **1.00±0.00** | 0.91±0.01 | **1.00±0.00** |
| Homogeneous Partition | IID | **1.00±0.00** | 0.00±0.00 | **1.00±0.00** | 0.71±0.01 | **1.00±0.00** |

Table 10: The average (highest) accuracies achieved by the client models on Sent140. The values show the top-1 test accuracy measured in percentage (i.e., %). Each value reports the mean of 3 independent evaluations and partition seedings.

| Category | Partitioning | FedAvg Finetune | LG-FedAvg | CGSV | Rank | IAFL |
|---|---|---|---|---|---|---|
| Label
Distribution Skew | $Dir(0.5)$
$\#C = 1$ | 68 (83)
50 (50) | 61 (77)
50 (50) | 61 (72)
50 (50) | 66 (83)
50 (50) | **71 (84)**
**56 (70)** |
| Quantity Skew | $Dir(0.5)$ | **85 (85)** | 72 (80) | 79 (81) | 76 (**85**) | **85 (85)** |
| Homogeneous Partition | IID | 84 (**85**) | 75 (76) | 81 (81) | 81 (83) | **85 (85)** |

# F  FREQUENTLY ASKED QUESTIONS AND DISCUSSIONS

**Question 1:** Is it more natural to incentivize the clients to contribute using their full capacity?

**Answer:** While having clients contribute to their full capacities for the best learning outcome is the ideal scenario, this goal is unrealistic because every client incurs non-trivial costs to contribute. Importantly, the actual contribution of clients depends on the interplay between rewards and costs for different contribution levels.

In our work, we propose the next achievable alternative: To incorporate the goal of "*incentivizing clients to contribute as much as possible*" on top of satisfying individual rationality. To this end, we designed IAFL such that Theorem 1 and 2 are fulfilled. These theorems suggest that clients will be rewarded with better models if they contribute more. This will incentivize clients to further contribute to their full capacity for improved model performance received, of course, subject to their marginal utility increment in the presence of a cost $c_i$. Clients will contribute up till the marginal increment in reward still surpasses the associated cost.

**Question 2:** What if a client does not faithfully compute and upload the full local model updates?

**Answer:** Our framework does not place any assumption on what kind of local model updates are computed and uploaded by a client. Instead, the client can freely decide whether he or she wants to make "full" or "zero" contribution. The contribution measures (in Section 4.2) capture the quality of the local model updates as contributions, e.g., FedSV (Wang et al., 2020b), ComFedSV (Fan et al., 2022), CGSV (Xu et al., 2021), FedFAIM (Shi et al., 2022) are examples of such contribution measures. We summarise the whole process here: The client's behavior model decides the quality of the local model updates being computed (e.g., using different portions of its local data). Then, the quality of the local model updates affects the client contribution level (as assessed by the contribution evaluation measures mentioned above), which in turn affects the reward rate in IAFL. Subsequently, the reward rates of clients affect their model convergence.

**Question 3:** Does convergence speed become faster as the number of agents $N$ grows?

**Answer:** The simple answer is, adding clients that "help than hurt" is going to improve the convergence speed.

To elaborate, the effect depends on the contribution levels of the clients. We can look at the term $C_T = \frac{H_T}{N} \sum_{m=1}^{N} \sum_{t=1}^{T} \left(\frac{T+\alpha}{t+\alpha}\right)^2 \left(\frac{1}{\gamma'_{m,t}} + \frac{1}{\gamma'_{i,t}} - 2\right)$, when $N$ increases, the denominator becomes bigger but at the same time there are additional terms in the summation.

We can consider a case where we increase $N = n$ to $N = n + 1$. Abstracting away the constants $H_T$, $T$ and $\alpha$ for simplicity, we can make a comparison for $C_T = \frac{1}{n} \sum_{m=1}^{n} \left(\frac{1}{\gamma'_{m,t}} + \frac{1}{\gamma'_{i,t}} - 2\right)$ when $N = n$ against the case for $C_T = \frac{1}{n+1} \sum_{m=1}^{n+1} \left(\frac{1}{\gamma'_{m,t}} + \frac{1}{\gamma'_{i,t}} - 2\right)$ when $N = n + 1$. Note that $\gamma'_{i,t}$ can also be treated as a constant here. Informally speaking, we observe that when $N$ increases, the term $C_T$ generally decreases in its scale if the added client has contribution $\gamma'_{m,t}$ at least the "average contribution" of the existing clients $\frac{1}{n} \sum_{m=1}^{n} \gamma'_{m,t}$. Therefore, the decrease in $C_T$ tightens the performance bound in Theorem 1 and makes the convergence faster.

An exception that we can derive from the above is that when the added clients contribute badly with a very low $\gamma'_{m,t}$, it could slow down the convergence despite the increase in $N$. In practice, this implies that the model update provided by the added client is bad, harmful, or even adversarial, and is detected by the contribution evaluation measure. Theoretically, the convergence is adversely affected if we add "harmful" clients. However, we could easily implement client selection methods based on this insight on contribution evaluation to filter such clients during the training process to ensure faster convergence when $N$ grows.

**Question 4:** Is it fair that for small clients, even if they contribute to their best, they still cannot receive a high-quality model? Yet the large clients only need to contribute above some threshold to receive the best model?

**Answer:** There are two prevalent concepts of fairness in literature: (a) collaborative fairness (Lyu et al., 2020) (i.e., contribute more get back more), and (b) equality (Li et al., 2020c). To be fair, people from one of the camps will not view the other as fair.

In our paper, we have demonstrated results for the collaborative fairness view. Specifically, the contributions of clients are fairly evaluated by a contribution evaluation measure, e.g., CGSV (Xu et al., 2021), and then used to fairly reward the clients with a model that has performance commensurate with their contributions.

If the designer of the FL process views fairness more towards equality such that "if everyone puts in their best efforts, they should be treated equally", it is possible to use "individual efforts percentage" (e.g., how many percentages a client contributes to his/her best effort) as a contribution evaluation measure (assuming that individual efforts of clients can be measured). This measure can be seamlessly combined with the IAFL framework, too. However, we would like to clarify that "individual efforts percentage" is not commonly used as a contribution measure in literature.

