# OpenReview forum: "Incentive-Aware Federated Learning with Training-Time Model Rewards"
_ICLR.cc/2024/Conference — ICLR 2024 poster_

### Official Review · Reviewer_EksS · 2023-10-31

**Soundness:** 2 fair
**Presentation:** 2 fair
**Contribution:** 2 fair
**Rating:** 6
**Confidence:** 2

**Summary:**

The paper proposes Incentive-Aware Federated Learning (IAFL), an FL algorithm which is generally applicable to varying measures of client contribution such as participation rate or local update steps. To incentive client contribution, IAFL takes a personalized FL approach where the server shares higher-quality model updates with clients with higher contribution. Additionally, the paper ensures that all clients, despite limited contribution, are able to reach the optimal model by stochastically synchronizing client models with a common reference model. The paper shows that IAFL outperforms various FL baselines in terms of IPR in several heterogeneous settings.

**Strengths:**

This work appears novel in the sense that it personalizes the outcome of each round to individual clients, whereas earlier approaches attempt to produce a single global model that is compatible with multiple clients' incentives.

As mentioned in the paper, it is applicable to settings where earlier incentive-aware FL works are not, such as partial participation and lack of server-side data.

**Weaknesses:**

The behavior of IAFL is not clearly explained in the experiments.
- What is client contribution here? Is it number of local updates? How is this set / varied across clients / time?
- How is incentivization determined? Do you compare a locally trained model to the (fully trained) server model?

6.1: " We measure the performance of a model using the test loss and the test accuracy, denoting them as IPR_loss and IPR_accu, respectively." This doesn't make sense to me. Doesn't IPR_accu (Table 1) refer to the fraction of clients who are "incentived" to participate, and not an accuracy metric?

Based on the results in Table 3 it is surprising to see that IAFL achieves much better accuracy than non-IR methods. However, shouldn't the other methods have an advantage when comparing raw accuracy, as they distribute a high-quality model to all clients without considering incentives? What exactly does this accuracy metric refer to?

**Questions:**

Reading through this paper, I assumed that client contribution is not being adjusted in response to the rewards. Please clarify if this is inaccurate.

Assuming contribution is participation rate, wouldn't there already be a disadvantage to partial participation if the server broadcasts an update (rather than the updated model) to the participating clients, as the local model could become desynchronized? Or does the paper assume the server is sending an updated model?

---

> ### Author Response · Authors · 2023-11-14
> **Thank you for the reviews (1/2)**
>
> We thank the reviewer for taking the time to review our paper and for recognizing our novel work that personalizes client model outcomes that are compatible with client incentives.
>
> We earnestly hope addressing the concerns as follows will improve your opinion of our work.
>
> >W1(a): "What is client contribution here" and "How is this set/varied across clients/time?"
>
> To clarify the experimental settings, as described in Section 6.1, one way to measure client contribution is to "use standalone accuracies as a surrogate for the client contributions in FL". In this way, the client contributions vary across clients.
>
> As an alternative way to measure client contribution, we also adopted CGSV (Xu et al., 2021) as the contribution measure in Appendix E.8. For CGSV, it not only considers variances of contribution values across clients, but also takes into account the changes occurring over time or iteration steps.
>
> >W1(b): "How is incentivization determined? Do you compare a locally trained model to the (fully trained) server model?"
>
> To clarify the metrics that we use to evaluate incentivization performance, we carefully utilize three distinct metrics covering different aspects of desired behaviors of IAFL in experiments. They are (1) incentivized participation rate (IPR), (2) correlation to contribution $\rho$, and (3) average/highest predictive performance. These three metrics are described in detail and thoroughly studied in Section 6.1. We do not compare to the fully trained server model because IAFL does not output any server global model.
>
> As for the (fully trained) server model, it actually corresponds to the highest predictive performance (point 3 above). The tables of results are pushed to Table 5 of Appendix E.3 and Table 10 of Appendix E.9 due to space constraints. This is because the best client in our experimental setting receives the best model updates aggregated from all clients in all training iterations, therefore, the highest predictive performance corresponds to the (fully trained) server model.
>
> >W2: About "IPR$_\text{accu}$"
>
> we apologize for the confusion in the quoted sentence. We need to associate the sentence with the two sentences before it and hence refer the reviewer to the rephrased version here:
>
> "We let IPR$\_\text{loss}$ be the percentage of clients receiving a model not worse than his standalone model in terms of the test loss. Likewise, We define IPR$\_\text{accu}$ but in terms of the test accuracy. Here, a standalone model refers to the model trained only with the client's local dataset. IPR therefore indicates IR (Definition 1)."

---

> ### Author Response · Authors · 2023-11-14
> **Thank you for the reviews (2/2)**
>
> >W3: About Table 3, "better accuracy than non-IR methods"
>
> The “acc.” metric in Table 3 refers to “Predictive performance” explained in the corresponding paragraph in Section 6.1.
>
> Now, we explain the results in Table 3. Other baseline methods yield low accuracies as they neglect the individual rationality (IR) of clients in their design in achieving fairness and personalization goals. As seen in the table below,
>
> | Methods  | IPR$\_\text{accu}$ | acc. (highest) |
> | -------- | --------- | ------- |
> | FedAvg Finetune | 0.28 | 0.12 (0.16) |
> | LG\-FedAvg      | 0.60 | 0.13 (0.15) |
> | CGSV            | 0.00 | 0.06 (0.07) |
> | Rank            | 0.94 | 0.24 (0.31) |
> | IAFL            | 1.00 | 0.44 (0.53) |
>
> the incentivized participation rates (IPR) achieved by baseline methods are low, indicating that the clients did not receive high-quality models (even lower than what they can achieve on their own), reflected as poor model performance in terms of predictive accuracies.
>
> >Q1: "I assumed that client contribution is not being adjusted in response to the rewards"
>
> Generally speaking, in the same iteration, the rewards are adjusted in response to the current client contributions. However, provided that the contribution evaluation measure uses models at previous iterations, it is possible that iterative contribution measurements are affected by previous reward models. For example, when using CGSV (Xu et al., 2021), since CGSV measures contribution iteratively, the reward model $\theta_{i,t}$ of iteration $t$ could affect the gradient updates and thus contributions $p_{i,t+1}$ of the next iteration $t+1$.
>
> >Q2: "partial participation" $\rightarrow$ "disadvantageous" since "the local model could become desynchronized?"
>
> To clarify, we assume that clients are always available to receive **model updates** of varying quality commensurate to their contributions (not an updated model) from the server in all training iterations, and the participation rate only reflects the client's rate of computing and sending the local updates to the server. Therefore, the client model quality varies mainly because of the quality of the updates the server sends to the client depending on the client's contribution, rather than issues caused by synchronization.
>
> In the event that it is impractical to assume the client's availability to receive model updates in every iteration, we can alternatively keep a database on the server side to store the cumulative model updates for clients until they become available.
>
> >To conclude
>
> We hope that we have clarified the questions about the experimental setups, better explained the empirical results, and sufficiently addressed your concerns. We hope that we could improve your opinion about our paper.

---

> ### Author Response · Authors · 2023-11-22
> **Follow-up**
>
> Dear Reviewer EksS,
>
> Thank you again for your time in reviewing our paper and for your particular interest in our experimental setups and empirical results.
>
> Please let us know whether our replies have sufficiently addressed your concerns. We will be happy to engage further within the discussion period.
>
> Best regards,
>
> Authors of Paper 5535

---

> ### Author Response · Authors · 2023-11-23
> **A summary and kind reminder**
>
> Dear Reviewer EksS,
>
> Thank you again for your time in reviewing our paper. In the rebuttal, we have made the following major clarifications to your questions:
>
> - We clarified the determination of client contributions and metrics for client incentivization in the experimental setting that aligns with your suggestion.
> - We rephrased the defition of IPR$_\text{accu}$ to clear the unintended confusion.
> - We explained for the results in Table 3.
> - We clarified our experimental settings for the case of partial participation.
>
>
> As today is the last day of the discussion period, please let us know whether our replies have sufficiently addressed your concerns. We will be happy to engage further within the discussion period.
>
> Best regards,
>
> Authors of Paper 5535

---

### Official Review · Reviewer_xzxw · 2023-11-01

**Soundness:** 3 good
**Presentation:** 3 good
**Contribution:** 3 good
**Rating:** 6
**Confidence:** 3

**Summary:**

In federated learning, each client contributes the gradient updates computed with its own local data and then shared with the center. The center aggregates the updates from clients to update the model, and then share the model to the clients to start the next round. In this process, selfish clients may get the most up-to-date model by free-riding, and hence hurt the overall performance of the system. Prior work proposes to incentivize the clients with monetary transfers, while this work focuses on designing an incentive mechanism to share the model in a way such that the more a client contributes, the better model it will receive.

The proposed mechanism includes two key features to incentive each client to contribute more:
* Sharing the model updates from a subset of the other clients, the size of the subset is proportional to this client’s contribution;
* With some probability, give the client the most up-to-date model to prevent the client’s local model being too off.

Theoretical results:
* All clients have strictly positive incentive to contribute more
* Each client is better off to participate in the federated learning (individually rational)
* The bound on the performance loss of the client models (against the optimal benchmark not suffering from any free-ride challenge), which converges to zero with additional assumptions.

Experiments:
* Partition training data to simulate the distributed data in the federated learning setting
* Evaluate the percentage of clients where the IR condition is respected
* Evaluate how the hyperparameters influence the performance of client models at different contribution levels.

**Strengths:**

* Very interesting idea to an important problem
* Solid results

**Weaknesses:**

* The incentive guarantee is weak in the sense that only a positive incentive is guaranteed, which might not be enough when the clients do suffer certain costs to contribute more to the center. When the cost is higher than the incentive, one may still not contribute 100% effort in the federated learning.
* The tradeoff between the strength of incentive and the loss of (center) model accuracy is not established, which might be more important in practice. In particular, if I understand correctly, this work assumes all clients contribute 100% of its effort given the constructed incentives. Hence the performance loss of the center model is considered as zero and not measured. However, I can see at least two reasons for the clients to not contribute at its full capacity:
  * Contribution already exceeds the threshold parameter $p_\mathsf{ceil}$
  * When maintaining a certain level of incentive is necessary, one may have to limit the sharing of the gradients, i.e., sufficiently low $\kappa$ and $q$. In this case, the performance loss of the center model should emerge as the client model might be quite off from the center model and lead to low quality of the gradient updates from local models.
* I would suggest the authors to at least discuss the above limitations

**Questions:**

* What is the tradeoff between the strength of incentive and the loss of (center) model accuracy?
* Is there a fairness concern that for small clients, even if they contribute to their best, they still cannot receive a high quality model? Yet the large clients only need to contribute above some threshold to receive the best model?
* It seems to me that the small clients may have incentive to cooperate to pretend as one big client to receive a better model without a significant cost overhead? Will this lead to exchange platforms where one can first send their gradient updates to the platform, then the platform aggregates the gradient updates from many small clients together, and finally pretend as one big client to cheat in the proposed incentive mechanism? (maybe good to call out the limitation of incentive mechanisms without monetary transfers)

---

> ### Author Response · Authors · 2023-11-14
> **Thank you for the reviews (1/2)**
>
> We thank the reviewer for taking the time to review our paper and for acknowledging the significance of the problem we studied and our solid results.
>
> We hope addressing the concerns as follows will improve your opinion of our work.
>
> >W1: "A positive incentive ... might not be enough" with "costs", not "100% effort"
>
> We indeed only have a relative guarantee for the proposed mechanism instead of absolute ones, i.e., clients have more incentives to contribute to IAFL as compared to FL.
>
> We assumed that your "positive incentive" refers to **the increase in utility**. In our paper, the utility is defined as "rewards minus costs". Therefore, our incentives design already takes cost $c_i$ of client $i$ into consideration. With that, incentivizing clients to contribute 100% effort is indeed the most ideal, but might not be achievable because costs are incurred by the clients to contribute more. A more practical perspective we have adopted in the paper is that a client will increase his contribution to a point where the gradient of utility is 0, i.e., the increment in reward balances the increment in cost. More specifically, in Proposition 1, we measure the gradient of the utility w.r.t. contribution $p_i$ to determine the client contribution level and show that a practical incentive guarantee with the consideration of client costs.
>
> >W2 & Q1: "Tradeoff between the strength of incentive and the loss of (center) model accuracy"
>
> By "strength of incentives", we assume you mean the **incentivization performance** in the paper. As for the "loss of (center) model accuracy", we understand it as the loss of client v.s. the (center) model, reflected as client model accuracies.
>
> Then, the two observations mentioned in the review align with our discussion in the paper in a slightly different form.
>
> - About "$p_{ceil}$"
>
> The reviewer's comments on $p_{ceil}$ is valid. It is worth noting that we have discussed the issue in the paragraph after Equation (1):
> “One limitation of this design is that the clients lose incentive for $p_i > p_{ceil}$. The server can mitigate this by setting a high $p_{ceil}$ upper limit (to keep incentivizing high contributors) while setting a larger $\kappa$ (to even out the fractions $(p_i/p_{ceil})^{1-\kappa}$ towards 1 to motivate low contributors).” Please let us know whether the discussion is sufficient.
>
> - About "$\kappa$" and "$q$"
>
> We would like to stress the importance of $\kappa$ and $q$ in the trade-off identified by the reviewer. We have discussed them in Section 6.1 of the main paper and we additionally have a dedicated section Appendix E.7 to discuss the **trade-off between model performance and client incentivization** (directly affected by $\kappa$ and $q$). In essence, decreasing $\kappa$ and $q$ results in stronger incentives. When the strength of incentives is stronger, we have higher incentivization performance (measured as IPR and $\rho$ defined in the paper) and lower client model accuracies (equivalent to higher loss of model accuracy).
>
> In summary, we have established the trade-off in another form in the paper, which can relate to the loss of center model and strength of incentives trade-off. We will add this discussion to the revised paper.

---

> ### Author Response · Authors · 2023-11-14
> **Thank you for the reviews (2/2)**
>
> >Q2: "Fairness concern for small and large clients"
>
> There are two prevalent concepts of fairness in literature: (a) collaborative fairness [1] (i.e., contribute more get back more), and (b) equality [2]. To be fair, people from one of the camps will not view the other as fair.
>
> In our paper, we have demonstrated results for the collaborative fairness view. Specifically, the contributions of clients are fairly evaluated by a contribution evaluation measure, e.g., CGSV (Xu et al., 2021), and then used to fairly reward the clients with a model that has performance commensurate with their contributions.
>
> If the designer of the FL process views fairness more towards equality such that "if everyone puts in their best efforts, they should be treated equally", it is possible to use "individual efforts percentage" (e.g., how many percentages a client contributes to his/her best effort, as suggested by the reviewer) as a contribution evaluation measure (assuming that individual efforts of clients can be measured). This measure can be seamlessly combined with the IAFL framework, too. However, we would like to clarify that "individual efforts percentage" is not commonly used as a contribution measure in literature.
>
> Thank you for the interesting insights and unique perspectives.
>
> References:
>
> [1] L. Lyu, X. Xu, Q. Wang, and H. Yu. Collaborative fairness in federated learning. In Q. Yang, L. Fan, and H. Yu, editors, Federated Learning, volume 12500 of Lecture Notes in Computer Science, pages 189–204. Springer, Cham, 2020.
>
> [2] T. Li, M. Sanjabi, A. Beirami, and V. Smith. Fair resource allocation in federated learning. In Proc. ICLR, 2020.
>
> >Q3: "Small clients may have the incentive to cooperate to pretend as one big client to receive a better model"
>
> Our framework does not cater to the scenario raised by the reviewer and it may require further work to incorporate this.
>
> Without further assumptions (e.g., there are clients who are not self-interested, the clients have easy access to the platform, etc.), it will be challenging to establish these platforms. Questions will arise: How will the participants of the platform split the rewards? Is the platform solely non-profit seeking? One may argue that such a platform does not require friendly clients to work together. However, if the participants are indeed self-interested, setting up such a platform reduces to our self-interested problem setting to begin with.
>
> We acknowledge the validity of the useful point raised by the reviewer and anticipate further investigation in future work.
>
> >To conclude
>
> To conclude, we would like to thank the reviewer for the interesting insights and perspectives. We will include the necessary contents discussed in the revised paper and hope that our clarifications have improved your opinion about our work.

---

> ### Author Response · Authors · 2023-11-22
> **Follow-up**
>
> Dear Reviewer xzxw,
>
> Thank you again for your time in reviewing our paper and for the interesting insights and perspectives with regard to our work.
>
> Please let us know whether our replies have sufficiently addressed your concerns. We will be happy to engage further within the discussion period.
>
> Best regards,
>
> Authors of Paper 5535

---

> ### Author Response · Authors · 2023-11-23
> **A summary and kind reminder**
>
> Dear Reviewer xzxw,
>
> Thank you again for your time in reviewing our paper. In the rebuttal, we have made the following major clarifications to your questions:
>
> - We have a relative guarantee for the proposed mechanism which already considers the role of costs in the client incentives.
> - The trade-off between incentivization and model performance is discussed in the paper in a slightly different form.
> - The paper demonstrated fairness results for the collaborative fairness view.
>
> As today is the last day of the discussion period, please let us know whether our replies have sufficiently addressed your concerns. We will be happy to engage further within the discussion period.
>
> Best regards,
>
> Authors of Paper 5535

---

### Official Review · Reviewer_dsGp · 2023-11-03

**Soundness:** 4 excellent
**Presentation:** 4 excellent
**Contribution:** 3 good
**Rating:** 5
**Confidence:** 2

**Summary:**

The paper proposes an incentive-aware federated learning algorithm that encourages client contribution by training-time rewards. Concretely, the authors propose a local reward scheme to ensure that a higher-contributing client receives a better final model.

**Strengths:**

The scope of the experiment is extensive. The authors experiment with different data partition methods, different metrics for measuring incentives, and benchmark against various baselines.

**Weaknesses:**

See questions.

**Questions:**

The problem this paper studies is interesting. However, it could be that I'm missing something, in Theorem 1 and Theorem 2, does convergence speed become faster as the number of agents $N$ grows? It would be helpful to simplify the bound and make the dependence on $N$ explicit. Does adding more clients lead to a faster convergence rate? I would happily increase my score if the question is addressed.

---

> ### Author Response · Authors · 2023-11-14
> **Thank you for the reviews**
>
> We greatly appreciate the reviewer's time and effort in providing constructive feedback on our paper, and we are encouraged by the high scores assigned to the soundness and presentation of our paper. We are also delighted to see that our scope of experiments is extensive.
>
> We hope that our responses to your concerns as follows will further enhance your perception of our work.
>
> >Q1: "Does convergence speed become faster as the number of agents $N$ grows?"
>
> The simple answer is, adding clients that **"help than hurt"** is going to improve the convergence speed.
>
> To elaborate, we first observe that it might be difficult to further simplify the bound because it has a dependence on every single client's contribution $\gamma_{m,t}'$ for $m \in [N]$. However, we can derive some insights on the convergence speed as $N$ grows.
>
> The effect depends on the **contribution levels of the clients**.
> We can look at the term $C_T = \frac{H_T}{N} \sum_{m=1}^N \sum_{t=1}^{T} \left(\frac{T+\alpha}{t+\alpha}\right)^2 \left(\frac{1}{\gamma_{m,t}'} + \frac{1}{\gamma_{i,t}'} -2 \right)$, when $N$ increases, the denominator becomes bigger but at the same time there are additional terms in the summation.
>
> We can consider a case where we increase $N=n$ to $N=n+1$. Abstracting away the constants $H_T$, $T$ and $\alpha$ for simplicity, we can make a comparison for $C_T = \frac{1}{n} \sum_{m=1}^{n} \left(\frac{1}{\gamma_{m,t}'} + \frac{1}{\gamma_{i,t}'} - 2 \right)$ when $N=n$ against the case for $C_T = \frac{1}{n+1} \sum_{m=1}^{n+1} \left(\frac{1}{\gamma_{m,t}'} + \frac{1}{\gamma_{i,t}'} - 2\right)$ when $N=n+1$. Note that $\gamma_{i,t}'$ can also be treated as a constant here.
>
> Informally speaking, we observe that when $N$ increases, the term $C_T$ generally decreases in its scale if the added client has contribution $\gamma_{m,t}'$ at least the **“average contribution”** of the existing clients $\frac{1}{n} \sum_{m=1}^{n} \gamma_{m,t}'$. Therefore, the decrease in $C_T$ tightens the performance bound in Theorem 1 and makes the convergence faster.
>
> An exception that we can derive from the above is that when the added clients contribute badly with a very low $\gamma_{m,t}'$, it could slow down the convergence despite the increase in $N$. In practice, this implies that the model update provided by the added client is bad, harmful, or even adversarial, and is detected by the contribution evaluation measure. Theoretically, the convergence is adversely affected if we add "harmful" clients. However, we could easily implement client selection methods based on this insight on contribution evaluation to filter such clients during the training process to ensure faster convergence when $N$ grows.
>
> Concluding, we hope our discussion above about the impact of $N$ on convergence speed will improve your opinion about our work.

---

> ### Author Response · Authors · 2023-11-22
> **Follow-up**
>
> Dear Reviewer dsGp,
>
> Thank you again for your time in reviewing our paper and for asking an insightful question about the convergence speed with respect to the number of clients.
>
> Please let us know whether our replies have sufficiently addressed your concerns. We will be happy to engage further within the discussion period.
>
> Best regards,
>
> Authors of Paper 5535

---

> ### Author Response · Authors · 2023-11-23
> **A summary and kind reminder**
>
> Dear Reviewer dsGp,
>
> Thank you again for your time in reviewing our paper. In the rebuttal, we have made the following major clarifications to your questions:
>
> - The convergence speed can relate well with the number of agents. Adding clients that "help than hurt" is going to improve the convergence speed. We also provide detailed case studies in the response above.
>
> As today is the last day of the discussion period, please let us know whether our replies have sufficiently addressed your concerns. We will be happy to engage further within the discussion period.
>
> Best regards,
>
> Authors of Paper 5535

---

### Official Review · Reviewer_LXR6 · 2023-11-05

**Soundness:** 3 good
**Presentation:** 2 fair
**Contribution:** 2 fair
**Rating:** 3
**Confidence:** 3

**Summary:**

This paper studies incentive mechanism for federated learning. Existing works in this direction typically incentive clients via post-training monetary rewards. The authors argue that, clients may anticipate timely rewards during FL process, and may decide to quit when not being properly incentivized. Moreover, monetary rewards may be infeasible in some situations, e.g., when revenue is unclear or budget is limited. Therefore, the authors propose a new formulation, where the clients are reward during the FL process in the form of global model updates of varying quality, depending on the contribution of each client. The authors derive a convergence guarantee of the proposed method, where the convergence rate of each client depends on its reward rate $\gamma_{i,t}$.

**Strengths:**

The idea of providing incentives during the FL process instead of postponing to the end of FL is novel and well-motivated.

**Weaknesses:**

1. What it means for a client to be incentivized is not well-defined in this paper.

From Proposition 1, it seems as long as the gradient of client $i$'s utility w.r.t. its contribution $p_{i}$ is higher than that under standard FL mechanism, we say the client $i$ is incentivized. It is not clear why we, as the designer of the mechanism, cares about whether the gradient of utility for each client is higher than what the client gets under a standard FL mechanism. Instead, a more natural goal is to incentivize the clients to contribute to FL using their full capacity in order to get the best learning outcome.

In this regard, the intrinsic cost $c_{i}$ of each client also plays an important role, i.e., it is possible that the cost value is high, such that we end up with a negative gradient of the utility (contributing more leads to even lower utility). Therefore, a rational client will decide to contribute $p=0$ in this case, which affects the convergence of the FL process.

2. Current convergence analysis over-simplifies the effect of contribution level on local gradients

Due to the simplification of the "contribution measurement" mentioned in Section 4.2, the current convergence result given in Theorem 2 is independent from client's behavior model. Currently, the only place that contribution level of a client plays a role in the convergence result, is the reward rate $\gamma_{i,t}$ (the quality of the global model that the server decides to give to this client). However, the contribution level should also affect the quality of the local gradient that client provides to the server, e.g., lower contribution means computing the local gradient using smaller portion of its local data (Other than simply saying the client will always faithfully compute the full local gradient w.r.t. the given global model).

In the extreme case mentioned in my first comment, where the client decides to make zero contribution, then the server will not get the local gradient from this client. However, in the current analysis, the authors assume that the server can always get the local gradients of all the clients no matter what, which does not seem to be reasonable.

**Questions:**

Can the authors elaborate on, in Theorem 1, why $o(1/\gamma_{i,T}^{\prime})$ suffices to make the convergence hold? Where did the $(T+\alpha)^{2}$ in the numerator go?

---

> ### Author Response · Authors · 2023-11-14
> **Thank you for the reviews (1/2)**
>
> We would like to express our gratitude to the reviewer for taking the time to review our paper and for acknowledging that our training-time incentive during FL is novel and well-motivated.
>
> We hope addressing the raised concerns as follows will improve your opinion of our work.
>
> >W1(a): "What it means for a client to be incentivized"
>
> In game theory, incentives are typically based on the potential for gaining a benefit or circumventing a cost. Consequently, a rational client is incentivized to perform an action when there is a positive gain in utility for doing so. We formalize this as Individual Rationality (IR) in Definition 1, implying that clients are incentivized to participate in our IAFL scheme, because every client will receive a reward at least as good as what they can achieve on their own. IR is an established concept for player incentivization in game theory.
>
> >W1(b): "A more natural goal is to incentivize the clients to contribute to FL using their full capacity"
>
> Incentivizing clients to contribute to the "full capacity" for the best learning outcome is indeed the most ideal, like the reviewer has said. However, every client incurs costs to contribute and the actual contribution of clients depends on the interplay between rewards and costs for different contribution levels.
>
> Hence, in our work, we propose the next achievable alternative: To incorporate the goal of **"incentivizing clients to contribute as much as possible"** on top of satisfying IR. To this end, we designed IAFL such that Theorem 1 and 2 are fulfilled. These theorems suggest that clients will be rewarded with better models if they contribute more. This will incentivize clients to further contribute to their full capacity for improved model performance received, of course, subject to their marginal utility increment in the presence of a cost $c_i$. Clients will contribute up till the marginal increment in reward still surpasses the associated cost.
>
> This is **important to the FL designer** because higher contributions from clients (e.g., in the form of better gradient updates, more frequent participation, etc.) could potentially lead to improved model performance for all clients (i.e., a better learning outcome).
>
> >W1(c): "The intrinsic cost $c_i$ of each client also plays an important role"
>
> The reviewers' observations regarding the critical role of $c_i$ are indeed accurate. As specified by Proposition 1, in the presence of a cost $c_i$, our IAFL incentivizes more than the standard FL.
>
> When $c_i$ is indeed high, our IAFL is still better than standard FL, since this cost is borne by both IAFL and standard FL. A client with a high cost $c_i$ will not participate in either IAFL or standard FL. Likewise, in the event of a strictly $p=0$ contribution, we can simply leave the client out of the collaboration to avoid detrimental effects on collaborative model training and convergence. Alternatively, the sharing parameter $\kappa$ presented in Equation (1) of the paper could be raised to enhance the convergence of the client models.
>
> Therefore, this weakness is not exclusive to our framework, and notably, our IAFL even offers potential remedies. In practice, if a rational client finds it possible to benefit from standard FL, they will definitely benefit more from our framework. Hence, our framework is deemed reasonable.

---

> ### Author Response · Authors · 2023-11-14
> **Thank you for the reviews (2/2)**
>
> >W2(a): "Current convergence result is independent from client's behavior model" and "the contribution level should also affect the quality of the local gradient"
>
> The contribution measures we refer to in Section 4.2 captures the quality of the local gradient as contributions, e.g., FedSV (Wang et al., 2020b), ComFedSV (Fan et al., 2022), CGSV (Xu et al., 2021), FedFAIM (Shi et al., 2022).
>
> Thus, the relationship might need to be turned the other way around. The client's behavior model decides the quality of the local gradient being computed (e.g., using different portions of its local data). Then, the quality of the local gradient affects the client contribution level (as assessed by the contribution evaluation measures mentioned above), which in turn affects the reward rate in IAFL. Subsequently, the reward rates of clients affect the model convergence.
>
> Hence, the current convergence results capture the client's decision on the contribution level through deciding the quality of their local gradients.
>
> >W2(b): "Other than simply saying the client will always faithfully compute the full local gradient"
>
> In the current analysis, we hope to clarify that we do not place an assumption on what kind of gradients we can obtain from the clients. We do not assume that clients faithfully compute and upload the full local gradients.
>
> >W2(c): "Extreme case" about "zero contribution"
>
> If a client decides to make zero contribution, this will be **captured by the contribution evaluation measure**. Then, the client does not need to upload gradients and it will not affect our IAFL algorithm.
>
> This is a simple fix for people with zero contribution: If the client is withdrawing, he/she should not be in the analysis to begin with. As a result, the extreme case does not affect the current analysis. We would like to thank the reviewer for the insightful observation, we will add a discussion on this matter in the revised paper.
>
> >Q1: "Why $o(1/\gamma_{i,T}')$ suffices to make the convergence hold" and "where did the $(T+\alpha)^2$ go"
>
>
> We are sorry that there is an unintentional typo that occurred while organizing the formula to make it more concise, the term $T^2$ was accidentally omitted.
>
> Referring to the paragraph on "error propagation and non-convergence", instead of $(H_T/T)\sum_{t=1}^T 1/t^2=O(T)$, it should be $(H_T/T)\sum_{t=1}^T T^2/t^2=O(T)$.
>
> With $H_T T^2/T = H_T T = O(T)$ and $1/{\gamma_{i,T}'}= o(1/T)$, they are jointly required for the convergence to 0. This typo does not affect any theoretical results in Theorem 1 or any subsequent theoretical results, and we hope to seek understanding from the reviewer for any unintentional confusion caused.
>
> > To conclude
>
> Overall, we hope that we have convinced you that our framework is reasonable. By clarifying the misunderstandings, the issues about (1) the incentivization goal and (2) the effect of client contribution on the convergence should have been resolved.
>
> We really hope that we have addressed your concerns and improved your opinion of our work.

---

> ### Author Response · Authors · 2023-11-22
> **Follow-up**
>
> Dear Reviewer LXR6,
>
> Thank you again for your time in reviewing our paper and for asking many insightful questions.
>
> Please let us know whether our replies have sufficiently addressed your concerns. We will be happy to engage further within the discussion period.
>
> Best regards,
>
> Authors of Paper 5535

---

> ### Author Response · Authors · 2023-11-23
> **A summary and kind reminder**
>
> Dear Reviewer LXR6,
>
> Thank you again for your time in reviewing our paper. In the rebuttal, we have made the following major clarifications to your questions:
>
> - The goal of incentivization is already defined and discussed with the consideration of client costs in the paper. We adopt the next most pragmatic goal of "incentivizing clients to contribute as much as possible" on top of satisfying IR.
> - Our current convergence analysis captures the client's decision on the contribution level through deciding the quality of their local gradients, hence affecting the model convergence.
>
>
> As today is the last day of the discussion period, please let us know whether our replies have sufficiently addressed your concerns. We will be happy to engage further within the discussion period.
>
> Best regards,
>
> Authors of Paper 5535

---

### Meta-Review · Area_Chair_FkNh · 2023-12-11

**Metareview:**

Thank you for your paper and diligently responding to reviewers' questions. All reviewers appreciate the paper's novel approach to providing incentives in FL. This proposed IAFL is also more compelling than prior approaches that provide monetary payments. After reading through the authors' responses, the AC believes that the reviewers' questions have been addressed. While there is still room for improvement (such as a more in-depth theoretical analysis of incentive-accuracy trade-offs), the paper opens an entirely new approach to studying incentives in FL, which may motivate more future work.

**Justification For Why Not Higher Score:**

The paper is at the borderline of acceptance.

**Justification For Why Not Lower Score:**

This paper introduces a new approach to thinking about an important research question.

---

### Decision · Program_Chairs · 2024-01-16

Accept (poster)